# Understanding Scaling Laws in Deep Neural Networks via Feature Learning Dynamics

## Abstract

The empirical success of deep learning is often attributed to scaling laws that predict consistent performance gains as model, data, and compute increase. However, large models often suffer severe training instability and diminishing returns, indicating that scaling laws only describe ***what*** success looks like but not ***when* and *why* scaling succeeds or fails**. A central barrier is the lack of a rigorous understanding of feature learning at ***large depth***: while $\mu$P provides a principled characterization of feature learning dynamics in the infinite-width limit and enables hyperparameter (HP) transfer across width, its depth extension, *i.e.*, depth-$\mu$P, faces critical challenges, especially in residual blocks with more than one internal layer. In this paper, we address this gap by deriving the ***Neural Feature Dynamics (NFD)***, a coupled forward-backward stochastic system that rigorously characterizes the training dynamics of ResNets in the joint infinite-width and infinite-depth limit. NFD reveals when scaling laws hold, explains diminishing returns, and shows that the ***gradient-independence assumption (GIA)***, known to fail during training at finite depth, becomes provably valid again at infinite depth, identifying a new regime where end-to-end feature learning remains ***tractable*** for analysis. Moreover, NFD uncovers a structural cause of the failure of depth-$\mu$P: representation learning collapses in the first layer of two-layer residual blocks. Motivated by this insight, we introduce a simple ***depth-aware learning-rate correction*** that restores depth-wise HP transfer and yields overall stronger performance.

## 1 Introduction

The remarkable success of deep neural networks (DNNs) has been driven by a key empirical observation, known as *scaling laws* (Kaplan et al., 2020; Hoffmann et al., 2022), which predicts consistent performance gains as model, data, and compute increase. These principles have guided the development of many large-scale models, including large language models (LLMs) (Achiam et al., 2023), vision transformers (ViTs) (Dosovitskiy et al., 2021), and deep generative models (Song et al., 2020). However, training these massive models is not without challenges. As models grow larger, especially deeper, they often encounter severe training instabilities (e.g., loss spikes and exploding gradients) (Le Scao et al., 2022; Chowdhery et al., 2023) and exhibit diminishing returns in performance (Kaplan et al., 2020; Hoffmann et al., 2022). These practical limitations highlight that empirical scaling laws only describe *what success looks like*, but do not explain *when and why scaling succeeds or fails*. Addressing this question requires a *rigorous analysis of training dynamics at scale*, particularly how feature learning occurs during training. However, most theoretical works on scaling laws focus on predicting scaling exponents using highly simplified solvable models (Bahri et al., 2024; Maloney et al., 2022; Simon et al., 2024; Bordelon et al., 2024a; Paquette et al., 2024), rather than the mechanisms for stable and nontrivial training dynamics in DNNs.

A prominent theoretical framework is the Neural Tangent Kernel (NTK) theory. In the large-width limit, neural networks evolve linearly around their initialization (Lee et al., 2019) and behave as kernel methods governed by the fixed NTK (Jacot et al., 2018b). While NTK explains why wide networks can perfectly fit training data and still generalize well (Du et al., 2019; Arora et al., 2019), it lies in the *lazy training* regime (Chizat & Bach, 2019), where neural features remain essentially fixed. Consequently, NTK cannot account for the rich representation learning that underpins modern breakthroughs such as in-context learning (Brown et al., 2020; Gao et al., 2021), multi-modal learning (Radford et al., 2021), and chain-of-thought reasoning (Wei et al., 2022). To address this limitation, a complementary line of work explores alternative network parameterizations in the large-width limit. Mean-field analyses (Mei et al., 2018) and the maximal update parameterization ($\mu$P) (Yang & Hu, 2021) establish width-wise scaling rules that preserve active *feature-learning* for DNN

training even at scale. Notably, $\mu$P also provides the practical benefit of Hyperparameter (HP) transfer across network widths (Yang et al., 2021). This success is rooted in its rigorous characterization of *feature-learning dynamics in the infinite-width limit* (Yang & Hu, 2021).

Despite its success in the width dimension, $\mu$P faces a critical crisis in the depth dimension: it does not provide stable training and HP transfer across network depths (Yang et al., 2024; Bordelon et al., 2024c). To mitigate this issue, recent works have proposed *depth-$\mu$P*, which applies a $1/\sqrt{L}$ scaling to the residual branch of a depth-$L$ ResNet (He et al., 2016a). While this approach initially showed promise for large-depth training and depth-wise HP transfer, subsequent studies revealed that it breaks down whenever residual blocks contain more than one internal layer (Yang et al., 2024). These failures have motivated alternative depth-scaling strategies (e.g., $1/L$) (Dey et al., 2025), and highlight a deeper issue: unlike width-wise $\mu$P, for depth-$\mu$P, we currently lack a rigorous characterization of *feature-learning dynamics in the infinite-depth regime*. While existing works provide useful insights, either from forward propagation analyses at initialization (Hayou & Yang, 2023) or from physics-inspired heuristic reasoning (Bordelon et al., 2024c), they do not offer a complete and principled understanding of *how depth reshapes feature and gradient evolution during training.*

To fill this gap, we develop a rigorous characterization of *feature-learning dynamics in the joint infinite-width and infinite-depth limit* by analyzing randomly initialized ResNets trained with SGD under depth-$\mu$P, augmented with a time-horizon $T$ to balance model capacity and training stability (see Section 3). In this setting, we establish a *coupled forward-backward stochastic differential equation (SDE) system* that governs the evolution of features and gradients during training, and use it to uncover new *depth-driven* phenomena and their practical consequences. Our main contributions are summarized as follows:

- **Pre-act vs. post-act debate in deep ResNets.** We first show that the post-act ResNet can diverge[1] even under the $1/\sqrt{L}$ scaling used in depth-$\mu$P for common activation functions (e.g., ReLU). In contrast, the pre-act design remains stable and admits a well-defined infinite-depth limit. This resolves an open architectural question and justifies the focus of our analysis on the pre-act variant.

- **Neural Feature Dynamics (NFD): a forward–backward SDE for training at infinite depth.** Under depth-$\mu$P in ResNets, we prove that as width $n$ and depth $L$ tend to infinity, the forward features and backward gradients converge to the solutions of a coupled forward–backward SDE system, named *Neural Feature Dynamics* (NFD). This provides, to our knowledge, the first *mathematically rigorous characterization* of feature-learning dynamics in the infinite-depth regime.

- **Independent Brownian motions (BMs) and restoration of GIA during training.** We show that the forward and backward components of NFD are driven by independent Brownian motions. This implies that the *gradient-independence assumption (GIA)*–a widely used heuristic in wide-network analysis that is rigorously justified only at initialization (Yang, 2020b) and proven to fail during training at finite depth (Yang & Hu, 2021; Bordelon et al., 2024a)–becomes *provably valid again throughout training* in the infinite-depth limit. This identifies a new regime where feature learning remains *tractable* for theoretical analysis.

- **A new vanishing phenomenon in two-layer residual blocks and a simple fix.** NFD reveals a structural separation in two-layer residual blocks: the first layer is responsible for *internal representation learning*, while the second governs the *residual-stream dynamics*. This view makes visible an FL collapse that is easily overlooked when only monitoring raw weight updates. Through this lens, we uncover the mechanism behind the long-observed failure of depth-$\mu$P in two-layer residual blocks: under depth-$\mu$P, the feature updates in the first layer vanish as depth grows, causing internal FL to collapse even though the residual-stream dynamics remain active. Guided by this insight, we introduce a *depth-aware learning-rate correction* that restores effective FL, recovers HP transfer across depth, and significantly improves empirical performance.

- **Capacity limits and insights into scaling behavior.** NFD shows that the success or failure of scaling depends on whether the network parameterization yields a well-posed limiting dynamics. When this holds, increasing width or depth improves performance by reducing the approximation error to the NFD limit, so model capacity increases as the finite network moves closer to this limit. However, the increasing expressive power of the finite network is capped by the NFD limit, which naturally produces diminishing returns as width or depth grows and the model saturates the capacity of the limit. The time horizon parameter $T$ can further enlarge the capacity of the limiting dynamics, but this comes at the cost of reduced training stability.

---

[1]Unless additional corrective operations are applied (see, *e.g.*, (Yang et al., 2024)).

## 2 PRELIMINARIES

**Depth-adapted ResNets.** We consider a ResNet processing input $\boldsymbol{x} \in \mathbb{R}^d$ via a residual stream:

$$f(\boldsymbol{x}; \boldsymbol{\theta}) = \tfrac{\alpha}{\sqrt{n}} \boldsymbol{v}^\top \boldsymbol{h}_L, \quad \boldsymbol{h}_\ell = \boldsymbol{h}_{\ell-1} + \sqrt{\tfrac{T}{Ln}} \boldsymbol{W}_\ell \, \phi(\boldsymbol{h}_{\ell-1}), \quad \boldsymbol{h}_0 = \tfrac{1}{\sqrt{d}} \boldsymbol{U} \boldsymbol{x}, \quad \ell \in [L], \quad (1)$$

where $\alpha > 0$ controls the training regime[2], $\phi$ is an activation function, $\boldsymbol{U} \in \mathbb{R}^{n \times d}$, $\boldsymbol{W}_\ell \in \mathbb{R}^{n \times n}$, and $\boldsymbol{v} \in \mathbb{R}^n$ are trainable parameters. These parameters are initialized from widely used random initialization (Glorot & Bengio, 2010; He et al., 2015): $\boldsymbol{v}_i, \boldsymbol{W}_{\ell,ij}, \boldsymbol{U}_{ij} \overset{\text{i.i.d.}}{\sim} \mathcal{N}(0,1)$. The scaling $1/\sqrt{n}$ in Eq. (1) is a standard mean-field scaling that keeps each coordinate of the features $\boldsymbol{h}_\ell$ at $\Theta(1)$ scale as $n \to \infty$. Similarly, the $\sqrt{T/L}$ follows prior work on depth scaling (Hayou et al., 2021; Marion et al., 2025) and ensures stability as depth $L \to \infty$. We augment this with the time-horizon parameter $T$, and show in Section 3 that increasing $T$ expands the model's capacity, whereas increasing width or depth only reduces the approximation error to the NFD limit. Given a loss $\mathcal{L}$, the ResNet is trained via online SGD, *i.e.*, $\boldsymbol{\theta}^{(k+1)} = \boldsymbol{\theta}^{(k)} - \eta \nabla_{\boldsymbol{\theta}} \mathcal{L}(f^{(k)}, y^{(k)})$, where $f^{(k)} := f(\boldsymbol{x}^{(k)}; \boldsymbol{\theta}^{(k)})$ is evaluated on a single data point $(\boldsymbol{x}^{(k)}, y^{(k)})$ sampled at iteration $k$.

**Tensor Programs (TPs).** The TP framework (Yang & Hu, 2021) provides a unified language for expressing forward and backward computations of DNNs and analyzing their infinite-width limits. Formally, a TP is a sequence of vectors and scalars recursively computed from an initial set of random parameters using basic operations (*e.g.*, MatMul, Nonlin, Moment). In Eq. (1), the weight matrices $\boldsymbol{W}_\ell$ belong to the initial set, while the features $\boldsymbol{h}_\ell$ are program variables. Crucially, the TP structure remains valid during training, as long as each recomputed variable can be expressed via valid TP operations on previously valid variables.[3]

A central theoretical result in TP theory is the *Master Theorem* (Yang & Hu, 2021, Theorem 7.4) that characterizes the behavior of TP program variables in the infinite-width limit. Specifically, for any finite collection $\{\boldsymbol{h}_s\}_{s=1}^M \in \mathbb{R}^n$ and sufficiently regular function $\psi : \mathbb{R}^M \to \mathbb{R}$,

$$\frac{1}{n} \sum_{i=1}^n \psi\left(\boldsymbol{h}_{1,i}, \ldots, \boldsymbol{h}_{M,i}\right) \xrightarrow{a.s.} \mathbb{E}\left[\psi(Z^{\boldsymbol{h}_1}, \ldots, Z^{\boldsymbol{h}_M})\right], \quad \text{as } n \to \infty,$$

where $Z^{\boldsymbol{h}_s}$ denotes the *mean-field limit* of $\boldsymbol{h}_s$. This result enables a rigorous analysis of feature learning dynamics in the infinite-width limit (Yang & Hu, 2021) for networks with depth $L = \Theta(1)$.

**Maximal Update Parameterization ($\mu$P).** The TP framework and its Master Theorem classify any network parameterization by its behavior in the large-width limit, a classification known as the *Dynamical Dichotomy* (Yang & Hu, 2021): *every stable parameterization lies in either the kernel regime or the FL regime, but **not both***. In the FL regime, $\mu$P (Yang & Hu, 2021) uniquely maximizes feature evolution per gradient update and enables HP transfer across network widths (Yang et al., 2021). Depth-$\mu$P extends this idea to the depth dimension by introducing a $1/\sqrt{L}$ scaling on the residual branch (Yang et al., 2024; Bordelon et al., 2024c). However, prior studies also noted that depth-$\mu$P maintains stable scaling and HP transfer only for one-layer residual blocks, while deeper residual blocks exhibit degraded behavior (Yang et al., 2024; Bordelon et al., 2024c), motivating the need for a principled understanding of feature learning in the infinite-depth regime.

**Gradient Independence Assumption (GIA).** The GIA is a widely used heuristic in wide-network analysis, including signal propagation, NNGP, and NTK (Schoenholz et al., 2017; Lee et al., 2018; Jacot et al., 2018a), that treats $\boldsymbol{W}_\ell^\top$ in backpropagation as independent from the forward weight $\boldsymbol{W}_\ell$. This assumption is rigorously justified in the infinite-width limit at initialization, provided the output layer is not used anywhere else in the interior of the network (Yang, 2020b). However, the GIA has been rigorously proven to ***fail during training*** at finite depth (Yang & Hu, 2021), which makes full training-dynamics analysis in the FL regime intractable. A key implication of our infinite-depth NFD analysis is that GIA is ***restored throughout training*** in the infinite-depth limit, enabling a tractable characterization of feature-learning dynamics, beyond NTK.

---

[2]Here, $\alpha = 1$ corresponds to NTK regime, while $\alpha = 1/\sqrt{n}$ to the FL regime (Yang & Hu, 2021).

[3]For example, after $k$ gradient updates, the feature vector $\boldsymbol{h}_\ell$ in the $k$-th forward pass Eq. (4) remains TP-valid, as it can be recursively constructed from earlier variables using only TP operations.

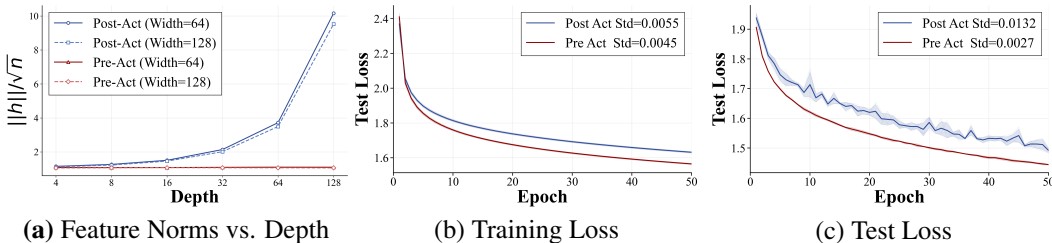

**(a)** Feature Norms vs. Depth    (b) Training Loss    (c) Test Loss

Figure 1: **Pre- and post-act ResNets debate under depth-$\mu$P.** In **(a)**, the pre-act variant maintains stable feature across depth, whereas the post-act exhibits rapid growth. In **(b)-(c)**, we train depth-64 width-128 ResNets with ReLU on CIFAR-10 under SGD (LR 0.01, batch size 128). The stability from pre-act design yields faster convergence and lower test loss with reduced variance across runs.

## 3 The Forward SDE Limit and Network Capacity at Initialization

We first analyze forward feature propagation at initialization and show that the capacity gains from increasing network size are ultimately capped by the SDE limit, explaining the diminishing returns when scaling up models. Moreover, we also demonstrate that increasing the SDE's time horizon $T$ can further expand the capacity, providing a principled way to bypass this fundamental ceiling.

### 3.1 The Pre- vs. post-act Debate

It is important to distinguish the two common ResNet designs: the *pre-act* style (He et al., 2016b), as defined in Eq. (1), and the *post-act* style, where the skip connection is applied after the $\phi$; see Appendix B for details. Although the post-act design was first introduced in the original ResNet paper (He et al., 2016a), the pre-act variant has become the preferred choice in modern deep networks, such as the Transformer architecture in GPT series (Radford et al., 2019; Brown et al., 2020).

However, this preference has been guided more by practice than theory. To address this, we provide a theoretical justification for the pre-act preference by showing that post-act ResNets can exhibit divergent hidden states for a broad class of activation functions as depth increases.

**Proposition 1.** *Let $\phi$ satisfy the following **positive dominance** condition: there exist nonnegative constants $c_1, c_2$, not both zero, such that*

$$\mathbb{E}[\phi(wx)] \geq c_1|x| + c_2, \quad \forall x \in \mathbb{R},$$

*where $w$ is the standard Gaussian random variable. Then, in a post-act ResNet as defined in Eq. (9), the expected hidden state satisfies:*

$$\mathbb{E}[\boldsymbol{h}_{L,i}] \geq c_1(1 + c_1\sqrt{T/Ln})^L \|\boldsymbol{x}\|/\sqrt{d} + c_2\sqrt{TL}, \quad \forall i \in [n].$$

This result implies that with activations such as ReLU ($c_1 = 1/\sqrt{2\pi}$, $c_2 = 0$), the features $\boldsymbol{h}_\ell$ may diverge as depth grows, even with the stabilizing factor $\sqrt{T/L}$ scaling. This explains why analyses of post-act ResNets often require extra normalization or finely balanced depth-width scaling (Peluchetti & Favaro, 2020; Yang et al., 2024; Li et al., 2022). Figure 1 illustrates this behavior.

### 3.2 Neural Feature Propagation as an SDE

Given the superior stability, the subsequent analysis focuses on the pre-act design. We use the synchronous coupling method to prove that, in the joint limit, each coordinate of the hidden state $\boldsymbol{h}_\ell$ can be viewed as a particle whose mean-field dynamics are characterized by a forward SDE.

**Proposition 2.** *Suppose $\phi$ is $K_1$-Lipschitz. In the joint limit $\min(n, L) \to \infty$, each coordinate of feature vectors $\boldsymbol{h}_\ell$ converges to the solution $h_t$ of a McKean–Vlasov SDE*

$$dh_t = \sigma_t \, dw_t, \quad h_0 \sim \mathcal{N}(0, \|\boldsymbol{x}\|^2/d), \quad \forall t \in [0, T], \tag{2}$$

*with $\sigma_t^2 := \mathbb{E}[\phi^2(h_t)]$ and a standard Brownian motion $\{w_t\}$, in mean-square convergence at rate*

$$\mathbb{E}|\boldsymbol{h}_{\ell,i} - h_{t_\ell}|^2 \leq C\left(L^{-1} + n^{-1}\right), \quad \forall \ell \in [L], \; i \in [n],$$

*where $t_\ell = \ell T/L$, and the constant $C > 0$ does not depend on width $n$ or depth $L$.*

This result generalizes (Hayou & Yang, 2023), which was restricted to ReLU, by establishing convergence bounds and implying width–depth commutativity for any Lipschitz continuous $\phi$.

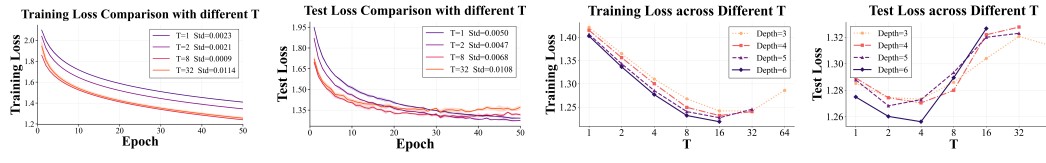

(a) Train loss with $L = 5$    (b) Test loss with $L = 5$    (c) Train loss (vary $L, T$)    (d) Test loss (vary $L, T$)

Figure 2: **Effect of the time horizon $T$ on ResNet performance.** We train width-128 ReLU ResNets on CIFAR-10 using SGD (LR 0.1, batch size 128) across 3 random seeds. (**a–b**) At fixed depth 5, moderate $T$ (e.g., 2) improves capacity and performance, whereas very large $T$ (e.g., 32) causes unstable training and degraded performance. (**c–d**) When varying both $T$ and depth $L$, moderate increases improve performance, while excessively large values of either induce instability, highlighting a trade-off between theoretical capacity gains and practical stability.

### 3.3 NETWORK CAPACITY IN THE KERNEL REGIME

We have shown that in the joint limit, the coordinates of $\boldsymbol{h}_\ell$ behave as independent particles governed by the limiting SDE, so the feature space geometry—and thus ResNet's capacity—is determined by the forward SDE's terminal state. This subsection analyzes this geometry in the *kernel regime* (at initialization); its evolution during training will be examined in Section 4 under the FL regime.

In the kernel regime ($\alpha = 1$), the following result shows the infinite-depth-width ResNet converges to a Gaussian process (GP) with a covariance determined by the SDE's terminal state. This neural network–Gaussian process (NNGP) correspondence, previously well established for finite-depth networks (Lee et al., 2018; Yang, 2019), now extends naturally to our infinite-depth setting.

**Proposition 3.** *Suppose $\alpha = 1$ and $\phi$ is $K_1$-Lipschitz continuous. Then, in the joint limit, the ResNet $f$ converges weakly to a Gaussian process with mean zero and covariance function $C_T(\boldsymbol{x}, \bar{\boldsymbol{x}}) = \frac{\langle \boldsymbol{x}, \bar{\boldsymbol{x}} \rangle}{d} + \int_0^T \mathbb{E}[\phi(h_s)\,\phi(\bar{h}_s)]\,ds$, where $h_s$ and $\bar{h}_s$ solve the limiting SDE defined in Eq. (2) initialized with $\boldsymbol{x}$ and $\bar{\boldsymbol{x}}$, respectively.*

Through the NNGP correspondence, we can characterize the ResNet's functional capacity in the kernel regime through a *reproducing kernel Hilbert space (RKHS)* $\mathcal{H}_t$ induced by the kernel $C_t$. The following result guarantees that $\mathcal{H}_t$ is *universal* under mild conditions for $\phi$.

**Proposition 4.** *Suppose $\phi$ is $K_1$-Lipschitz, nonlinear, and non-polynomial. Then for any $t > 0$, the kernel $C_t$ is strictly positive definite (SPD)[4] and the corresponding RKHS $\mathcal{H}_t$ is dense in $\mathcal{C}(\mathbb{S}^{d-1})$[5].*

Together with Proposition 2, the universality of the RKHS $\mathcal{H}_t$ implies that the function class realized by ResNets is universal in the joint limit, as characterized by their associated NNGP kernel $C_t$.

However, this perspective also reveals a fundamental *capacity ceiling*: scaling width or depth only reduces approximation error to the SDE limit without enlarging the functional class further. This explains the diminishing returns observed in practice—once the network closely approximates the limiting SDE, additional scaling provides little benefit as representational capacity is saturated.

At the same time, the SDE viewpoint suggests a natural way to *bypass* this ceiling: by modifying the parameters of the SDE itself. In particular, we show that increasing the time horizon $t$ provides a simple way to further expand the induced RKHS.

**Proposition 5.** *For any $0 < t \le t' < \infty$, the RKHS $\mathcal{H}_t$ is contractively contained in $\mathcal{H}_{t'}$.*

Thus, enlarging $T$ yields a more expressive feature space, which may improve performance. To test this, we vary $T$ and evaluate both training and test performance. As shown in Figure 2, moderate increases in $T$ indeed yield better performance, but excessively large values introduce training instability that outweighs the theoretical capacity gains.

---

[4] For any finite set of distinct points $\{\boldsymbol{x}_i\}_{i=1}^n$, the Gram matrix $\boldsymbol{A}_{ij} = C_t(\boldsymbol{x}_i, \boldsymbol{x}_j)$ is strictly positive definite.
[5] Here $\mathcal{C}(\mathbb{S}^{d-1})$ denotes the space of continuous real-valued functions on the unit sphere $\mathbb{S}^{d-1}$.

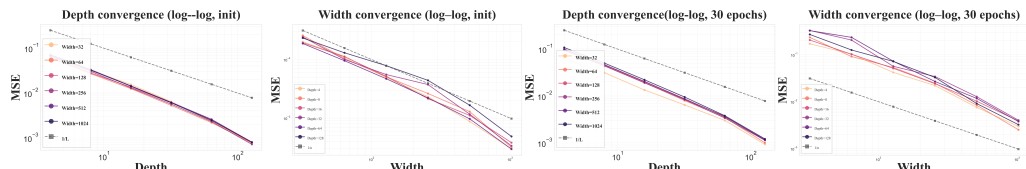

Figure 3: **Convergence to NFD at initialization and after 30 epochs.** We evaluate depth-$\mu$P ResNets on CIFAR-10 using SGD (LR 0.01, batch size 128). Widths range from 32 to 1024, and depths from 4 to 128. The approximation error decays as $\mathcal{O}(1/L)$ when increasing depth and as $\mathcal{O}(1/n)$ when increasing width. This uniform behavior confirms that the width and depth limits are empirically commutable both at initialization and after training.

## 4 FEATURE LEARNING DYNAMICS IN THE INFINITE-DEPTH LIMIT

Previously, we established an SDE view for forward feature propagation in ResNets at initialization under the joint limit. Here, we show that in the FL regime, this view extends to training, where backpropagation dynamically reshapes the forward SDE to align the feature space with data.

### 4.1 BACKWARD SDE INDUCED BY BACKPROPAGATION

During training, the gradients are computed through backpropagation. To analyze this process in the joint limit, our analysis focuses on the following backward information flow:

$$\boldsymbol{g}_L = \frac{\sqrt{n}}{\alpha}\frac{\partial f}{\partial \boldsymbol{h}_L} = \boldsymbol{v}, \quad \boldsymbol{g}_{\ell-1} = \frac{\sqrt{n}}{\alpha}\frac{\partial f}{\partial \boldsymbol{h}_{\ell-1}} = \boldsymbol{g}_\ell + \sqrt{\frac{T}{Ln}}\phi'(\boldsymbol{h}_{\ell-1}) \odot \boldsymbol{W}_\ell^\top \boldsymbol{g}_\ell, \quad \forall \ell \in [L], \quad (3)$$

where $\odot$ is element-wise multiplication. Notably, the coordinates of $\boldsymbol{g}_\ell$ remain $\Theta(1)$ w.r.t. width $n$, independent of the choice of $\alpha$. Under mild regularity assumptions on $\phi$ and $\phi'$, the limiting behaviors of $\boldsymbol{g}_\ell$ in the joint limit are described by a *backward* SDE[6].

**Proposition 6.** *Suppose $\phi$ and $\phi'$ are $K_1$- and $K_2$-Lipschitz continuous, respectively. Then, as $\min(n, L) \to \infty$, each coordinate of $\boldsymbol{g}_\ell$ converge to the solution of a McKean–Vlasov SDE*

$$dg_t = \tilde{\sigma}_t \, d\tilde{w}_t, \quad g_T \sim \mathcal{N}(0,1), \quad \forall t \in [0, T],$$

*with $\tilde{\sigma}_t^2 := |\phi'(h_t)|^2 \, \mathbb{E}[g_t^2]$, and $\{\tilde{w}_t\}$ as a standard Brownian motion **independent** of the forward Brownian motion $\{w_t\}$ in Eq. (2), with mean-square convergence at rate*

$$\mathbb{E}\left|\boldsymbol{g}_{\ell,i} - g_{t_\ell}\right|^2 \leq C\left(L^{-1} + n^{-1}\right), \quad \forall \ell \in [L],$$

*where the constant $C > 0$ does not depend on width $n$ or depth $L$.*

Similar to the feature propagation in Proposition 2, this convergence is also *commutable*. Moreover, the forward and backward SDEs are driven by ***independent*** Brownian motions, $\{\boldsymbol{w}_t\}$ and $\{\tilde{\boldsymbol{w}}_t\}$, respectively. This independence corresponds to treating the forward weights $\boldsymbol{W}_\ell$ in Eq. (1) and the backward weights $\boldsymbol{W}_\ell^\top$ in Eq. (3) as independent. This is a widely used assumption, known as the *gradient independence assumption* (GIA), in prior works on signal propagation in deep networks in the lazy regime (Schoenholz et al., 2017; Lee et al., 2018; Jacot et al., 2018a).

The GIA is rigorously justified at initialization in the infinite-width limit for finite-depth networks (Yang, 2020b), but it was later shown to ***break down during training*** due to correlations that accumulate between the forward and backward paths (Yang & Hu, 2021). Remarkably, we will show in Section 4.2 that in the ***infinite-depth limit under Depth-$\mu$P*** these correlations vanish, so ***GIA becomes provably valid again throughout training***.

### 4.2 NEURAL FEATURE DYNAMICS

We now investigate how features and gradients evolve during training in the joint limit. We demonstrate that the SDE view derived in Section 3 continues to hold throughout training. Particularly, the

---

[6]Here the term means that the SDE starts from time $T$ and goes backward to time 0, c.f. Remark 2. This term, and the term of forward-backward SDE in Definition 1, although used throughout the paper, should not to be confused with the classic terminology BSDE and FBSDE in probability theory.

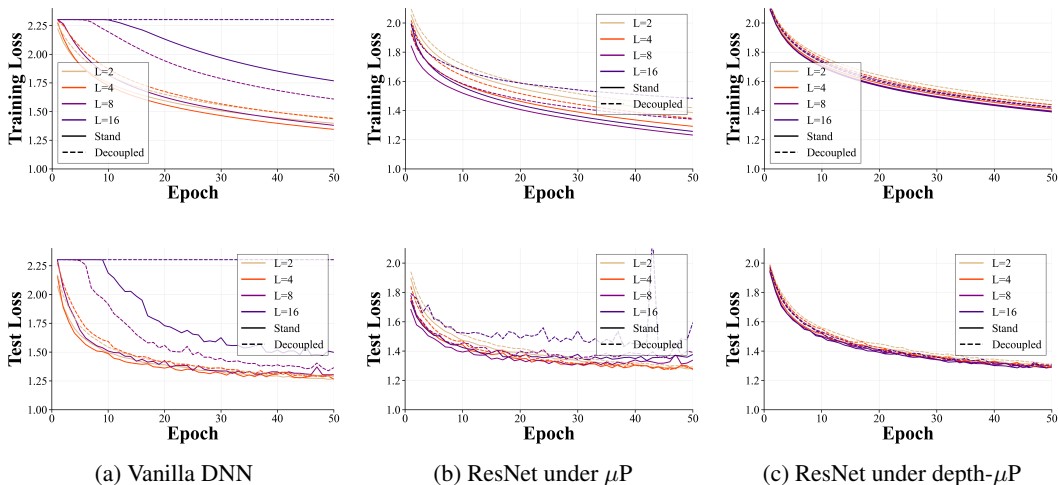

(a) Vanilla DNN  (b) ResNet under $\mu$P  (c) ResNet under depth-$\mu$P

Figure 4: **Empirical evaluation of GIA restoration.** We train all models with width 128 on CIFAR-10 using SGD (LR 0.1 and batch size 128), comparing *standard* training (shared forward/backward weights) with a *decoupled* setup where the backward pass uses an i.i.d. copy of the forward weights. As depth increases: **(a)** Vanilla DNNs suffer from vanishing gradients and make little progress, and the two trajectories remain misaligned. **(b)** ResNets under $\mu$P also show no alignment and eventually overfit at large depth (e.g., 16). **(c)** Depth-$\mu$P improves both training and test performance with depth, and standard and decoupled trajectories converge to one another, demonstrating empirical restoration of GIA. Appendix I shows that larger widths reduce instability in (a,b) but do not eliminate their pathologies, while further reinforcing the alignment and performance gains of depth-$\mu$P.

backward SDE from Section 4.1, induced by backpropagation, dynamically modifies the forward SDE, yielding a coupled stochastic system that we refer to as the *Neural Feature Dynamics (NFD)*.

To ensure that the network operates in the FL regime (rather than NTK), we adopt the $\mu$P scaling from the TP framework by setting $\alpha = 1/\sqrt{n}$ and using a learning rate $\eta = \eta_c n$, where $\eta_c > 0$ is an absolute constant. Under this setup, the hidden features $\boldsymbol{h}_\ell^{(k)}$ at the $k$-th SGD iteration evolve as:

$$\boldsymbol{h}_\ell^{(k)} = \boldsymbol{h}_{\ell-1}^{(k)} - \eta_c \sum_{i=0}^{k-1} \frac{T}{L} \mathcal{L}'(f^{(i)}, y^{(i)}) \frac{\left\langle \phi(\boldsymbol{h}_{\ell-1}^{(i)}), \phi(\boldsymbol{h}_{\ell-1}^{(k)}) \right\rangle}{n} \boldsymbol{g}_{\ell-1}^{(i)} + \sqrt{\frac{T}{Ln}} \boldsymbol{W}_\ell \phi(\boldsymbol{h}_{\ell-1}^{(k)}), \quad (4)$$

where $\boldsymbol{g}_\ell^{(i)}$ is defined in Eq. (3) and see Appendix E for detailed derivation for $\boldsymbol{h}_\ell^{(k)}$ and $\boldsymbol{g}_\ell^{(k)}$.

From Eq. (4), it can be inductively verified that all $\{\boldsymbol{h}_\ell^{(k)}, \boldsymbol{g}_\ell^{(k)}\}$ are valid TP variables. This ensures that the entire training trajectory under SGD remains analytically tractable in the infinite-width limit.

**Proposition 7.** *Training a randomly initialized ResNet defined in Eq. (1) under SGD is a valid TP.*

This enables the application of the Master Theorem (Yang & Hu, 2021, Theorem 7.4) to study the training trajectory in the mean-field limit as the width tends to infinity.

**Proposition 8.** *Suppose $\mathcal{L}'$, $\phi$, and $\phi'$ are pseudo-Lipschitz continuous. Then, as $n \to \infty$, the output $f^{(k)}$ converges a.s. to $\mathring{f}^{(k)} = \mathbb{E}[Z^{\boldsymbol{g}_L^{(k)}} Z^{\boldsymbol{h}_L^{(k)}}]$, where the hidden states evolve recursively as:*

$$Z^{\boldsymbol{h}_\ell^{(k)}} = Z^{\boldsymbol{h}_{\ell-1}^{(k)}} + \sqrt{\tau} Z^{\boldsymbol{W}_\ell \phi_{\ell-1}^{(k)}} - \tau \sum_{i=0}^{k-1} \eta_c \mathcal{L}'(\mathring{f}^{(i)}, y^{(i)}) \mathbb{E}[\phi(Z^{\boldsymbol{h}_{\ell-1}^{(i)}}) \phi(Z^{\boldsymbol{h}_{\ell-1}^{(k)}})] Z^{\boldsymbol{g}_\ell^{(i)}}$$

$$- \tau^2 \sum_{i=0}^{k-1} \eta_c \mathcal{L}'(\mathring{f}^{(i)}, y^{(i)}) \mathbb{E}[\phi(Z^{\boldsymbol{h}_{\ell-2}^{(i)}}) \phi(Z^{\boldsymbol{h}_{\ell-2}^{(k)}})] \mathbb{E}[\phi'(Z^{\boldsymbol{h}_{\ell-1}^{(i)}}) \phi'(Z^{\boldsymbol{h}_{\ell-1}^{(k)}})] Z^{\boldsymbol{g}_\ell^{(i)}}, \quad (5)$$

where $\tau = T/L$ and $\{Z^{\boldsymbol{W}_\ell \phi_{\ell-1}^{(i)}}\}_{\ell,i}$ are a mutually independent set of centered joint Gaussian defined based on (Yang & Hu, 2021, Definition 7.3).

The final term in Eq. (5) reflects an additional correlation by reusing the same weights $\boldsymbol{W}_\ell$ during backpropagation. This correlation is the precise reason why **GIA fails during training** in finite-depth networks: the forward and backward paths become statistically coupled, making feature-learning dynamics analytically intractable. Remarkably, stochastic calculus shows that this correlation term is of order $\tau^2$ and hence vanishes under the Euler–Maruyama scheme as $\tau \to 0$ (equivalently, $L \to \infty$). Thus, **in the infinite-depth limit, GIA becomes asymptotically valid: $\boldsymbol{W}_\ell^\top$ in the backward pass can be treated as independent from forward weight $\boldsymbol{W}_\ell$ in the forward pass.** This **restoration** of independence is a central and previously unexplored finding: it creates the first setting where FL dynamics remain tractable throughout training, beyond initialization. A detailed analysis of this interaction and its vanishing is provided in (Yang & Hu, 2021, Section 6) and Appendix F and G.

Hence, in the FL regime, we expect the behavior of large-depth ResNets to be captured by a coupled stochastic system driven by independent Brownian motions, *i.e.*, NFD. This system describes the co-evolution of forward features and backward gradients. Below, we formally define NFD and show that, under mild regularity assumptions, finite ResNets converge to this limiting dynamics.

**Definition 1** (Neural Feature Dynamics (NFD)). *The NFD for ResNets is a coupled forward-backward SDE system driven by* **independent** *Brownian motions that describes the evolution of features and gradients over training iteration $k$ in the joint infinite-width-depth limit:*

$$h_0^{(k)} = \hat{h}_0^{(k)} - \eta_c \sum_{i=0}^{k-1} \mathcal{L}'(\mathring{f}^{(i)}, y^{(i)}) \frac{\langle \boldsymbol{x}^{(i)}, \boldsymbol{x}^{(k)} \rangle}{d} g_0^{(i)},$$

$$dh_t^{(k)} = -\sum_{i=0}^{k-1} \eta_c \mathcal{L}'(\mathring{f}^{(i)}, y^{(i)}) \mathbb{E}[\phi(h_t^{(i)})\phi(h_t^{(k)})] g_t^{(i)} \, dt + d\boldsymbol{w}_{t,k}^{(k)},$$

$$g_T^{(k)} = \hat{g}_T^{(k)} - \eta_c \sum_{i=0}^{k-1} \mathcal{L}'(\mathring{f}^{(i)}, y^{(i)}) h_T^{(i)},$$

$$dg_t^{(k)} = -\sum_{i=0}^{k-1} \eta_c \mathcal{L}'(\mathring{f}^{(i)}, y^{(i)}) \mathbb{E}[g_t^{(i)} g_t^{(k)}] \phi(h_t^{(i)})\phi'(h_t^{(k)}) \, dt + \phi'(h_t^{(k)}) \, d\tilde{\boldsymbol{w}}_{t,k}^{(k)},$$

*where $\eta_c > 0$ is the fixed effective learning rate and $\{\hat{h}_0^{(i)}\}$ and $\{\hat{g}_T^{(i)}\}$ are centered Gaussians with*

$$\mathrm{Cov}(\hat{h}_0^{(i)}, \hat{h}_0^{(j)}) = \boldsymbol{x}^{(i)\top} \boldsymbol{x}^{(j)}/d, \quad \mathrm{Cov}(\hat{g}_T^{(i)}, \hat{g}_T^{(j)}) = 1.$$

*The (k+1)-dimensional Brownian motions $\{\boldsymbol{w}_t^{(k)}\}$ and $\{\tilde{\boldsymbol{w}}_t^{(k)}\}$ have time-varying covariances:*

$$\frac{d}{dt}\mathbb{E}[\boldsymbol{w}_{t,i}^{(k)} \boldsymbol{w}_{t,j}^{(k)}] = \boldsymbol{\Sigma}_{t,ij}^{(k)} = \mathbb{E}[\phi(h_t^{(i)})\phi(h_t^{(j)})], \quad \frac{d}{dt}\mathbb{E}[\tilde{\boldsymbol{w}}_{t,i}^{(k)} \tilde{\boldsymbol{w}}_{t,j}^{(k)}] = \boldsymbol{\Theta}_{t,ij}^{(k)} = \mathbb{E}[g_t^{(i)} g_t^{(j)}].$$

**Assumption 1.** *We assume:*

1. *$\mathcal{L}'$, $\phi$, and $\phi'$ are Lipschitz continuous.*

2. *There exists a solution to the NFD system such that $\{\boldsymbol{\Sigma}_t^{(k)}, \boldsymbol{\Theta}_t^{(k)}\}_{t \in [0,T]}$ are uniformly SPD, that is, for each $k$, $\min_{t \in [0,T]} \left\{ \lambda_{\min}(\boldsymbol{\Sigma}_t^{(k)}), \lambda_{\min}(\Theta_t^{(k)}) \right\} \geq \varepsilon$ for some constant $\varepsilon > 0$.*

**Theorem 1.** *Suppose $\alpha = 1/\sqrt{n}$, $\eta = \eta_c n$, and Assumption 1 holds. As $n, L \to \infty$ sequentially, the ResNet output $f^{(k)}$ converges a.s. to $\mathring{f}^{(k)} = \mathbb{E}[g_T^{(k)} h_T^{(k)}]$, where $h_t^{(k)}$ and $g_t^{(k)}$ evolve under the NFD in Definition 1.*

The NFD formalism in Definition 1 captures the full co-evolution of forward features and backward gradients through a coupled forward–backward SDE. Both the drift terms and the driving Brownian motions encode accumulated training effects. Notably, the Brownian motions in NFD are not fixed-dimensional: their dimensions grow with the number of training iterations, reflecting the continual expansion of learned representations. Most importantly, the forward and backward SDEs are driven by independent Brownian motions without additional correlation terms, revealing that the statistical coupling responsible for the failure of GIA at finite depth disappears in the large-depth limit.

**Corollary 1** (Restoration of GIA in the infinite-depth limit). *Under the assumptions of Theorem 1, consider a ResNet $\widetilde{f}$ under depth-$\mu$P defined in Eq. (1) in which backpropagation is decoupled by*

*replacing each $\boldsymbol{W}_\ell^\top$ with an independent copy $\widetilde{\boldsymbol{W}}_\ell$. As $n, L \to \infty$ sequentially, the resulting output $\widetilde{f}^{(k)}$ converges to the same limit $\mathring{f}^{(k)} = \mathbb{E}[g_T^{(k)} h_T^{(k)}]$ as the original ResNet $f^{(k)}$.*

A relevant analysis is given in (Bordelon et al., 2024c), which also derives a training-time SDE using the dynamical mean-field theory (DMFT). However, the DMFT used there is a physics-style heuristic method, rather than a mathematically rigorous framework. Consequently, their derivation does not obtain the restoration of GIA. Instead, their formulation explicit ***response functions*** to account for the forward–backward correlations. In contrast, our Theorem 1 and Corollary 1 show that these correlations vanish in the infinite-depth limit. We further conduct an empirical evaluation of this restoration, and the results (Figure 4) confirm that standard and decoupled dynamics align only under depth-$\mu$P. This restoration of independence is what makes NFD the ***first mathematically rigorous setting*** in which FL dynamics remain tractable beyond initialization and kernel regime, and it further clarifies why HP transfer deteriorates with depth and how it can be systematically corrected; see Section 5 for details.

Theorem 1 establishes convergence of ResNet outputs to the NFD limit under the width-then-depth scaling order. Consistent with Propositions 2 and 6 at initialization, our experiments further demonstrate that this convergence is in fact *commutative*; see Figure 3. We also verify Assumption 1 by showing that the smallest eigenvalues of the covariance matrices $\boldsymbol{\Sigma}_t^{(k)}$ and $\Theta_t^{(k)}$ remain strictly positive throughout training for sufficiently wide networks (Figure 6). Together, these results provide both theoretical and empirical support for NFD as the first principled and well-posed description of feature learning in the joint infinite-width and infinite-depth limit.

## 5 Vanishing Feature Learning in Multi-Layer Residual Blocks

Many practical architectures, including Transformers, use multi-layer residual blocks. Although depth-$\mu$P supports HP transfer for one-layer blocks (Yang et al., 2024; Bordelon et al., 2024c), multiple studies have observed that this behavior breaks for two-layer blocks (Yang et al., 2024; Bordelon et al., 2024b; Dey et al., 2025). The underlying cause, however, has remained underexplored. Here, we explain this failure through the lens of NFD.

We analyze a two-layer residual block of the form:

$$\boldsymbol{h}_\ell = \boldsymbol{h}_{\ell-1} + \frac{1}{\sqrt{Ln}}\boldsymbol{W}_{\ell,2}\phi(\boldsymbol{x}_\ell), \quad \boldsymbol{x}_\ell = \frac{1}{\sqrt{n}}\boldsymbol{W}_{\ell,1}\boldsymbol{h}_{\ell-1}, \quad \forall\ell \in \{1, 2, \cdots, L\}, \tag{6}$$

where $\boldsymbol{x}_\ell$ is the preactivation after the first internal layer. From the perspective of the NFD, the learning roles of these two layers are fundamentally different:

- The first layer $\boldsymbol{W}_{\ell,1}$ determines the **internal representation** $\boldsymbol{x}_{t_\ell}$;
- The second layer $\boldsymbol{W}_{\ell,2}$ governs the **residual-stream dynamics** $\partial_t \boldsymbol{h}_{t_\ell} \approx (\boldsymbol{h}_\ell - \boldsymbol{h}_{\ell-1})/\tau$.

This ***structural separation*** is **not** visible from weight-space analysis but emerges naturally in the NFD limit. This viewpoint implies that the meaningful quantity for analyzing learning in the large-depth limit is ***not*** the size of the weight update itself, but the induced change in $\boldsymbol{x}_{t_\ell}$ and $\partial_t \boldsymbol{h}_{t_\ell}$. Crucially, even if the raw weight updates shrink with depth in both layers, their effects on $\boldsymbol{x}_{t_\ell}$ and $\partial_t \boldsymbol{h}_{t_\ell}$ can behave very differently.

Indeed, under standard depth-$\mu$P with learning rate $\eta = \eta_c n$, we observe the following:

$$\|\boldsymbol{x}_\ell^{(k)} - \widetilde{\boldsymbol{x}}_\ell^{(k)}\|/\sqrt{n} \sim \mathcal{O}\left(1/\sqrt{L}\right), \quad \|\partial\boldsymbol{h}_{t_\ell}^{(k)} - \partial\widetilde{\boldsymbol{h}}_{t_\ell}^{(k)}\|/\sqrt{n} \sim \mathcal{O}(1), \tag{7}$$

where $\widetilde{\boldsymbol{x}}_\ell^{(k)}$ and $\partial\widetilde{\boldsymbol{h}}_{t_\ell}^{(k)}$ are ***frozen-weight*** baselines defined in Appendix H. Consequently, as depth grows, feature learning in the **first** internal layer collapses, $\boldsymbol{x}_\ell$ receives almost no learnable update, while the **second** layer, responsible for residual-stream dynamics update, remains fully active. This asymmetric collapse explains why depth-wise HP transfer breaks for multi-layer blocks.

Guided by this NFD-based insight, we introduce a simple **depth-aware learning-rate correction**:

$$\textbf{First layer:} \quad \eta_1 = \eta_c\, n\sqrt{L}, \quad \textbf{Second layer:} \quad \eta_2 = \eta_c\, n. \tag{8}$$

This removes the $1/\sqrt{L}$ suppression on the first layer, restoring meaningful updates to $\boldsymbol{x}_\ell$ while preserving the correct scaling for residual-stream dynamics. As a result, depth-wise HP transfer is re-established in two-layer residual blocks. A numerical example[7] is shown in Figure 5.

---

[7]Train loss is epoch-averaged; test loss is end-of-epoch, hence often lower.

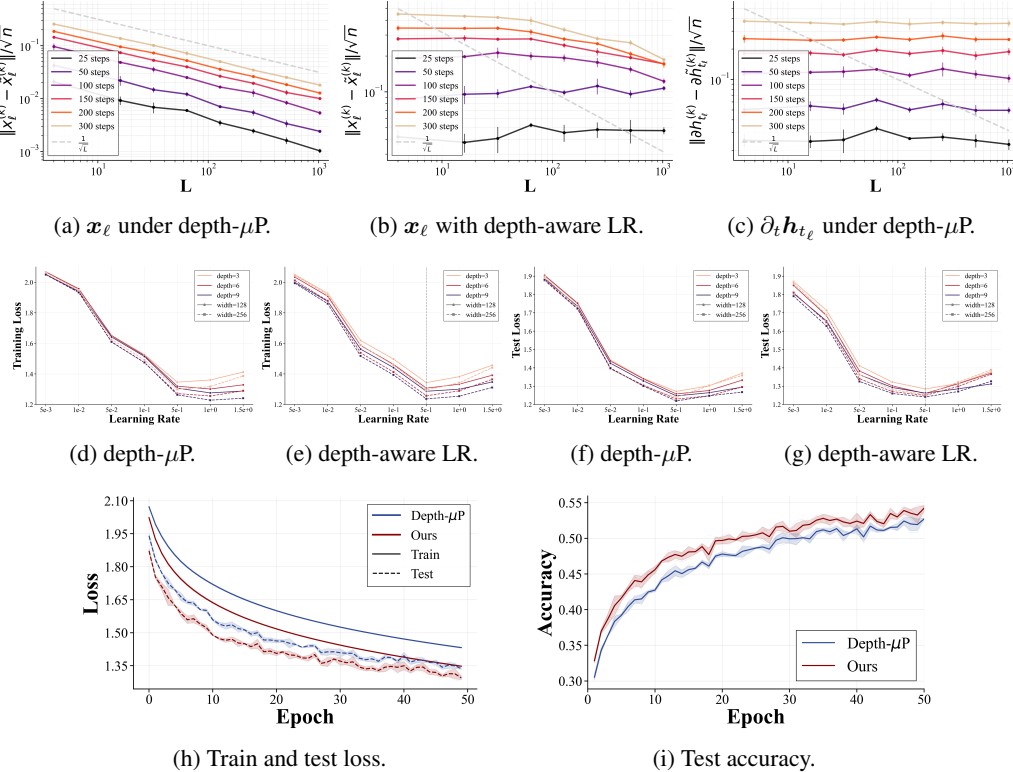

Figure 5: **Vanishing feature learning and recovery in two-layer residual blocks.** In **Row 1**, we train a width-128 ResNet with ReLU on CIFAR-10 using SGD (batch size 128, $\eta_c = 0.1$) for 300 steps, across depths $L$. Under standard depth-$\mu$P, the feature-update error collapses at rate $1/\sqrt{L}$, indicating that the first layer stops learning as depth grows. Our depth-aware learning rate $\eta_1 = \eta_c n\sqrt{L}$ restores active feature learning, eliminating the vanishing behavior. In **Row 2**, we evaluate depth-wise HP transfer on CIFAR-10 by training ResNets with two-layer blocks at different widths and depths across choices of $\eta_c$. Vanilla depth-$\mu$P displays inconsistent optima across depths, yielding poor HP transfer. With the proposed depth-aware LR in Eq. (8), both train and test loss align across depths and uniformly improve, demonstrating restored depth-wise HP transfer. For example, in panel (d), the optimal $\eta_c$ for depths 6 and 9 occurs near $\eta_c = 1$, whereas depth 3 attains its optimum around $\eta_c = 0.5$. Moreover, the corresponding test curves from (g) show that depths 6 and 9 already overfit at $\eta_c = 1$. In contrast, the depth-aware LR consistently achieves its optimal performance at $\eta_c = 0.5$ across both train and test loss. In **Row 3**, on a depth-64 width-128 ResNet, our correction achieves consistently lower train and test losses, and higher test accuracy than vanilla depth-$\mu$P, demonstrating stronger overall performance.

# 6    CONCLUSIONS

We introduced the first mathematically rigorous framework—Neural Feature Dynamics (NFD)—for analyzing feature learning in deep ResNets. Under depth-$\mu$P, we proved that the evolution of features and gradients converges to a coupled stochastic system driven by *independent* Brownian motions, thereby restoring the gradient-independence assumption (GIA) in the infinite-depth limit and yielding a tractable characterization of feature evolution throughout training. Building on this framework, NFD revealed a depth-induced vanishing effect that suppresses first-layer learning in two-layer residual blocks and explains the longstanding failure of HP transfer across depths. Guided by this insight, we proposed a simple depth-aware learning-rate correction that restores feature learning and HP transfer, yielding consistent performance improvements across depths; experiments on CIFAR-10 confirm the theory's predictions. Together, these results provide a principled foundation for depth-aware scaling in modern deep networks and open avenues for extending NFD to broader architectures and optimizers.

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

## A   USEFUL MATHEMATICAL RESULTS

**Lemma 1** (Gronwall's inequality). *Let $I = [a, b]$ for an interval such that $a < b < \infty$. Let $u$, $\alpha$, $\beta$ be real-valued continuous functions such that $\beta$ is non-negative and $u$ satisfies the integral inequality*

$$u(t) \leq \alpha(t) + \int_0^t \beta(s)u(s)ds, \quad \forall t \in I.$$

*Then*

$$u(t) \leq \alpha(t) + \int_0^t \alpha(s)\beta(s)\exp\left(\int_s^t \beta(r)dr\right), \quad \forall t \in I.$$

*If, in addition, $\alpha(t)$ is non-decreasing, then*

$$u(t) \leq \alpha(t)\exp\left(\int_0^t \beta(s)ds\right), \quad \forall t \in I.$$

**Lemma 2** (Gronwall's inequality (discrete version)). *Let $(u_n)$ and $(\beta_n)$ be non-negative sequences satisfying*

$$u_n \leq \alpha + \sum_{k=0}^{n-1} \beta_k u_k, \quad \forall n,$$

*where $\alpha \geq 0$. Then*

$$u_n \leq \alpha \exp\left(\sum_{k=0}^{n-1} \beta_k\right), \quad \forall n.$$

## B   POST-ACTIVATION VS. PRE-ACTIVATION RESNET DESIGN

The ResNet architecture defined in Eq. (2) follows the so-called *pre-activation style* (He et al., 2016b), where the residual (skip) connection is applied to the *pre-activation* feature vectors. In this convention, the hidden feature vectors $\boldsymbol{h}_\ell$ are the raw, pre-activation representations. There is, however, another widely used variant: the *post-activation style* (He et al., 2016a), in which the residual connection is applied after the nonlinearity. The corresponding recursion is

$$\boldsymbol{h}_0 = \phi(\frac{1}{\sqrt{d}}\boldsymbol{U}\boldsymbol{x}), \quad \boldsymbol{h}_\ell = \boldsymbol{h}_{\ell-1} + \sqrt{\frac{T}{L}}\phi(\frac{1}{\sqrt{n}}\boldsymbol{W}_\ell\boldsymbol{h}_{\ell-1}), \quad \forall \ell \in \{1, 2, \ldots, L\}. \tag{9}$$

Historically, the post-activation formulation appeared first in the original ResNet paper (He et al., 2016a), but subsequent large-scale architectures, including modern Transformers (Vaswani, 2017), overwhelmingly favor the pre-activation design. This shift was motivated primarily by empirical practice: pre-activation models exhibit improved trainability and stability when scaled to hundreds or thousands of layers. However, the theoretical reasons for this preference are less often articulated.

In what follows, we provide a theoretical perspective. We show that pre-activation is essentially more stable as the depth $L \to \infty$. By contrast, in the post-activation formulation Eq. (9), it is easy to find commonly used activation functions (e.g., ReLU) for which the hidden features $\boldsymbol{h}_\ell$ become unstable in this infinite-depth regime. This analysis gives formal justification for the empirical consensus that pre-activation ResNets are better suited for ultra-deep architectures.

*Proof of Proposition 1.* Recall that $\phi$ is assumed to be positive dominate, that is, there exist non-negative constants $c_1, c_2$, not both zero, such that $\mathbb{E}[\phi(xZ)] \geq c_1|x| + c_2, \forall x \in \mathbb{R}$, where $Z$ is the standard Gaussian random variable.

First note that $(\boldsymbol{U}\boldsymbol{x})_i \sim \mathcal{N}(0, \|\boldsymbol{x}\|^2)$ and hence $\mathbb{E}[\boldsymbol{h}_{0,i}] \geq c_1\frac{\|\boldsymbol{x}\|}{\sqrt{d}} + c_2$. Let $\mathcal{B}_\ell$ be the $\sigma$-algebra generated by $\{\boldsymbol{h}_0, \cdots, \boldsymbol{h}_{\ell-1}\}$. Then, observe that

$$\mathbb{E}[\boldsymbol{h}_{\ell,i} \mid \mathcal{B}_\ell] = \boldsymbol{h}_{\ell-1,i} + \sqrt{\frac{T}{L}}\mathbb{E}\left[\phi\left(\frac{\|\boldsymbol{h}_{\ell-1}\|}{\sqrt{n}}Z\right) \mid \mathcal{B}_\ell\right] \geq \boldsymbol{h}_{\ell-1,i} + \sqrt{\frac{T}{L}}\left(c_1\frac{\|\boldsymbol{h}_{\ell-1}\|}{\sqrt{n}} + c_2\right)$$

$$\geq \boldsymbol{h}_{\ell-1,i} + \sqrt{\frac{T}{L}}\left(c_1\frac{\boldsymbol{h}_{\ell-1,i}}{\sqrt{n}} + c_2\right) = \left(1 + c_1\sqrt{\frac{T}{Ln}}\right)\boldsymbol{h}_{\ell-1,i} + c_2\sqrt{\frac{T}{L}}.$$

Then taking expectation of $\mathcal{B}_\ell$ yields

$$\mathbb{E}[\boldsymbol{h}_{\ell,i}] \geq \left(1 + c_1\sqrt{\frac{T}{Ln}}\right)\mathbb{E}[\boldsymbol{h}_{\ell-1,i}] + c_2\sqrt{\frac{T}{L}}.$$

Therefore, we obtain

$$\mathbb{E}[\boldsymbol{h}_{\ell,i}] \geq \left(1 + c_1\sqrt{\frac{T}{Ln}}\right)^\ell \mathbb{E}[\boldsymbol{h}_{0,i}] + c_2\sqrt{\frac{T}{L}}\sum_{j=0}^{\ell-1}\left(1 + c_1\sqrt{\frac{T}{Ln}}\right)^j$$

$$\geq \left(1 + c_1\sqrt{\frac{T}{Ln}}\right)^\ell\left(c_1\frac{\|\boldsymbol{x}\|}{\sqrt{d}} + c_2\right) + c_2\sqrt{\frac{T}{L}}\ell.$$

Therefore we obtain

$$\mathbb{E}[\boldsymbol{h}_{L,i}] \geq c_1\frac{\|\boldsymbol{x}\|}{\sqrt{d}}\left(1 + c_1\sqrt{\frac{T}{Ln}}\right)^L + c_2\sqrt{TL} \to \infty$$

as $L \to \infty$, provided either $c_1 > 0$ or $c_2 > 0$. $\qquad\square$

## C  PROOFS OF PROPOSITIONS 2 AND 6

In this section, we focus our convergence analysis of feature and gradient propagation at initialization, considering both the first forward feature propagation and the first backward gradient propagation. The overall approach first takes the width $n \to \infty$, showing that the coordinates of the feature and gradient vectors become asymptotically independent and are governed by their respective mean-field recursions. This argument can be made precise using classical propagation of chaos techniques, in particular synchronous coupling together with moment bounds and discrete Gronwall's inequality. This establishes simplified finite-depth recursions. We then let the depth $L \to \infty$, interpret the Gaussian increments as Brownian motion steps, and recognize the resulting dynamics as Euler–Maruyama discretizations of McKean–Vlasov SDEs. By combining moment bounds with the Lipschitz continuity of the variance functional, we obtain explicit convergence rates along with the proof of existence and uniqueness of the limiting forward and backward SDEs stated in Propositions 2 and 6.

### C.1  FIRST FORWARD — PROOF OF PROPOSITION 2

Suppose $\phi$ satisfies the assumption in Proposition 2 throughout. Consider

$$\boldsymbol{h}_\ell = \boldsymbol{h}_{\ell-1} + \sqrt{\frac{T}{Ln}}\boldsymbol{W}_\ell\phi(\boldsymbol{h}_{\ell-1})$$

which is equivalent to

$$\boldsymbol{h}_\ell = \boldsymbol{h}_{\ell-1} + \sqrt{\frac{T}{Ln}}\|\phi(\boldsymbol{h}_\ell)\|\boldsymbol{z}_\ell, \quad \boldsymbol{z}_\ell \sim \mathcal{N}(0, \boldsymbol{I}_n).$$

To distinguish from quantities after taking limits of $n \to \infty$ and $L \to \infty$, we add superscripts and write each coordinate as

$$h_{\ell,i}^{n,L} = h_{\ell-1,i}^{n,L} + \sqrt{\frac{T}{Ln}}\|\phi(\boldsymbol{h}_{\ell-1}^{n,L})\|z_{\ell,i}.$$

We want to show that each coordinate converges to

$$h_{\ell,i}^L = h_{\ell-1,i}^L + \sqrt{\frac{T}{L}}\sqrt{\mathbb{E}\phi^2(h_{\ell-1,i}^L)}z_{\ell,i}$$

as $n \to \infty$. We will use $C$ to denote positive constants that depend only on $T$, $K_1$, and $\phi(0)$. Its value may change from line to line.

**Lemma 3.** *For each $i \in \mathbb{N}$, $\sup\limits_{L \geq 1} \sup\limits_{\ell=0,\ldots,L} \mathbb{E}[h_{\ell,i}^L]^4 < \infty$ and $\inf\limits_{L \geq 1} \inf\limits_{\ell=0,\ldots,L} \mathbb{E}\phi^2(h_{\ell,i}^L) > 0$.*

*Proof of Lemma 3.* By symmetry, we only have to consider a fixed $i \in \mathbb{N}$. Using independence of $z_{\ell,i}$, we have

$$\mathbb{E}[h_{\ell,i}^L]^4 \leq C\mathbb{E}[h_{0,i}^L]^4 + C\mathbb{E}\left[\sum_{u=1}^{\ell} \sqrt{\frac{T}{L}} \sqrt{\mathbb{E}\phi^2(h_{u-1,i}^L)} z_{u,i}\right]^4$$

$$\leq C + \frac{C}{L}\sum_{u=1}^{\ell} \mathbb{E}\phi^4(h_{u-1,i}^L) \leq C + \frac{C}{L}\sum_{u=0}^{\ell-1} \mathbb{E}[h_{u,i}^L]^4.$$

It then follows from discrete Gronwall's inequality (Lemma 2) that

$$\mathbb{E}[h_{\ell,i}^L]^4 \leq Ce^{C\ell/L}. \tag{10}$$

This gives the first assertion.

For the second assertion, note that $\phi$ is a continuous function and not identically zero. So there exists some interval $(a, b) \subset \mathbb{R}$ such that $\inf_{a < x < b} \phi^2(x) > 0$. Since $h_{\ell-1,i}^L$ and $z_{\ell,i}$ are independent, we have

$$\text{Var}(h_{\ell,i}^L) \geq \text{Var}(h_{\ell-1,i}^L) \geq \cdots \geq \text{Var}(h_{0,i}^L) = C_1 > 0.$$

Also

$$\text{Var}(h_{\ell,i}^L) \leq \mathbb{E}[h_{\ell,i}^L]^2 \leq C_2.$$

So $\{h_{\ell,i}^L\}$ are Gaussian random variables with mean zero and variance in $[C_1, C_2]$. Therefore,

$$\inf_{L \geq 1} \inf_{\ell=0,\ldots,L} \mathbb{P}(h_{\ell,i}^L \in (a, b)) > 0.$$

This gives the second assertion. $\qquad\square$

**Proposition 9.** *For each $i \in \mathbb{N}$,*

$$\sup_{L \geq 1} \sup_{\ell=0,\ldots,L} \mathbb{E}(h_{\ell,i}^{n,L} - h_{\ell,i}^L)^2 \leq C/n.$$

*Proof of Proposition 9.* Since $h_{0,i}^{n,L} = h_{0,i}^L$, we have

$$\mathbb{E}(h_{\ell,i}^{n,L} - h_{\ell,i}^L)^2 = \mathbb{E}\left[\sum_{u=1}^{\ell}\left(\frac{\|\phi(h_{u-1}^{n,L})\|}{\sqrt{n}} - \sqrt{\mathbb{E}\phi^2(h_{u-1,i}^L)}\right)\sqrt{\frac{T}{L}}z_{u,i}\right]^2$$

$$= \frac{C}{L}\mathbb{E}\sum_{u=1}^{\ell}\left(\frac{\|\phi(h_{u-1}^{n,L})\|}{\sqrt{n}} - \sqrt{\mathbb{E}\phi^2(h_{u-1,i}^L)}\right)^2,$$

where the second line uses the fact that $\{z_{\ell,i}\}_\ell$ are independent standard normal random variables. By adding and subtracting terms, we have

$$\mathbb{E}\left(\frac{\|\phi(h_{u-1}^{n,L})\|}{\sqrt{n}} - \sqrt{\mathbb{E}\phi^2(h_{u-1,i}^L)}\right)^2 \leq 2\mathbb{E}\left(\sqrt{\frac{1}{n}\sum_{j=1}^{n}\phi^2(h_{u-1,j}^{n,L})} - \sqrt{\frac{1}{n}\sum_{j=1}^{n}\phi^2(h_{u-1,j}^L)}\right)^2$$

$$+ 2\mathbb{E}\left(\sqrt{\frac{1}{n}\sum_{j=1}^{n}\phi^2(h_{u-1,j}^L)} - \sqrt{\mathbb{E}\phi^2(h_{u-1,i}^L)}\right)^2. \tag{11}$$

For the first term on the right hand side, using Minkowski's inequality, we have

$$\mathbb{E}\left(\sqrt{\frac{1}{n}\sum_{j=1}^{n}\phi^2(h_{u-1,j}^{n,L})} - \sqrt{\frac{1}{n}\sum_{j=1}^{n}\phi^2(h_{u-1,j}^{L})}\right)^2$$

$$\leq \mathbb{E}\left(\sqrt{\frac{1}{n}\sum_{j=1}^{n}\left(\phi(h_{u-1,j}^{n,L}) - \phi(h_{u-1,j}^{L})\right)^2}\right)^2$$

$$\leq \frac{K_1^2}{n}\sum_{j=1}^{n}\mathbb{E}(h_{u-1,j}^{n,L} - h_{u-1,j}^{L})^2 = K_1^2\mathbb{E}(h_{u-1,i}^{n,L} - h_{u-1,i}^{L})^2.$$

For the second term on the right hand side of Eq. (11), we have

$$\mathbb{E}\left(\sqrt{\frac{1}{n}\sum_{j=1}^{n}\phi^2(h_{u-1,j}^{L})} - \sqrt{\mathbb{E}\phi^2(h_{u-1,i}^{L})}\right)^2$$

$$= \mathbb{E}\left(\frac{\frac{1}{n}\sum_{j=1}^{n}\phi^2(h_{u-1,j}^{L}) - \frac{1}{n}\sum_{j=1}^{n}\mathbb{E}\phi^2(h_{u-1,j}^{L})}{\sqrt{\frac{1}{n}\sum_{j=1}^{n}\phi^2(h_{u-1,j}^{L})} + \sqrt{\mathbb{E}\phi^2(h_{u-1,i}^{L})}}\right)^2$$

$$\leq C\frac{1}{n^2}\sum_{j=1}^{n}\mathbb{E}\left[\phi^2(h_{u-1,j}^{L}) - \mathbb{E}\phi^2(h_{u-1,j}^{L})\right]^2 \leq C/n,$$

where the last line uses the independence of $\{h_{u-1,j}^{L}\}_j$ and Lemma 3. Therefore, we obtain

$$\mathbb{E}(h_{\ell,i}^{n,L} - h_{\ell,i}^{L})^2 \leq \frac{C}{L}\sum_{u=0}^{\ell-1}\mathbb{E}(h_{u,i}^{n,L} - h_{u,i}^{L})^2 + \frac{C}{n}.$$

By discrete Gronwall's inequality (Lemma 2), we have the desired result. $\qquad\square$

Next, to analyze the limit of $h_{\ell,i}^{L}$ as $L \to \infty$, we omit the subscript $i$ and view

$$\sqrt{\frac{T}{L}}z_\ell = w\left(\frac{\ell T}{L}\right) - w\left(\frac{(\ell-1)T}{L}\right)$$

for a standard Brownian motion $w$. Then we can write $h_\ell^L = h_{\ell T/L}^{(L)}$, where

$$dh_t^{(L)} = \sqrt{\mathbb{E}\phi^2(h_{t_L}^{(L)})}\,dw_t$$

and $t_L := \lfloor\frac{t}{T/L}\rfloor\frac{T}{L}$ for $t \in [0, T]$. Consider the McKean–Vlasov process

$$dh_t = \sqrt{\mathbb{E}\phi^2(h_t)}\,dw_t.$$

Then $\{h_t^{(L)}\}$ is just the Euler–Maruyama discretization for $\{h_t\}$ with step size $\Delta t = T/L$.

The following is a standard result (see e.g. (Sznitman, 1991, Section I.1)) and we only provide a sketch of the proof.

**Proposition 10.** *There exists a unique $\{h_t\}$ and*

$$\sup_{0\leq t\leq T}\mathbb{E}h_t^2 < \infty, \quad \mathbb{E}[h_t - h_{t_L}]^2 \leq C(t - t_L) \leq C/L. \tag{12}$$

*Proof of Proposition 10.* The evolution of $h_t$ can be written as

$$dh_t = \sigma(\mu_t)\,dw_t, \quad h_0 \sim \mathcal{N}(0, \|\boldsymbol{x}\|^2/d),$$

where $\mu_t = \mathrm{Law}(h_t)$ and

$$\sigma(\nu) := \sqrt{\int \phi^2(x)\,\nu(dx)}$$

for $\nu \in \mathcal{P}(\mathbb{R})$. Note that for any $X \sim \mu \in \mathcal{P}(\mathbb{R})$ and $Y \sim \nu \in \mathcal{P}(\mathbb{R})$, using Minkowski's inequality we have

$$|\sigma(\mu) - \sigma(\nu)| = |\sqrt{\mathbb{E}\phi^2(X)} - \sqrt{\mathbb{E}\phi^2(Y)}| \le \sqrt{\mathbb{E}[\phi(X) - \phi(Y)]^2}.$$

By Lipschitz property of $\phi$, we have

$$|\sigma(\mu) - \sigma(\nu)| \le CW_2(\mu,\nu), \tag{13}$$

where $W_2(\cdot,\cdot)$ is the Wasserstein metric on $\mathcal{P}(\mathbb{R})$. Therefore, $\sigma$ is a Lipschitz function. Now let

$$\mathcal{M} := \{\mu \in \mathcal{P}(\mathbb{C}([0,T]:\mathbb{R})) : \sup_{0 \le t \le T} \int x^2\,\mu_t(dx) < \infty\}.$$

For $\mu \in \mathcal{M}$, consider the process

$$dX_t = \sigma(\mu_t)\,dw_t, \quad X_0 = h_0.$$

It is well-defined and $\mathrm{Law}(X) \in \mathcal{M}$, by Lipschitz property of $\phi$. Denote the map from $\mu \in \mathcal{M}$ to $\mathrm{Law}(X) \in \mathcal{M}$ by $\Gamma$. For $\mu,\nu \in \mathcal{M}$, denote the Wasserstein metric by

$$W_{2,t}(\mu,\nu) := \inf\{\left(\mathbb{E}[\sup_{u \le t}|X_u - Y_u|^2]\right)^{1/2} : \mathrm{Law}(X) = \mu, \mathrm{Law}(Y) = \nu\}.$$

Now given $\mu,\nu \in \mathcal{M}$, let

$$dX_t = \sigma(\mu_t)\,dw_t, \quad dY_t = \sigma(\nu_t)\,dw_t, \quad X_0 = Y_0 = h_0.$$

Then using Doob's maximal inequality, we have

$$W_{2,t}^2(\Gamma(\mu),\Gamma(\nu)) \le \mathbb{E}[\sup_{u \le t}|X_u - Y_u|^2] = \mathbb{E}[\sup_{u \le t}|\int_0^u [\sigma(\mu_s) - \sigma(\nu_s)]\,dw_s|^2]$$

$$\le 4\mathbb{E}|\int_0^t [\sigma(\mu_s) - \sigma(\nu_s)]\,dw_s|^2 = 4\int_0^t [\sigma(\mu_s) - \sigma(\nu_s)]^2\,ds$$

$$\le C\int_0^t W_2^2(\mu_s,\nu_s)\,ds \le C\int_0^t W_{2,s}^2(\mu,\nu)\,ds.$$

Existence and uniqueness of $\{h_t\}$ then follows from standard arguments (cf. (Sznitman, 1991, Section I.1)). The first estimate in Eq. (12) follows from standard arguments on observing that $\phi$ is Lipscthiz and hence has linear growth. From this we immediately get the second estimate in Eq. (12). $\square$

The following result quantifies the error as $L \to \infty$. This is not the stronger result one would usually get for Euler–Maruyama approximations. But it is sufficient for our use and also will be used in later inductive arguments for traning steps.

**Proposition 11.** *For all $L \ge 1$,*

$$\sup_{\ell=0,1,\dots,L} \mathbb{E}[h_\ell^L - h_{\ell T/L}]^2 = \sup_{\ell=0,1,\dots,L} \mathbb{E}[h_{\ell T/L}^{(L)} - h_{\ell T/L}]^2 \le C/L.$$

*Proof of Proposition 11.* Let $s_L := \lfloor \frac{s}{T/L}\rfloor\frac{T}{L}$. Since $h_0^{(L)} = h_0$, we have

$$\mathbb{E}[h_\ell^L - h_{\ell T/L}]^2 = \mathbb{E}[h_{\ell T/L}^{(L)} - h_{\ell T/L}]^2$$

$$= \int_0^{\ell T/L} |\sqrt{\mathbb{E}\phi^2(h_{s_L}^{(L)})} - \sqrt{\mathbb{E}\phi^2(h_s)}|^2\,ds$$

$$\le 2\int_0^{\ell T/L} |\sqrt{\mathbb{E}\phi^2(h_{s_L}^{(L)})} - \sqrt{\mathbb{E}\phi^2(h_{s_L})}|^2\,ds + 2\int_0^{\ell T/L} |\sqrt{\mathbb{E}\phi^2(h_{s_L})} - \sqrt{\mathbb{E}\phi^2(h_s)}|^2\,ds$$

$$\le \frac{C}{L}\sum_{u=0}^{\ell-1} \mathbb{E}[h_u^L - h_{uT/L}]^2 + \frac{C}{L},$$

where the last line uses the Lipschitz property in Eq. (13) and Lemma 10. It then follows from discrete Gronwall's inequality (Lemma 2) that

$$\mathbb{E}[h_\ell^L - h_{\ell T/L}]^2 \leq \frac{C}{L}e^{C\ell/L}.$$

This completes the proof. $\qquad\square$

Combining Propositions 9 and 11, we get Proposition 2.

### C.2 FIRST BACKWARD — PROOF OF PROPOSITION 6

Suppose $\phi$ satisfies the assumption in Proposition 6 throughout.

Recall $\boldsymbol{g}_\ell$ in Eq. (3):

$$\boldsymbol{g}_L = \frac{\sqrt{n}}{\alpha}\frac{\partial f}{\partial \boldsymbol{h}_L} = \boldsymbol{v}, \quad \boldsymbol{g}_{\ell-1} = \frac{\sqrt{n}}{\alpha}\frac{\partial f}{\partial \boldsymbol{h}_\ell} = \boldsymbol{g}_\ell + \sqrt{\frac{T}{Ln}}\phi'(\boldsymbol{h}_{\ell-1}) \odot \boldsymbol{W}_\ell^\top \boldsymbol{g}_\ell, \quad \forall \ell \in [L].$$

Under the gradient independence assumption, this is equivalent to

$$g_{L,i}^{n,L} = v_i,$$

$$g_{\ell-1,i}^{n,L} = g_{\ell,i}^{n,L} + \sqrt{\frac{T}{Ln}}\phi'(h_{\ell-1,i}^{n,L})\|\boldsymbol{g}_\ell^{n,L}\|\tilde{z}_{\ell,i},$$

where $\{\tilde{z}_{\ell,i}\}$ are independent standard normal random variables and also independent of $\{z_{\ell,i}\}$.

We note that the evolution of the first backward $g_{\ell-1,i}^{n,L}$ is very similar to that of the first forward $h_{\ell,i}^{n,L}$, except that the last term involves an extra term $\phi'(h_{\ell-1,i}^{n,L})$ and it depends on $\|\boldsymbol{g}_\ell^{n,L}\|$ without through the activation function $\phi$. Therefore, the proof of Proposition 6 is very similar to that of Proposition 2, thanks to the assumption that $\phi'$ is Lipschitz. Hence we will only state the following results and omit most proofs. We will use $C$ to denote positive constants that depend only on $T$, $K_1$, $K_2$, and $\phi(0)$.

First we want to show as $n \to \infty$, $\{g_{\ell,i}^{n,L}\}$ converges to

$$g_{\ell-1,i}^L = g_{\ell,i}^L + \sqrt{\frac{T}{L}}\phi'(h_{\ell-1,i}^L)\sqrt{\mathbb{E}(g_{\ell,i}^L)^2}\tilde{z}_{\ell,i}.$$

**Lemma 4.** *For each $i \in \mathbb{N}$, $\sup_{L \geq 1} \sup_{\ell=0,\ldots,L} \mathbb{E}[g_{\ell,i}^L]^4 < \infty$ and $\inf_{L \geq 1} \inf_{\ell=0,\ldots,L} \mathbb{E}(g_{\ell,i}^L)^2 > 0$.*

*Proof of Lemma 4.* Proof of the first assertion is omitted. For the second assertion, since $g_{L,i}^{n,L} = v_i$ is independent of $\{h_{\ell,i}^L\}$ and $\{\tilde{z}_{\ell,i}\}$, we have

$$\mathbb{E}(g_{\ell,i}^L)^2 \geq \text{Var}(g_{\ell,i}^L) \geq \text{Var}(v_i) = C > 0.$$

This completes the proof. $\qquad\square$

**Proposition 12.** *For each $i \in \mathbb{N}$,*

$$\sup_{L \geq 1} \sup_{\ell=0,\ldots,L} \mathbb{E}(g_{\ell,i}^{n,L} - g_{\ell,i}^L)^2 \leq C/n.$$

Next, to analyze the limit of $g_{\ell,i}^L$ as $L \to \infty$, we omit the subscript $i$ and view

$$\sqrt{\frac{T}{L}}\tilde{z}_\ell = \tilde{w}\left(\frac{(\ell-1)T}{L}\right) - \tilde{w}\left(\frac{\ell T}{L}\right)$$

for a standard Brownian motion $\tilde{w}$ that goes backward in time. Then we can write $g_\ell^L = g_{\ell T/L}^{(L)}$, where

$$dg_t^{(L)} = \phi'(h_{t_L}^{(L)})\sqrt{\mathbb{E}(g_{\tilde{t}_L}^{(L)})^2}\, d\tilde{w}_t$$

and $\tilde{t}_L := \lceil \frac{t}{T/L} \rceil \frac{T}{L}$. Consider the McKean–Vlasov process (that goes backward in time)

$$dg_t = \sqrt{\mathbb{E}g_t^2} \, d\tilde{w}_t.$$

Then $\{g_t^{(L)}\}$ is just the Euler–Maruyama discretization for $\{g_t\}$ with step size $\Delta t = T/L$.

**Proposition 13.** *There exists a unique $\{g_t\}$ and*

$$\sup_{0 \le t \le T} \mathbb{E}g_t^2 < \infty, \quad \mathbb{E}[g_t - g_{\tilde{t}_L}]^2 \le C(\tilde{t}_L - t) \le C/L. \tag{14}$$

**Proposition 14.** *For all $L \ge 1$,*

$$\sup_{\ell=0,1,\dots,L} \mathbb{E}[g_\ell^L - g_{\ell T/L}]^2 = \sup_{\ell=0,1,\dots,L} \mathbb{E}[g_{\ell T/L}^{(L)} - g_{\ell T/L}]^2 \le C/L.$$

Combining Propositions 12 and 14, we get Proposition 6.

## D    Neural Feature Space Geometry in the RKHS Regime

In this section, we study the feature space geometry of infinite-width-depth ResNet at initialization in the kernel regime. We first establish the NNGP correspondence for this ResNet.

*Proof of Proposition 3.* By Proposition 2, the coordinate process of the hidden feature converges to the McKean–Vlasov SDE $dh_t = \sqrt{\mathbb{E}\phi^2(h_t)} \, dw_t$ with $h_0 \sim \mathcal{N}\left(0, \|x\|^2/d\right)$, where $w_t$ is a standard Brownian motion. Since this SDE is linear in the noise with a deterministic-in-time scalar diffusion, $h_t$ is Gaussian for all $t$, centered, and

$$\mathbb{E}[h_t^2] \;=\; \|x\|^2/d \;+\; \int_0^t \mathbb{E}\phi^2(h_s) \, ds.$$

For two inputs $x, \bar{x}$, let $(h_t, \bar{h}_t)$ denote the coupled limit (driven by the same weight noise, hence the same Brownian path). Then $(h_t, \bar{h}_t)$ is jointly Gaussian and its covariance $c_t(x, \bar{x}) := \mathbb{E}[h_t \bar{h}_t]$ satisfies

$$c_t(x, \bar{x}) := \mathbb{E}[h_t \bar{h}_t] = \frac{\langle x, \bar{x} \rangle}{d} + \int_0^t \mathbb{E}[\phi(h_s)\phi(\bar{h}_s)] \, ds,$$

In particular, $h_T$ and $\bar{h}_T$ are centered Gaussians with covariance $c_T(x, \bar{x})$.

Now consider the readout $f(x) = \alpha^{-1}\sqrt{n} \, \partial f/\partial h_L = n^{-1/2} \, v^\top h_L(x)$ with $\alpha = 1$ and $v \sim \mathcal{N}(0, I_n)$ independent of $h_L$. Conditioning on $h_L$, we have

$$f(x) \mid h_L \;\sim\; \mathcal{N}\left(0, \|h_L(x)\|^2/n\right)$$

with $\mathrm{Cov}\left((f, \bar{f}) \mid h_L\right) = h_L^\top \bar{h}_L/n$. By propagation of chaos and the law of large numbers at width $n \to \infty$, we have $\langle h_L, \bar{h}_L \rangle/n \to \mathbb{E}[h_T \bar{h}_T] = c_T(x, \bar{x})$ almost surely and in $L^1$. Therefore the finite-dimensional distributions of $\{f(x)\}_x$ converge to those of a centered Gaussian process with covariance kernel $C_T$. Since Gaussianity is preserved in the limit and covariances converge, we obtain convergence to the NNGP in the kernel regime, as claimed. $\square$

In Gaussian process regression, it is well known that the posterior mean predictor lies in the Reproducing Kernel Hilbert Space (RKHS) associated with the covariance kernel, while the posterior variance quantifies uncertainty outside this space. Therefore, the RKHS induced by the NNGP kernel $c_t$ provides a natural characterization of the geometry of the feature space at initialization: it determines which functions can be well-approximated by the ResNet under kernel regression, and how generalization behavior is shaped by the kernel structure.

This motivates the RKHS analysis below. We review the basic concepts of kernels and RKHS, introduce conditions for universality, and then study the specific kernels induced by neural network dual activations. Our main focus is to establish universality and strict positive definiteness of the induced kernels, and to analyze the nested structure of the expanding feature spaces $\{\mathcal{H}_t\}_{t \ge 0}$ defined by the time-dependent kernels $c_t$.

**Definition 2.** *Let $X$ be a set. A function $k : X \times X \to \mathbb{R}$ is called a **positive definite kernel**, or simply a **kernel**, if and only if it is symmetric, i.e., $k(x, \bar{x}) = k(\bar{x}, x)$, and for any finite collection $\{x_i\}_{i=1}^n \subset X$, the matrix $\boldsymbol{K} := [k(x_i, x_j)]_{i,j=1}^n$ is positive semi-definite. If, in addition, the matrix $K$ is strictly positive definite for any finite but distinct $\{x_i\}_{i=1}^n$, then the kernel $k$ is called a **strictly positive define kernel**.*

**Theorem 2** (Moore–Aronszajn). *Let $X$ be a nonempty set. A function $k : X \times X \to \mathbb{R}$ is a positive definite kernel if and only if there exists a unique Hilbert space $\mathcal{H}_k$ of functions $f : X \to \mathbb{R}$ s.t.:*

*1. For all $x \in X$, the function $k(\cdot, x) \in \mathcal{H}_k$,*

*2. For all $f \in \mathcal{H}_k$ and $x \in X$, the **reproducing property** holds:*

$$f(x) = \langle f, k(\cdot, x) \rangle_{\mathcal{H}_k},$$

*3. The linear span of $\{k(\cdot, x) : x \in X\}$ is dense in $\mathcal{H}_k$.*

*The space $\mathcal{H}_k$ is called the **Reproducing Kernel Hilbert Space (RKHS)** associated with the kernel $k$, and the function $k$ is the **reproducing kernel** of $\mathcal{H}_k$.*

**Definition 3** (Universal Kernel). *A continuous positive definite kernel $k$ on a compact set $X$ is called **universal kernel** if its associated RKHS $\mathcal{H}_k$ is **dense in** continuous function space $C(X)$ w.r.t. uniform norm.*

In this paper, we are primarily interested in dot-product kernels. (Micchelli et al., 2006, Corollary 8) provides a sufficient condition under which a dot-product kernel constructed from an entire function with positive coefficients is universal.

**Lemma 5** (Micchelli et al. (2006), Corollary 8). *Let $G(z) = \sum_{n=0}^{\infty} a_n z^n$ be an entire function on $\mathbb{C}$ with $a_n > 0$ for all $n \geq 0$. Then the induced dot-product kernel*

$$k(\boldsymbol{x}, \bar{\boldsymbol{x}}) = G(\boldsymbol{x}^\top \bar{\boldsymbol{x}})$$

*is universal on any compact subset of $\mathbb{R}^d$ or $\mathbb{C}^d$.*

Observe that

$$k_t(\boldsymbol{x}, \bar{\boldsymbol{x}}) = \mathbb{E}\phi(h_t)\phi(\bar{h}_t)$$

where $h_t$ is the limiting McKean–Vlasov SDE defined in Eq. (2) with $h_0 \sim \mathcal{N}(0, \|\boldsymbol{x}\|^2/d)$.

Here, we review some preliminary results of kernels induced by dual activation using Hermitian polynomials. Please review the detailed analysis in (Gao et al., 2025, Appendix F). As $\phi$ is $K_1$-Lipschitz, we have Hermitian expansion for $\phi$ as follows

$$\phi(\boldsymbol{x}) = \sum_{n=0}^{\infty} c_n h_n(\boldsymbol{x}),$$

where $c_n = \mathbb{E}_{z \sim \mathcal{N}(0,1)}[\phi(z) h_n(z)]$ is the Hermitian coefficient and $h_n$ are the (normalized) Hermite polynomials.

$$h_n(x) = \frac{1}{\sqrt{n!}} (-1)^n e^{x^2/2} D^n (e^{-x^2/2}).$$

Note that $\{h_n\}$ is an orthonormal basis in $L^2$ function space with standard Gaussian measure.

The *dual activation* $\hat{\phi} : [-1, 1] \to \mathbb{R}$ of $\phi$ is given by

$$\hat{\phi}(\rho) = \mathbb{E}\phi(u)\phi(v), \quad (u, v) \sim \mathcal{N}\left(0, \begin{bmatrix} 1 & \rho \\ \rho & 1 \end{bmatrix}\right)$$

Hence, the dual activation naturally induces a kernel defined on sphere $\mathbb{S}^{d-1}$:

$$k_\phi(\boldsymbol{x}, \bar{x}) = \hat{\phi}(\boldsymbol{x}^\top \bar{\boldsymbol{x}}).$$

By using the Hermitian expansion of $\phi$, we can further have an Hermitian expansion of $\hat{\phi}$:

$$\hat{\phi}(\rho) = \sum_{n=0}^{\infty} c_n^2 \rho^n.$$

Altogether, we have

$$k_\phi(\boldsymbol{x}, \bar{\boldsymbol{x}}) = \sum_{n=0}^{\infty} c_n^2 (\boldsymbol{x}^\top \bar{\boldsymbol{x}})^n.$$

It can be shown, as stated in (Gao et al., 2025, Theorem 11), that the kernel $k_\phi$ is strictly positive definite if $c_n^2 > 0$ for all $n$.

Assuming some additional nonlinear assumption on $\phi$, we can prove Propostion 4

**Lemma 6** (Propostion 4). *Suppose $\phi$ is $K_1$-Lipschitz continuous, nonlinear but non-polynomial. Then $k_0$ is strictly positive definite kernel and universal on $\mathbb{S}^{d-1}$.*

*Proof.* As $\boldsymbol{x}, \bar{\boldsymbol{x}} \in \mathbb{S}^{d-1}$, we have $k_0 = k_\phi$. As $\phi$ is non-polynomial, we have $c_n^2 > 0$ for all $n$. Hence, Lemma 5 implies $k_0$ universal on $\mathbb{S}^{d-1}$ as unit sphere is compact. $\qquad\square$

**Theorem 3** (Paulsen & Raghupathi (2016), Corollary 5.3). *Let $\mathcal{H}_i$ be RKHS's on a set $X$ with reproducing kernel $k_i$ for $i \in \{1, 2\}$. Then $k_2 - k_1$ is a kernel iff $\mathcal{H}_1$ is **contractively contained** in $\mathcal{H}_2$, that is, $\mathcal{H}_1 \subseteq \mathcal{H}_2$ and $\|f\|_{\mathcal{H}_2} \leq \|f\|_{\mathcal{H}_1}$ for all $f \in \mathcal{H}_1$.*

**Proposition 15** (Propostion 5). *For any $s \leq t < \infty$, $\mathcal{H}(c_s)$ is contractively contained in $\mathcal{H}(c_t)$.*

*Proof.* Observe that let $c_t(\boldsymbol{x}, \bar{\boldsymbol{x}}) := \mathbb{E}(h_t \bar{h}_t)$. Then we have

$$c_t(\boldsymbol{x}, \bar{\boldsymbol{x}}) = c_0(\boldsymbol{x}, \bar{\boldsymbol{x}}) + \int_0^t \mathbb{E}[\phi(h_s)\phi(\bar{h}_s)]ds = c_0(\boldsymbol{x}, \bar{\boldsymbol{x}}) + \int_0^t k_s(\boldsymbol{x}, \bar{\boldsymbol{x}})ds.$$

Clearly, $c_t$ is also a kernel since $k_s$ is. Hence, for any $t_1 \leq t_2$, we have $\mathcal{H}(c_{t_1})$ is contractively contained in $\mathcal{H}(c_{t_2})$. The proof is complete by using Theorem 3. $\qquad\square$

# E GRADIENT COMPUTATION

Recall that we have defined the ResNet as follows:

$$\boldsymbol{h}_0 = \frac{1}{\sqrt{d}} \boldsymbol{U} \boldsymbol{x},$$

$$\boldsymbol{h}_\ell = \boldsymbol{h}_{\ell-1} + \sqrt{\frac{T}{Ln}} \boldsymbol{W}_\ell \phi(\boldsymbol{h}_{\ell-1}), \quad \forall \ell \in [L],$$

$$f(\boldsymbol{x}) = \frac{\alpha}{\sqrt{n}} \boldsymbol{v}^\top \boldsymbol{h}_L.$$

Then we consider the backward propagation of gradients as follows:

$$\boldsymbol{g}_L = \frac{\sqrt{n}}{\alpha} \frac{\partial f}{\partial \boldsymbol{h}_L} = \boldsymbol{v},$$

$$\boldsymbol{g}_{\ell-1} = \frac{\sqrt{n}}{\alpha} \frac{\partial f}{\partial \boldsymbol{h}_{\ell-1}} = \boldsymbol{g}_\ell + \sqrt{\frac{T}{Ln}} \phi'(\boldsymbol{h}_{\ell-1}) \odot \boldsymbol{W}_\ell^\top \boldsymbol{g}_\ell, \quad \forall \ell \in [L].$$

The gradients of $f$ w.r.t. trainable parameters are given by

$$\frac{\partial f}{\partial \boldsymbol{v}} = \frac{\alpha}{\sqrt{n}} \boldsymbol{h}_L,$$

$$\frac{\partial f}{\partial \boldsymbol{W}_\ell} = \sqrt{\frac{T}{Ln}} \frac{\partial f}{\partial \boldsymbol{h}_\ell} \phi(\boldsymbol{h}_{\ell-1})^\top,$$

$$\frac{\partial f}{\partial \boldsymbol{U}} = \frac{1}{\sqrt{d}} \frac{\partial f}{\partial \boldsymbol{h}_0} \boldsymbol{x}^\top.$$

Given a loss function $\mathcal{L}$, the SGD with a single sample is given by

$$\boldsymbol{v}^+ = \boldsymbol{v} - \eta \mathcal{L}'(f, y)\frac{\alpha}{\sqrt{n}}\boldsymbol{h}_L,$$

$$\boldsymbol{W}_\ell^+ = \boldsymbol{W}_\ell - \eta \mathcal{L}'(f, y)\sqrt{\frac{T}{Ln}}\frac{\partial f}{\partial \boldsymbol{h}_\ell}\phi(\boldsymbol{h}_{\ell-1})^\top = \boldsymbol{W}_\ell - \eta \frac{\alpha}{\sqrt{n}}\mathcal{L}'(f, y)\sqrt{\frac{T}{Ln}}\boldsymbol{g}_\ell\phi(\boldsymbol{h}_{\ell-1})^\top,$$

$$\boldsymbol{U}^+ = \boldsymbol{U} - \eta \mathcal{L}'(f, y)\frac{1}{\sqrt{d}}\frac{\partial f}{\partial \boldsymbol{h}_0}\boldsymbol{x}^\top = \boldsymbol{U} - \eta \frac{\alpha}{\sqrt{n}}\mathcal{L}'(f, y)\frac{1}{\sqrt{d}}\boldsymbol{g}_0\boldsymbol{x}^\top.$$

Then, after $k$ step gradient updates, the forward propagation becomes:

$$\boldsymbol{h}_0^{(k)} = \frac{1}{\sqrt{d}}\boldsymbol{U}^{(k)}\boldsymbol{x}^{(k)}$$

$$= \frac{1}{\sqrt{d}}\left[\boldsymbol{U} - \eta \sum_{i=0}^{k-1}\mathcal{L}'(f^{(i)}, y^{(i)})\frac{1}{\sqrt{d}}\frac{\partial f^{(i)}}{\partial \boldsymbol{h}_0^{(i)}}\boldsymbol{x}^{(i)\top}\right]\boldsymbol{x}^{(k)}$$

$$= \frac{1}{\sqrt{d}}\boldsymbol{U}\boldsymbol{x}^{(k)} - \eta \frac{\alpha}{\sqrt{n}}\sum_{i=0}^{k-1}\mathcal{L}'(f^{(i)}, y^{(i)})\frac{\left\langle \boldsymbol{x}^{(i)}, \boldsymbol{x}^{(k)}\right\rangle}{d}\boldsymbol{g}_0^{(i)},$$

$$\boldsymbol{h}_\ell^{(k)} = \boldsymbol{h}_{\ell-1}^{(k)} + \sqrt{\frac{T}{Ln}}\boldsymbol{W}_\ell^{(k)}\phi(\boldsymbol{h}_{\ell-1}^{(k)})$$

$$= \boldsymbol{h}_{\ell-1}^{(k)} + \sqrt{\frac{T}{Ln}}\left[\boldsymbol{W}_\ell - \eta \sum_{i=0}^{k-1}\mathcal{L}'(f^{(i)}, y^{(i)})\sqrt{\frac{T}{Ln}}\frac{\partial f^{(i)}}{\partial \boldsymbol{h}_\ell^{(i)}}\phi(\boldsymbol{h}_{\ell-1}^{(i)})^\top\right]\phi(\boldsymbol{h}_{\ell-1}^{(k)})$$

$$= \boldsymbol{h}_{\ell-1}^{(k)} - \eta \frac{\alpha}{\sqrt{n}}\frac{T}{L}\sum_{i=0}^{k-1}\mathcal{L}'(f^{(i)}, y^{(i)})\frac{\left\langle \phi(\boldsymbol{h}_{\ell-1}^{(i)}), \phi(\boldsymbol{h}_{\ell-1}^{(k)})\right\rangle}{n}\boldsymbol{g}_\ell^{(i)}$$

$$\quad + \sqrt{\frac{T}{Ln}}\boldsymbol{W}_\ell\phi(\boldsymbol{h}_{\ell-1}^{(k)}),$$

$$f^{(k)} = \frac{\alpha}{\sqrt{n}}\left\langle \boldsymbol{v}^{(k)}, \boldsymbol{h}_L^{(k)}\right\rangle$$

$$= \frac{\alpha}{\sqrt{n}}\left[\boldsymbol{v} - \eta \sum_{i=0}^{k-1}\mathcal{L}'(f^{(i)}, y^{(i)})\frac{\alpha}{\sqrt{n}}\boldsymbol{h}_L^{(i)}\right]^\top \boldsymbol{h}_L^{(k)}$$

$$= \frac{\alpha}{\sqrt{n}}\boldsymbol{v}^\top\boldsymbol{h}_L^{(k)} - \eta\alpha^2\sum_{i=0}^{k-1}\mathcal{L}'(f^{(i)}, y^{(i)})\frac{\left\langle \boldsymbol{h}_L^{(i)}, \boldsymbol{h}_L^{(k)}\right\rangle}{n}.$$

Consequentially, the backward propagation becomes

$$\boldsymbol{g}_L^{(k)} = \boldsymbol{v}^{(k)} = \boldsymbol{v} - \eta \sum_{i=0}^{k-1}\mathcal{L}'(f^{(i)}, y^{(i)})\frac{\alpha}{\sqrt{n}}\boldsymbol{h}_L^{(i)},$$

$$\boldsymbol{g}_{\ell-1}^{(k)} = \boldsymbol{g}_\ell^{(k)} + \sqrt{\frac{T}{Ln}}\phi'(\boldsymbol{h}_{\ell-1}^{(k)})\odot \boldsymbol{W}_\ell^{(k)\top}\boldsymbol{g}_\ell^{(k)}$$

$$= \boldsymbol{g}_\ell^{(k)} + \sqrt{\frac{T}{Ln}}\phi'(\boldsymbol{h}_{\ell-1}^{(k)})\odot \left[\boldsymbol{W}_\ell - \eta \sum_{i=0}^{k-1}\mathcal{L}'(f^{(i)}, y^{(i)})\sqrt{\frac{T}{Ln}}\frac{\partial f^{(i)}}{\partial \boldsymbol{h}_\ell^{(i)}}\phi(\boldsymbol{h}_{\ell-1}^{(i)})^\top\right]^\top \boldsymbol{g}_\ell^{(k)}$$

$$= \boldsymbol{g}_\ell^{(k)} - \eta \frac{\alpha}{\sqrt{n}}\frac{T}{L}\sum_{i=0}^{k-1}\mathcal{L}'(f^{(i)}, y^{(i)})\frac{\left\langle \boldsymbol{g}_\ell^{(i)}, \boldsymbol{g}_\ell^{(k)}\right\rangle}{n}\left[\phi(\boldsymbol{h}_{\ell-1}^{(i)})\odot \phi'(\boldsymbol{h}_{\ell-1}^{(k)})\right]$$

$$\quad + \sqrt{\frac{T}{Ln}}\phi'(\boldsymbol{h}_{\ell-1}^{(k)})\odot \boldsymbol{W}_\ell^\top\boldsymbol{g}_\ell^{(k)}.$$

# F  FEATURE LEARNING DYNAMICS IN THE INFINITE-WIDTH LIMIT

In this section, we provide intuition for Proposition 8, aimed at readers who may not be familiar with the Tensor Program framework.

To analyze how the feature space evolves during training, we focus on the *feature learning regime* with scaling $\alpha = 1/\sqrt{n}$ and learning rate $\eta = \eta_c n$, where $\eta_c > 0$ is a fixed constant. Under this setting, the forward and backward recursions take the form:

$$\boldsymbol{h}_0^{(k)} = \frac{1}{\sqrt{d}} \boldsymbol{U} \boldsymbol{x}^{(k)} - \eta_c \sum_{i=0}^{k-1} \mathcal{L}'(f^{(i)}, y^{(i)}) \frac{\langle \boldsymbol{x}^{(i)}, \boldsymbol{x}^{(k)} \rangle}{d} \boldsymbol{g}_0^{(i)},$$

$$\boldsymbol{h}_\ell^{(k)} = \boldsymbol{h}_{\ell-1}^{(k)} - \eta_c \frac{T}{L} \sum_{i=0}^{k-1} \mathcal{L}'(f^{(i)}, y^{(i)}) \frac{\langle \phi(\boldsymbol{h}_{\ell-1}^{(i)}), \phi(\boldsymbol{h}_{\ell-1}^{(k)}) \rangle}{n} \boldsymbol{g}_\ell^{(i)} + \sqrt{\frac{T}{Ln}} \boldsymbol{W}_\ell \phi(\boldsymbol{h}_{\ell-1}^{(k)}),$$

$$f^{(k)} = \frac{1}{n} \boldsymbol{v}^\top \boldsymbol{h}_L^{(k)} - \eta_c \sum_{i=0}^{k-1} \mathcal{L}'(f^{(i)}, y^{(i)}) \frac{\langle \boldsymbol{h}_L^{(i)}, \boldsymbol{h}_L^{(k)} \rangle}{n},$$

and the backward recursion:

$$\boldsymbol{g}_L^{(k)} = \boldsymbol{v} - \eta_c \sum_{i=0}^{k-1} \mathcal{L}'(f^{(i)}, y^{(i)}) \boldsymbol{h}_L^{(i)},$$

$$\boldsymbol{g}_{\ell-1}^{(k)} = \boldsymbol{g}_\ell^{(k)} - \eta_c \frac{T}{L} \sum_{i=0}^{k-1} \mathcal{L}'(f^{(i)}, y^{(i)}) \frac{\langle \boldsymbol{g}_\ell^{(i)}, \boldsymbol{g}_\ell^{(k)} \rangle}{n} \left[ \phi(\boldsymbol{h}_{\ell-1}^{(i)}) \odot \phi'(\boldsymbol{h}_{\ell-1}^{(k)}) \right]$$

$$+ \sqrt{\frac{T}{Ln}} \, \phi'(\boldsymbol{h}_{\ell-1}^{(k)}) \odot \boldsymbol{W}_\ell^\top \boldsymbol{g}_\ell^{(k)}.$$

Observe that the training process indeed defines a valid Tensor Program: each vector $\{\boldsymbol{h}_\ell^{(k)}, \boldsymbol{g}_\ell^{(k)}\}$ is generated sequentially from previously constructed vectors or from the initial random parameters $\{\boldsymbol{v}, \boldsymbol{W}_\ell, \boldsymbol{U}\}$ via standard TP operations (MatMul, Nonlin, Moment). Consequently, we may directly apply the Master Theorem of (Yang & Hu, 2021, Theorem 7.4) to characterize their infinite-width mean-field limits.

In particular, for any finite collection of valid TP vectors $\{\boldsymbol{h}_s\}_{s=1}^M$ and any sufficiently regular test function $\psi : \mathbb{R}^M \to \mathbb{R}$, we have

$$\frac{1}{n} \sum_{i=1}^n \psi(\boldsymbol{h}_{1,i}, \boldsymbol{h}_{2,i}, \ldots, \boldsymbol{h}_{M,i}) \xrightarrow[n \to \infty]{\text{a.s.}} \mathbb{E}\left[\psi(Z^{\boldsymbol{h}_1}, Z^{\boldsymbol{h}_2}, \ldots, Z^{\boldsymbol{h}_M})\right],$$

where each $Z^{\boldsymbol{h}}$ denotes the *mean-field variable* associated with $\boldsymbol{h}$, i.e., the limiting distribution of a typical coordinate of $\boldsymbol{h}$ as width grows.

**First Forward Pass**  At initialization, the ResNet can be written as

$$\boldsymbol{h}_0 = \frac{1}{\sqrt{d}} \boldsymbol{U} \boldsymbol{x},$$

$$\boldsymbol{h}_\ell = \boldsymbol{h}_{\ell-1} + \sqrt{\frac{\tau}{n}} \boldsymbol{W}_\ell \phi(\boldsymbol{h}_{\ell-1}),$$

$$f(\boldsymbol{x}) = \frac{1}{n} \boldsymbol{v}^\top \boldsymbol{h}_L,$$

where $\tau := T/L$. Since $\boldsymbol{U}_{ij} \sim \mathcal{N}(0, 1)$ are i.i.d., each coordinate of $\boldsymbol{h}_0$ is i.i.d. with distribution $Z^{\boldsymbol{h}_0} \sim \mathcal{N}(0, \|\boldsymbol{x}\|^2/d)$. Applying the nonlinearity $\phi$ element-wise preserves independence across coordinates, so $\phi(\boldsymbol{h}_0)$ also has i.i.d. coordinates. Similarly, because $\boldsymbol{W}_{\ell,ij} \sim \mathcal{N}(0, 1)$ are i.i.d., each coordinate of $\frac{1}{\sqrt{n}} \boldsymbol{W}_\ell \phi(\boldsymbol{h}_{\ell-1})$ is approximately Gaussian with mean zero and variance $\frac{1}{n} \|\phi(\boldsymbol{h}_{\ell-1})\|^2$. By the Master Theorem, as $n \to \infty$, these coordinates converge in distribution to a

mean-field variable $Z^{\boldsymbol{W}_\ell \phi_{\ell-1}}$ with variance $\mathbb{E}[\phi^2(Z^{\boldsymbol{h}_{\ell-1}})]$, where, inductively, we use the fact that each coordinate of $\boldsymbol{h}_\ell$ converges to a mean-field variable $Z^{\boldsymbol{h}_\ell}$ in the infinite-width limit. Finally, since $\boldsymbol{v}_i \sim \mathcal{N}(0,1)$ are i.i.d., the network output satisfies

$$f(\boldsymbol{x}) = \frac{1}{n} \sum_{i=1}^{n} \boldsymbol{v}_i \boldsymbol{h}_{L,i} \xrightarrow[n\to\infty]{a.s.} \mathring{f} := \mathbb{E}[Z^{\boldsymbol{v}} Z^{\boldsymbol{h}_L}],$$

by the law of large numbers.

Hence, in the infinite-width limit, the ResNet converges to the mean-field process

$$\begin{aligned} Z^{\boldsymbol{h}_0} &= \mathcal{N}\big(0, \|\boldsymbol{x}\|^2/d\big), \\ Z^{\boldsymbol{h}_\ell} &= Z^{\boldsymbol{h}_{\ell-1}} + \sqrt{\tau}\, Z^{\boldsymbol{W}_\ell \phi_{\ell-1}}, \quad \forall \ell \in [L], \\ \mathring{f} &= \mathbb{E}[Z^{\boldsymbol{v}} Z^{\boldsymbol{h}_L}], \end{aligned}$$

where $\{Z^{\boldsymbol{W}_\ell \phi_{\ell-1}}\}_{\ell=1}^{L}$ are centered jointly Gaussian random variables with covariance

$$\mathrm{Cov}\big(Z^{\boldsymbol{W}_\ell \phi_{\ell-1}}, Z^{\boldsymbol{W}_k \phi_{k-1}}\big) = \delta_{\ell,k}\, \mathbb{E}\big[\phi(Z^{\boldsymbol{h}_{\ell-1}})\phi(Z^{\boldsymbol{h}_{k-1}})\big], \quad \forall \ell, k \in [L],$$

and independent of $Z^{\boldsymbol{v}} \sim \mathcal{N}(0,1)$.

**First Backward Pass** In the feature learning regime, the first backward recursion for computing gradients is

$$\begin{aligned} \boldsymbol{g}_L &= \boldsymbol{v}, \\ \boldsymbol{g}_{\ell-1} &= \boldsymbol{g}_\ell + \sqrt{\tfrac{\tau}{n}}\, \phi'(\boldsymbol{h}_{\ell-1}) \odot \boldsymbol{W}_\ell^\top \boldsymbol{g}_\ell. \end{aligned}$$

Since $\boldsymbol{v}$ has i.i.d. standard Gaussian coordinates independent of the forward activations $\{\boldsymbol{h}_\ell\}$, the recursion in the mean-field limit begins with $Z^{\boldsymbol{g}_L} = Z^{\boldsymbol{v}} \sim \mathcal{N}(0,1)$. Because the output head $\boldsymbol{v}$ is not reused elsewhere, the *gradient independence assumption (GIA)* (Yang, 2020b) holds in the infinite-width limit. This permits replacing $\boldsymbol{W}_\ell^\top$ in the backward pass with an independent copy $\widetilde{\boldsymbol{W}}_\ell^\top$, so that $\widetilde{\boldsymbol{W}}_\ell^\top \boldsymbol{g}_\ell$ is independent of the forward variables $\boldsymbol{h}_\ell$. The coordinates of $\frac{1}{\sqrt{n}} \widetilde{\boldsymbol{W}}_\ell^\top \boldsymbol{g}_\ell$ are then approximately i.i.d. Gaussian, and by the Master Theorem converge to a mean-field random variable $Z^{\boldsymbol{W}_\ell^\top \boldsymbol{g}_\ell}$.

Thus, in the infinite-width limit, the backward recursion is characterized by

$$\begin{aligned} Z^{\boldsymbol{g}_L} &= Z^{\boldsymbol{v}} \sim \mathcal{N}(0,1), \\ Z^{\boldsymbol{g}_{\ell-1}} &= Z^{\boldsymbol{g}_\ell} + \sqrt{\tau}\, \phi'(Z^{\boldsymbol{h}_{\ell-1}})\, Z^{\boldsymbol{W}_\ell^\top \boldsymbol{g}_\ell}, \quad \forall \ell \in [L], \end{aligned}$$

where $\{Z^{\boldsymbol{W}_\ell^\top \boldsymbol{g}_\ell}\}_{\ell=1}^{L}$ are centered jointly Gaussian random variables with covariance

$$\mathrm{Cov}\big(Z^{\boldsymbol{W}_\ell^\top \boldsymbol{g}_\ell}, Z^{\boldsymbol{W}_k^\top \boldsymbol{g}_k}\big) = \delta_{\ell,k}\, \mathbb{E}[Z^{\boldsymbol{g}_\ell} Z^{\boldsymbol{g}_k}], \quad \forall \ell, k \in [L].$$

**Second Forward Pass** After one gradient update, the forward pass with a new input $\bar{\boldsymbol{x}}$ is given by

$$\begin{aligned} \bar{\boldsymbol{h}}_0 &= \tfrac{1}{\sqrt{d}}\, \boldsymbol{U}\bar{\boldsymbol{x}} - \eta_c \mathcal{L}'(f,y) \tfrac{\langle \boldsymbol{x}, \bar{\boldsymbol{x}} \rangle}{d}\, \boldsymbol{g}_0, \\ \bar{\boldsymbol{h}}_\ell &= \bar{\boldsymbol{h}}_{\ell-1} - \tau \eta_c \mathcal{L}'(f,y) \tfrac{\langle \phi(\boldsymbol{h}_{\ell-1}), \phi(\bar{\boldsymbol{h}}_{\ell-1}) \rangle}{n}\, \boldsymbol{g}_\ell + \sqrt{\tfrac{\tau}{n}}\, \boldsymbol{W}_\ell \phi(\bar{\boldsymbol{h}}_{\ell-1}). \end{aligned}$$

Since $f \to \mathring{f}$ in the infinite-width limit, continuity of $\mathcal{L}'$ ensures $\mathcal{L}'(f,y) \to \mathcal{L}'(\mathring{f},y)$. Hence,

$$Z^{\bar{\boldsymbol{h}}_0} = Z^{\boldsymbol{U}\bar{\boldsymbol{x}}} - \eta_c \mathcal{L}'(\mathring{f},y) \tfrac{\langle \boldsymbol{x}, \bar{\boldsymbol{x}} \rangle}{d} Z^{\boldsymbol{g}_0},$$

where $Z^{\boldsymbol{U}\bar{\boldsymbol{x}}}$ is centered Gaussian correlated with $Z^{\boldsymbol{U}\boldsymbol{x}}$, with covariance $\mathrm{Cov}(Z^{\boldsymbol{U}\boldsymbol{x}}, Z^{\boldsymbol{U}\bar{\boldsymbol{x}}}) = \frac{1}{d}\boldsymbol{x}^\top \bar{\boldsymbol{x}}$.

For hidden states, the inner product $\frac{1}{n} \langle \phi(\boldsymbol{h}_{\ell-1}), \phi(\bar{\boldsymbol{h}}_{\ell-1}) \rangle$ is a Moment operation in the TP, and by the Master Theorem converges to $\mathbb{E}[\phi(Z^{\boldsymbol{h}_{\ell-1}})\phi(Z^{\bar{\boldsymbol{h}}_{\ell-1}})]$.

Next consider $\frac{1}{\sqrt{n}} \boldsymbol{W}_\ell \phi(\bar{\boldsymbol{h}}_{\ell-1})$. If we adopt the decoupled analysis (replacing $\boldsymbol{W}_\ell^\top$ in the backward pass with $\tilde{\boldsymbol{W}}_\ell^\top$), then by the CLT heuristic the coordinates converge to a Gaussian random variable $Z^{\boldsymbol{W}_\ell \bar{\phi}_{\ell-1}}$, correlated with $Z^{\boldsymbol{W}_\ell \phi_{\ell-1}}$ with

$$\operatorname{Cov}\left( Z^{\boldsymbol{W}_\ell \phi_{\ell-1}}, Z^{\boldsymbol{W}_\ell \bar{\phi}_{\ell-1}} \right) = \mathbb{E}[\phi(Z^{\boldsymbol{h}_{\ell-1}})\phi(Z^{\bar{\boldsymbol{h}}_{\ell-1}})].$$

This CLT heuristic is valid only because we assume $\widetilde{\boldsymbol{W}}_\ell^\top$ is used in the backward pass; otherwise, $\bar{\boldsymbol{h}}_\ell$ and $\boldsymbol{W}_\ell^\top$ are strongly correlated through $\boldsymbol{g}_{\ell-1}$. Before exploring this coupling scenario, the second forward pass under the decoupling scenario is described as follows:

$$\begin{aligned} Z^{\bar{\boldsymbol{h}}_0} &= Z^{\boldsymbol{U}\bar{\boldsymbol{x}}} - \eta_c \mathcal{L}'(\mathring{f}, y) \frac{\langle \boldsymbol{x}, \bar{\boldsymbol{x}}\rangle}{d} Z^{\boldsymbol{g}_0}, \\ Z^{\bar{\boldsymbol{h}}_\ell} &= Z^{\bar{\boldsymbol{h}}_{\ell-1}} - \tau \eta_c \mathcal{L}'(\mathring{f}, y) \mathbb{E}[\phi(Z^{\boldsymbol{h}_{\ell-1}})\phi(Z^{\bar{\boldsymbol{h}}_{\ell-1}})] Z^{\boldsymbol{g}_\ell} + \sqrt{\tau} Z^{\boldsymbol{W}_\ell \bar{\phi}_{\ell-1}}. \end{aligned}$$

Now, we focus on the normal gradient update to expose the effect of the reuse of $\boldsymbol{W}_\ell$ in the backward pass. For intuition, set $\phi = \mathrm{id}$. Expanding $\bar{\boldsymbol{h}}_{\ell-1}$ yields

$$\bar{\boldsymbol{h}}_{\ell-1} = \bar{\boldsymbol{h}}_{\ell-2} + \tau a_{\ell-2} \boldsymbol{g}_\ell + \tau a_{\ell-2}\sqrt{\frac{\tau}{n}} \boldsymbol{W}_\ell^\top \boldsymbol{g}_\ell + \sqrt{\frac{\tau}{n}} \boldsymbol{W}_{\ell-1}\bar{\boldsymbol{h}}_{\ell-2},$$

where

$$a_\ell := -\eta_c \mathcal{L}'(f, y)\frac{\langle \phi(\boldsymbol{h}_\ell), \phi(\bar{\boldsymbol{h}}_\ell)\rangle}{n} \xrightarrow{\text{a.s.}} \mathring{a}_\ell := -\eta_c \mathcal{L}'(\mathring{f}, y)\mathbb{E}[\phi(Z^{\boldsymbol{h}_\ell})\phi(Z^{\bar{\boldsymbol{h}}_\ell})],$$

since $a_\ell$ is a valid TP scalar. Substituting into $\sqrt{\frac{\tau}{n}} \boldsymbol{W}_\ell \bar{\boldsymbol{h}}_{\ell-1}$ yields

$$\sqrt{\frac{\tau}{n}} \boldsymbol{W}_\ell \bar{\boldsymbol{h}}_{\ell-1} = \sqrt{\frac{\tau}{n}} \boldsymbol{W}_\ell \left( \bar{\boldsymbol{h}}_{\ell-2} + \tau a_{\ell-2}\boldsymbol{g}_\ell + \sqrt{\frac{\tau}{n}} \boldsymbol{W}_{\ell-1}\bar{\boldsymbol{h}}_{\ell-2} \right) + \tau^2 a_{\ell-2}\frac{1}{n} \boldsymbol{W}_\ell \boldsymbol{W}_\ell^\top \boldsymbol{g}_\ell.$$

The $i$-th coordinate of the $\frac{1}{n} \boldsymbol{W}_\ell \boldsymbol{W}_\ell^\top \boldsymbol{g}_\ell$ can be decomposed into

$$\boldsymbol{g}_{\ell,i} \cdot \frac{1}{n}\sum_j \boldsymbol{W}_{\ell,ij}^2 + \frac{1}{n}\sum_j \sum_{k\neq i} \boldsymbol{W}_{\ell,ij}\boldsymbol{W}_{\ell,kj}\boldsymbol{g}_{\ell,k}.$$

By the law of large numbers, the first term converges to $Z^{\boldsymbol{g}_\ell}$, and the second converges (by CLT) to a Gaussian field $\hat{Z}^{\boldsymbol{W}_\ell \bar{\phi}_{\ell-1}}$. Notably, under GIA, we replace $\boldsymbol{W}_{\ell,ij}^2$ with $\boldsymbol{W}_{\ell,ij}\widetilde{\boldsymbol{W}}_{\ell,ij}$. Hence, the first term vanishes as $n \to \infty$ because of the independence and zero mean.

Thus, for $\phi = \mathrm{id}$,

$$\left[ \sqrt{\frac{\tau}{n}} \boldsymbol{W}_\ell \bar{\boldsymbol{h}}_{\ell-1} \right]_i \xrightarrow{\text{a.s.}} \sqrt{\tau} \hat{Z}^{\boldsymbol{W}_\ell \bar{\phi}_{\ell-1}} + \tau^2 \mathring{a}_{\ell-2} Z^{\boldsymbol{g}_\ell}.$$

Here $\hat{Z}^{\boldsymbol{W}_\ell \bar{\phi}_{\ell-1}}$ is Gaussian, correlated with $Z^{\boldsymbol{W}_\ell \phi_{\ell-1}}$ from the first forward pass with covariance $\mathbb{E}[\phi(Z^{\boldsymbol{h}_{\ell-1}})\phi(Z^{\bar{\boldsymbol{h}}_{\ell-1}})]$.

Putting the pieces together, the mean-field feature dynamics after one gradient step are

$$\begin{aligned} Z^{\bar{\boldsymbol{h}}_0} &= Z^{\boldsymbol{U}\bar{\boldsymbol{x}}} - \eta_c \mathcal{L}'(\mathring{f}, y)\frac{\langle \boldsymbol{x}, \bar{\boldsymbol{x}}\rangle}{d} Z^{\boldsymbol{g}_0}, \\ Z^{\bar{\boldsymbol{h}}_\ell} &= Z^{\bar{\boldsymbol{h}}_{\ell-1}} - \tau \eta_c \mathcal{L}'(\mathring{f}, y)\mathbb{E}[Z^{\boldsymbol{h}_\ell} Z^{\bar{\boldsymbol{h}}_{\ell-1}}] Z^{\boldsymbol{g}_\ell} + \sqrt{\tau} \hat{Z}^{\boldsymbol{W}_\ell \bar{\phi}_{\ell-1}} \\ &\quad - \tau^2 \eta_c \mathcal{L}'(\mathring{f}, y)\mathbb{E}[Z^{\boldsymbol{h}_{\ell-2}} Z^{\bar{\boldsymbol{h}}_{\ell-2}}] Z^{\boldsymbol{g}_\ell}. \end{aligned}$$

Comparing this result with the decoupling scenario, the final correction term reflects the additional interaction in both forward and backward paths due to the reuse of $\boldsymbol{W}_\ell$. Moreover, its scaling is $\tau^2$ due to depth-adaptive ResNet normalization. By the Euler–Maruyama convergence perspective, this higher-order term vanishes as $L \to \infty$, provided the other quantities remain well behaved.

**Second Backward Pass** Analogously, we describe the backward propagation after one step of gradient descent in the mean-field limit. Given the second forward pass with input $\bar{\boldsymbol{x}}$, the second backward recursion is

$$\begin{aligned} \bar{\boldsymbol{g}}_L &= \boldsymbol{v} - \eta_c \mathcal{L}'(f, y)\boldsymbol{h}_L, \\ \bar{\boldsymbol{g}}_{\ell-1} &= \bar{\boldsymbol{g}}_\ell - \tau \eta_c \mathcal{L}'(f, y)\frac{\langle \boldsymbol{g}_\ell, \bar{\boldsymbol{g}}_\ell\rangle}{n}\left[ \phi(\boldsymbol{h}_{\ell-1}) \odot \phi'(\bar{\boldsymbol{h}}_{\ell-1}) \right] + \sqrt{\frac{\tau}{n}} \phi'(\bar{\boldsymbol{h}}_{\ell-1}) \odot \boldsymbol{W}_\ell^\top \bar{\boldsymbol{g}}_\ell. \end{aligned}$$

Assume we decouple the two backward passes by replacing $\boldsymbol{W}_\ell^\top$ with an independent copy $\widetilde{\boldsymbol{W}}_\ell^\top$ in the second backward recursion. Then, in the mean-field limit, the coordinates of $\frac{1}{\sqrt{n}}\widetilde{\boldsymbol{W}}_\ell^\top \bar{\boldsymbol{g}}_\ell$ converge (by the Master Theorem) to a centered Gaussian random variable $Z^{\boldsymbol{W}_\ell^\top \bar{\boldsymbol{g}}_\ell}$, correlated with $Z^{\boldsymbol{W}_\ell^\top \boldsymbol{g}_\ell}$ from the first backward pass via

$$\mathrm{Cov}\big(Z^{\boldsymbol{W}_\ell^\top \bar{\boldsymbol{g}}_\ell}, Z^{\boldsymbol{W}_\ell^\top \boldsymbol{g}_\ell}\big) \;=\; \mathbb{E}\big[Z^{\bar{\boldsymbol{g}}_\ell} Z^{\boldsymbol{g}_\ell}\big].$$

Passing to the limit (and using continuity of $\mathcal{L}'$ so that $\mathcal{L}'(f,y) \to \mathcal{L}'(\mathring{f},y)$), we obtain

$$Z^{\bar{\boldsymbol{g}}_L} = Z^{\boldsymbol{g}_L} - \eta_c \mathcal{L}'(\mathring{f},y)\, Z^{\boldsymbol{h}_L},$$
$$Z^{\bar{\boldsymbol{g}}_{\ell-1}} = Z^{\bar{\boldsymbol{g}}_\ell} - \tau\eta_c \mathcal{L}'(\mathring{f},y)\, \mathbb{E}\big[Z^{\boldsymbol{g}_\ell} Z^{\bar{\boldsymbol{g}}_\ell}\big]\, \phi(Z^{\boldsymbol{h}_{\ell-1}})\,\phi'(Z^{\bar{\boldsymbol{h}}_{\ell-1}}) + \sqrt{\tau}\, \phi'(Z^{\bar{\boldsymbol{h}}_{\ell-1}})\, Z^{\tilde{\boldsymbol{W}}_\ell^\top \bar{\boldsymbol{g}}_\ell}.$$

When the same weights $\boldsymbol{W}_\ell^\top$ are reused in both backward passes, additional correlations appear. For intuition, set $\phi = \mathrm{id}$ so $\phi' \equiv 1$. The second backward state expands as

$$\bar{\boldsymbol{g}}_\ell = \bar{\boldsymbol{g}}_{\ell+1} + \tau b_{\ell+1}\, \boldsymbol{h}_{\ell-1} + \tau b_{\ell+1}\sqrt{\tfrac{\tau}{n}}\, \boldsymbol{W}_\ell \boldsymbol{h}_{\ell-1} + \sqrt{\tfrac{\tau}{n}}\, \boldsymbol{W}_{\ell+1}^\top \bar{\boldsymbol{g}}_{\ell+1},$$

with

$$b_\ell := -\eta_c\, \mathcal{L}'(f,y)\, \frac{\langle \boldsymbol{g}_\ell, \bar{\boldsymbol{g}}_\ell\rangle}{n} \xrightarrow{\text{a.s.}} \mathring{b}_\ell := -\eta_c\, \mathcal{L}'(\mathring{f},y)\, \mathbb{E}\big[Z^{\boldsymbol{g}_\ell} Z^{\bar{\boldsymbol{g}}_\ell}\big],$$

where the convergence follows from the law of large numbers intuition via the Master Theorem. Consequently, the term $\sqrt{\tfrac{\tau}{n}}\, \boldsymbol{W}_\ell^\top \bar{\boldsymbol{g}}_\ell$ contains the correlation-driving factor

$$\tfrac{1}{n}\, \boldsymbol{W}_\ell^\top \boldsymbol{W}_\ell\, \boldsymbol{h}_{\ell-1},$$

whose $i$-th coordinate decomposes into

$$\frac{1}{n}\sum_j \boldsymbol{W}_{\ell,ji}^2\, \boldsymbol{h}_{\ell-1,i} \;+\; \frac{1}{n}\sum_j \sum_{k\neq i} \boldsymbol{W}_{\ell,ji}\boldsymbol{W}_{\ell,jk}\, \boldsymbol{h}_{\ell-1,k}.$$

By the law of large numbers, the first term converges to $Z^{\boldsymbol{h}_{\ell-1}}$, while the second behaves like a CLT term and is absorbed into a Gaussian field. Hence,

$$\left[\sqrt{\tfrac{\tau}{n}}\, \boldsymbol{W}_\ell^\top \bar{\boldsymbol{g}}_\ell\right]_i \xrightarrow{\text{a.s.}} \sqrt{\tau}\, \hat{Z}^{\boldsymbol{W}_\ell^\top \bar{\boldsymbol{g}}_\ell} \;+\; \tau^2\, \mathring{b}_{\ell+1}\, Z^{\boldsymbol{h}_{\ell-1}},$$

where $\hat{Z}^{\boldsymbol{W}_\ell^\top \bar{\boldsymbol{g}}_\ell}$ is centered Gaussian and correlated with $Z^{\boldsymbol{W}_\ell^\top \boldsymbol{g}_\ell}$ via

$$\mathrm{Cov}\big(\hat{Z}^{\boldsymbol{W}_\ell^\top \bar{\boldsymbol{g}}_\ell}, Z^{\boldsymbol{W}_\ell^\top \boldsymbol{g}_\ell}\big) \;=\; \mathbb{E}\big[Z^{\bar{\boldsymbol{g}}_\ell} Z^{\boldsymbol{g}_\ell}\big].$$

Collecting terms (still with $\phi = \mathrm{id}$), the coupled mean-field recursion becomes

$$Z^{\bar{\boldsymbol{g}}_L} = Z^{\boldsymbol{g}_L} - \eta_c \mathcal{L}'(\mathring{f},y)\, Z^{\boldsymbol{h}_L},$$
$$Z^{\bar{\boldsymbol{g}}_{\ell-1}} = Z^{\bar{\boldsymbol{g}}_\ell} - \tau\eta_c \mathcal{L}'(\mathring{f},y)\, \mathbb{E}\big[Z^{\boldsymbol{g}_\ell} Z^{\bar{\boldsymbol{g}}_\ell}\big]\, Z^{\boldsymbol{h}_{\ell-1}}$$
$$+ \sqrt{\tau}\, Z^{\boldsymbol{W}_\ell^\top \bar{\boldsymbol{g}}_\ell} \;-\; \tau^2\eta_c \mathcal{L}'(\mathring{f},y)\, \mathbb{E}\big[Z^{\boldsymbol{g}_{\ell+1}} Z^{\bar{\boldsymbol{g}}_{\ell+1}}\big]\, Z^{\boldsymbol{h}_{\ell-1}}.$$

**Remark 1.** *The last (higher-order) correction arises from reusing $\boldsymbol{W}_\ell$ and $\boldsymbol{W}_\ell^\top$ in both the forward and backward passes, and scales as $\tau^2$ due to depth-adaptive normalization in ResNets. By Euler–Maruyama convergence, this term vanishes as $L \to \infty$ (with $\tau = T/L$), assuming the remaining quantities are well behaved. Moreover, the intuition developed here for the second forward and backward passes extends directly to any $k$-th pass.*

## G  DEPTH CONVERGENCE — PROOF OF THEOREM 1

In this section we use Proposition 8 to prove the convergence in Theorem 1 and show that the rate of convergence as $L \to \infty$ is $1/L$. Recall Assumption 1. For ease of writing, we consider the one sample case.

The limit as $n \to \infty$ for the $K$-th iteration can be written as

$$h_\ell^{(K),L} = h_{\ell-1}^{(K),L} - \eta_0 \frac{T}{L} \sum_{k=0}^{K-1} \mathcal{L}'(k,L) g_\ell^{(k),L} \mathbb{E}(\phi(h_{\ell-1}^{(k),L})\phi(h_{\ell-1}^{(K),L})) + \sqrt{\frac{T}{L}} z_\ell^{(K),L}$$

$$- \eta_0 \frac{T^2}{L^2} \sum_{k=0}^{K-1} \mathcal{L}'(k,L) g_\ell^{(k),L} \mathbb{E}(\phi(h_{\ell-2}^{(k),L})\phi(h_{\ell-2}^{(K),L})) \mathbb{E}(\phi'(h_{\ell-1}^{(k),L})\phi'(h_{\ell-1}^{(K),L})),$$

$$g_{\ell-1}^{(K),L} = g_\ell^{(K),L} - \eta_0 \frac{T}{L} \phi'(h_{\ell-1}^{(K),L}) \sum_{k=0}^{K-1} \mathcal{L}'(k,L) \phi(h_{\ell-1}^{(k),L}) \mathbb{E}(g_\ell^{(k),L}, g_\ell^{(K),L}) + \sqrt{\frac{T}{L}} \phi'(h_{\ell-1}^{(K),L}) \tilde{z}_\ell^{(K),L}$$

$$- \eta_0 \frac{T^2}{L^2} \phi'(h_{\ell-1}^{(K),L}) \sum_{k=0}^{K-1} \mathcal{L}'(k,L) \phi(h_{\ell-1}^{(k),L}) \mathbb{E}(g_{\ell+1}^{(k),L} g_{\ell+1}^{(K),L}) \mathbb{E}[\phi'(h_\ell^{(k),L})\phi'(h_\ell^{(K),L})],$$

$$(15)$$

where $\{(z_\ell^{(k),L})_k, (\tilde{z}_\ell^{(k),L})_k : \ell = 1, \ldots, L\}$ are independent Gaussian random vectors with mean 0 and variance-covariance matrix

$$\mathrm{Cov}(z_\ell^{(k),L}, z_\ell^{(k'),L}) = \mathbb{E}[\phi(h_{\ell-1}^{(k),L})\phi(h_{\ell-1}^{(k'),L})],$$

$$\mathrm{Cov}(\tilde{z}_\ell^{(k),L}, \tilde{z}_\ell^{(k'),L}) = \mathbb{E}[g_\ell^{(k),L} g_\ell^{(k'),L}],$$

and

$$h_0^{(K),L} = \frac{\|\boldsymbol{x}\|}{\sqrt{d}} u(0) - \eta_0 \sum_{k=0}^{K-1} \mathcal{L}'(k,L) g_0^{(k),L} \frac{\langle \boldsymbol{x}, \boldsymbol{x} \rangle}{d},$$

$$g_L^{(K),L} = v(0) - \eta_0 \sum_{k=0}^{K-1} \mathcal{L}'(k,L) h_L^{(k),L},$$

$$\mathcal{L}'(k,L) = \mathcal{L}'(\mathbb{E}[g_L^{(k),L} h_L^{(k),L}], y),$$

and $u(0), v(0)$ are standard Gaussian.

Letting $L \to \infty$, we expect to have

$$dh_t^{(K)} = -\eta_0 \sum_{k=0}^{K-1} \mathcal{L}'(k) g_t^{(k)} \mathbb{E}[\phi(h_t^{(k)})\phi(h_t^{(K)})] dt + dw_t^{(K)}, \quad \forall t \in [0,T],$$

$$dg_t^{(K)} = -\eta_0 \phi'(h_t^{(K)}) \sum_{k=0}^{K-1} \mathcal{L}'(k) \phi(h_t^{(k)}) \mathbb{E}[g_t^{(k)} g_t^{(K)}] dt + \phi'(h_t^{(K)}) d\tilde{w}_t^{(K)}, \quad \forall t \in [0,T].$$

where $\{(w_t^{(k)})_k, (\tilde{w}_t^{(k)})_k : \ell = 1, \ldots, L\}$ are indpendent Brownian motions with mean 0 and cross-variations

$$d\langle w^{(k)}, w^{(k')} \rangle_t = \mathbb{E}[\phi(h_t^{(k)})\phi(h_t^{(k')})] \, dt,$$

$$d\langle \tilde{w}^{(k)}, \tilde{w}^{(k')} \rangle_t = \mathbb{E}[g_t^{(k)} g_t^{(k')}] \, dt,$$

and

$$h_0^{(K)} = \frac{\|\boldsymbol{x}\|}{\sqrt{d}} u(0) - \eta_0 \sum_{k=0}^{K-1} \mathcal{L}'(k) g_0^{(k)} \frac{\|\boldsymbol{x}\|^2}{d},$$

$$g_T^{(K)} = v(0) - \eta_0 \sum_{k=0}^{K-1} \mathcal{L}'(k) h_T^{(k)},$$

$$\mathcal{L}'(k) = \mathcal{L}'(\mathbb{E}[g_T^{(k)} h_T^{(k)}], y).$$

**Remark 2.** *(a) The evolution of $g_t^{(K)}$ is written for ease of notation and is interpreted backward from $t = T$ to $t = 0$. This should not to be confused with the classic notion of backward stochastic differential equations. The precise meaning, instead, is that $(g_t, \tilde{w}_t) = (\widehat{g}_{T-t}, \widehat{w}_{T-t})$ and*

$$d\widehat{g}_t^{(K)} = -\eta_0 \phi'(h_{T-t}^{(K)}) \sum_{k=0}^{K-1} \mathcal{L}'(k)\phi(h_{T-t}^{(k)})\mathbb{E}[\widehat{g}_t^{(k)}\widehat{g}_t^{(K)}]dt + \phi'(h_{T-t}^{(K)})d\widehat{w}_t^{(K)}, \quad \forall t \in [0, T].$$

*(b) The first forward $\{h_t^{(0)}\}$ is adapted to the driven Brownian motion. Due to the backpropagation, $g_t^{(0)}$ and $\{h_t^{(k)}, g_t^{(k)}, k = 1, 2, \dots\}$ are not adapted any more. However, thanks to the deterministic diffusion coefficients in front of $dw_t$ and $d\tilde{w}_t$, which are automatically adapted, the SDEs are well-posed, as is justified in Propositions 10 and 13 above and Proposition 16 below.*

For ease of analysis, we write the above dynamics of $h_t := (h_t^{(0)}, \dots, h_t^{(K)})$ and $g_t := (g_t^{(0)}, \dots, g_t^{(K)})$ in the following more standard manner of McKean–Vlasov equations:

$$dh_t = b_t \, dt + \sigma_t \, dW_t,$$
$$dg_t = c_t \, dt + D_t\theta_t \, dB_t,$$

where $b_t = (b_{t,k})_{k=0}^K$ and $c_t = (c_{t,k})_{k=0}^K$ are vectors given by

$$b_{t,k} = -\eta_0 \sum_{i=0}^{k-1} \mathcal{L}'(i)g_t^{(i)}\mathbb{E}[\phi(h_t^{(i)})\phi(h_t^{(k)})],$$

$$c_{t,k} := -\eta_0\phi'(h_t^{(k)}) \sum_{i=0}^{k-1} \mathcal{L}'(i)\phi(h_t^{(i)})\mathbb{E}[g_t^{(i)}g_t^{(k)}],$$

$D_t$ is a diagonal matrix given by

$$D_t = \text{diag}\{\phi'(h_t^{(0)}), \dots, \phi'(h_t^{(K)})\},$$

$\sigma_t$ and $\theta_t$ are (the Cholesky decomposition) such that

$$\sigma_t\sigma_t^\top = \Sigma_t := \mathbb{E}[(\phi(h_t^{(0)}), \dots, \phi(h_t^{(K)}))^\top(\phi(h_t^{(0)}), \dots, \phi(h_t^{(K)}))],$$
$$\theta_t\theta_t^\top = \Theta_t := \mathbb{E}[(g_t^{(0)}, \dots, g_t^{(K)})^\top(g_t^{(0)}, \dots, g_t^{(K)})],$$

and $W_t$ and $B_t$ are independent $(K + 1)$-dimensional standard Brownian motions.

We note that the above system is nested: when $K$ increases by 1, one simply adds one additional dimension to the evolution of $h_t$ and $g_t$. This allows us to apply induction arguments in the proofs of later results. We also note that the existence of solutions to the above system is already guaranteed via the convergence of $n \to \infty$.

Denote by $\| \cdot \|$ the Frobenius norm of matrices (and vectors). Denote by $\lambda_k(A)$ the eigenvalues of a symmetric matrix $A$.

**Lemma 7.** *If $\{h_t^{(k)}, g_t^{(k)}, k = 0, 1, \dots, K\}$ is a solution, then*

$$\sup_{0 \le t \le T} \mathbb{E}\|h_t\|^2 < \infty, \quad \sup_{0 \le t \le T} \mathbb{E}\|g_t\|^2 < \infty.$$

*Proof of Lemma 7.* We will prove by induction. The statement holds for $K = 0$ by Propositions 10 and 13.

Now suppose the statement holds for $K$, namely

$$\sup_{0 \le t \le T} \sum_{k=0}^K \mathbb{E}[h_t^{(k)}]^2 < \infty, \quad \sup_{0 \le t \le T} \sum_{k=0}^K \mathbb{E}[g_t^{(k)}]^2 < \infty.$$

We will show that

$$\sup_{0 \le t \le T} \mathbb{E}[h_t^{(K+1)}]^2 < \infty, \quad \sup_{0 \le t \le T} \mathbb{E}[g_t^{(K+1)}]^2 < \infty.$$

For $h_t^{(K+1)}$, using Cauchy-Schwarz inequality, Lipschitz property of $\phi$, and induction assumption, we have

$$\mathbb{E}[h_t^{(K+1)}]^2 \leq C\mathbb{E}[h_0^{(K+1)}]^2 + C\sum_{k=0}^{K} \mathbb{E}\left(\int_0^t g_s^{(k)}\mathbb{E}[\phi(h_s^{(k)})\phi(h_s^{(K+1)})]\,ds\right)^2 + C\mathbb{E}[w_t^{(K+1)}]^2$$

$$\leq C + C\sum_{k=0}^{K}\int_0^t \mathbb{E}[g_s^{(k)}]^2\mathbb{E}\phi^2(h_s^{(k)})\mathbb{E}\phi^2(h_s^{(K+1)})\,ds + C\int_0^t \mathbb{E}\phi^2(h_s^{(K+1)})\,ds$$

$$\leq C + C\int_0^t \mathbb{E}[h_s^{(K+1)}]^2\,ds.$$

It then follows from Gronwall's lemma that $\sup_{0\leq t\leq T} \mathbb{E}[h_t^{(K+1)}]^2 < \infty$. Since $\phi'$ is bounded, using similar arguments as above we can get $\sup_{0\leq t\leq T} \mathbb{E}[g_t^{(K+1)}]^2 < \infty$. Therefore the statement holds for $K+1$ and this completes the proof by induction. $\qquad\square$

**Proposition 16.** *Suppose Assumption 1 holds. Then pathwise uniqueness holds for $\{h_t^{(k)}, g_t^{(k)}, k = 0, 1, \ldots, K\}$.*

*Proof of Proposition 16.* We will prove by induction. By Propositions 10 and 13, $h_t^{(0)}$ and $g_t^{(0)}$ are unique. So the statement holds for $K = 0$.

Now suppose the statement holds for $K$, namely $h_t^{(k)}$ and $g_t^{(k)}$, $k = 0, 1, \ldots, K$, are unique. We will show that $h_t^{(k)}$ and $g_t^{(k)}$, $k = 0, 1, \ldots, K+1$ are unique. Consider the solution $(h_t^{(k)}, g_t^{(k)})_{k=0}^{K+1}$ and any other solution $(\tilde{h}_t^{(k)}, \tilde{g}_t^{(k)})_{k=0}^{K+1}$. By the induction assumption on uniqueness, we must have $(h_t^{(k)}, g_t^{(k)})_{k=0}^{K} = (\tilde{h}_t^{(k)}, \tilde{g}_t^{(k)})_{k=0}^{K}$. Recall

$$\sigma_t\sigma_t^\top = \Sigma_t = \mathbb{E}[(\phi(h_t^{(0)}), \ldots, \phi(h_t^{(K+1)}))^\top(\phi(h_t^{(0)}), \ldots, \phi(h_t^{(K+1)}))]$$

and let

$$\tilde{\sigma}_t\tilde{\sigma}_t^\top = \tilde{\Sigma}_t = \mathbb{E}[(\phi(\tilde{h}_t^{(0)}), \ldots, \phi(\tilde{h}_t^{(K+1)}))^\top(\phi(\tilde{h}_t^{(0)}), \ldots, \phi(\tilde{h}_t^{(K+1)}))]$$

$$= \mathbb{E}[(\phi(h_t^{(0)}), \ldots, \phi(h_t^{(K)}), \phi(\tilde{h}_t^{(K+1)}))^\top(\phi(h_t^{(0)}), \ldots, \phi(h_t^{(K)}), \phi(\tilde{h}_t^{(K+1)}))].$$

Write $\Sigma_t$ in block matrix form

$$\Sigma_{11,t} = \mathbb{E}[(\phi(h_t^{(0)}), \ldots, \phi(h_t^{(K)}))^\top(\phi(h_t^{(0)}), \ldots, \phi(h_t^{(K)}))],$$

$$\Sigma_{21,t} = \Sigma_{12,t}^\top = \mathbb{E}[\phi(h_t^{(K+1)})(\phi(h_t^{(0)}), \ldots, \phi(h_t^{(K)}))],$$

$$\Sigma_{22,t} = \mathbb{E}[\phi^2(h_t^{(K+1)})],$$

corresponding to coordinates $0, 1, \ldots, K$ and $K+1$. Also write $\sigma_t$, $\tilde{\Sigma}_t$ and $\tilde{\sigma}_t$ is the similar way. Then by Cholesky decomposition, we have

$$\sigma_{11,t}\sigma_{11,t}^\top = \Sigma_{11,t}, \qquad\qquad \tilde{\sigma}_{11,t} = \sigma_{11,t},$$

$$\sigma_{12,t} = 0, \qquad\qquad \tilde{\sigma}_{12,t} = 0,$$

$$\sigma_{21,t}^\top = \sigma_{11,t}^{-1}\Sigma_{12,t}, \qquad\qquad \tilde{\sigma}_{21,t}^\top = \tilde{\sigma}_{11,t}^{-1}\tilde{\Sigma}_{12,t} = \sigma_{11,t}^{-1}\tilde{\Sigma}_{12,t},$$

$$\sigma_{22,t} = \sqrt{\Sigma_{22,t} - \sigma_{21,t}\sigma_{21,t}^\top}, \qquad\qquad \tilde{\sigma}_{22,t} = \sqrt{\tilde{\Sigma}_{22,t} - \tilde{\sigma}_{21,t}\tilde{\sigma}_{21,t}^\top}.$$

By Lipschitz property of $\phi$, we have

$$\|\Sigma_{12,t} - \tilde{\Sigma}_{12,t}\|^2 + \|\Sigma_{22,t} - \tilde{\Sigma}_{22,t}\|^2 \leq C\mathbb{E}[h_t^{(K+1)} - \tilde{h}_t^{(K+1)}]^2. \tag{16}$$

By Assumption 1, there exits some $\varepsilon > 0$ such that all eigenvalues of $\Sigma_t$ are at least $\varepsilon$. It then follows from the eigenvalue interlacing theorem (of principal submatrix) that all eigenvalues of $\Sigma_{11,t}$ are at least $\varepsilon$. Then we have

$$\|\sigma_{11,t}^{-1}\|^2 = \text{trace}((\sigma_{11,t}^{-1})^\top\sigma_{11,t}^{-1}) = \text{trace}(\Sigma_{11,t}^{-1}) = \sum_{k=0}^{K}\frac{1}{\lambda_k(\Sigma_{11,t})} \leq \frac{K+1}{\varepsilon}. \tag{17}$$

Therefore

$$\|\sigma_{21,t} - \tilde\sigma_{21,t}\|^2 = \|\sigma_{11,t}^{-1}(\Sigma_{12,t} - \tilde\Sigma_{12,t})\|^2 \le \|\sigma_{11,t}^{-1}\|^2\|\Sigma_{12,t} - \tilde\Sigma_{12,t}\|^2 \le C\mathbb{E}[h_t^{(K+1)} - \tilde h_t^{(K+1)}]^2.$$
(18)

where the last inequality uses Eq. (17) and Eq. (16). Similarly,

$$
\begin{aligned}
|\sigma_{21,t}\sigma_{21,t}^\top - \tilde\sigma_{21,t}\tilde\sigma_{21,t}^\top|^2 &= |(\sigma_{21,t} - \tilde\sigma_{21,t})(\sigma_{21,t} + \tilde\sigma_{21,t})^\top|^2 \\
&= |(\sigma_{21,t} - \tilde\sigma_{21,t})\sigma_{11,t}^{-1}(\Sigma_{12,t} + \tilde\Sigma_{12,t})|^2 \le \|\sigma_{21,t} - \tilde\sigma_{21,t}\|^2\|\sigma_{11,t}^{-1}\|^2\|\Sigma_{12,t} + \tilde\Sigma_{12,t}\|^2 \\
&\le C\mathbb{E}[h_t^{(K+1)} - \tilde h_t^{(K+1)}]^2,
\end{aligned}
$$
(19)

where we have used Lemma 7 to get $\|\Sigma_{12,t} + \tilde\Sigma_{12,t}\|^2 \le C$. Also, note that

$$\sigma_{22,t}^2 = \Sigma_{22,t} - \sigma_{21,t}\sigma_{21,t}^\top = \Sigma_{22,t} - \Sigma_{21,t}\Sigma_{11,t}^{-1}\Sigma_{12,t}$$

is the Schur complement of the block $\Sigma_{11,t}$ of the matrix $\Sigma_t$, so that its eigenvalues are at least $\varepsilon$ as well. Therefore $\sigma_{22,t} \ge \sqrt{\varepsilon}$ and hence

$$
\begin{aligned}
\|\sigma_{22,t} - \tilde\sigma_{22,t}\|^2 &= \left(\frac{\sigma_{22,t}^2 - \tilde\sigma_{22,t}^2}{\sigma_{22,t} + \tilde\sigma_{22,t}}\right)^2 \le \frac{1}{\varepsilon}[(\Sigma_{22,t} - \sigma_{21,t}\sigma_{21,t}^\top) - (\tilde\Sigma_{22,t} - \tilde\sigma_{21,t}\tilde\sigma_{21,t}^\top)]^2 \\
&\le C|\Sigma_{22,t} - \tilde\Sigma_{22,t}|^2 + C|\sigma_{21,t}\sigma_{21,t}^\top - \tilde\sigma_{21,t}\tilde\sigma_{21,t}^\top|^2 \le C\mathbb{E}[h_t^{(K+1)} - \tilde h_t^{(K+1)}]^2,
\end{aligned}
$$
(20)

where the last inequality uses Eq. (16) and Eq. (19). Now note that

$$
\begin{aligned}
h_u^{(K+1)} - \tilde h_u^{(K+1)} &= -\eta_0 \sum_{k=0}^{K} \int_0^u \mathcal{L}'(k)g_s^{(k)}\mathbb{E}[\phi(h_s^{(k)})(\phi(h_s^{(K+1)}) - \phi(\tilde h_s^{(K+1)}))]\,ds \\
&\quad + \int_0^u (\sigma_{21,s} - \tilde\sigma_{21,s}, \sigma_{22,s} - \tilde\sigma_{22,s})\,dW_s.
\end{aligned}
$$

Therefore

$$
\begin{aligned}
\mathbb{E}[\sup_{u \le t} |h_u^{(K+1)} - \tilde h_u^{(K+1)}|^2] &\le C\sum_{k=0}^{K} \mathbb{E}\left[\sup_{u \le t}\left|\int_0^u \mathcal{L}'(k)g_s^{(k)}\mathbb{E}[\phi(h_s^{(k)})(\phi(h_s^{(K+1)}) - \phi(\tilde h_s^{(K+1)}))]\,ds\right|^2\right] \\
&\quad + C\mathbb{E}\left[\sup_{u \le t}\left|\int_0^u (\sigma_{21,s} - \tilde\sigma_{21,s}, \sigma_{22,s} - \tilde\sigma_{22,s})\,dW_s\right|^2\right].
\end{aligned}
$$
(21)

Here using Cauchy-Schwarz inequality, we can bound the first term on the right side by

$$
C\sum_{k=0}^{K}\mathbb{E}\int_0^t \left|g_s^{(k)}\mathbb{E}[\phi(h_s^{(k)})(\phi(h_s^{(K+1)}) - \phi(\tilde h_s^{(K+1)}))]\right|^2 ds
$$

$$
\le C\int_0^t \mathbb{E}[\phi(h_s^{(K+1)}) - \phi(\tilde h_s^{(K+1)})]^2\,ds \le C\int_0^t \mathbb{E}[\sup_{u \le s}|h_u^{(K+1)} - \tilde h_u^{(K+1)}|^2]\,ds.
$$

Using Doob's maximal inequality, we can bound the second term on the right side of Eq. (21) by

$$
C\mathbb{E}\left|\int_0^t (\sigma_{21,s} - \tilde\sigma_{21,s}, \sigma_{22,s} - \tilde\sigma_{22,s})\,dW_s\right|^2 = C\int_0^t [\|\sigma_{21,s} - \tilde\sigma_{21,s}\|^2 + \|\sigma_{22,s} - \tilde\sigma_{22,s}\|^2]\,ds
$$

$$
\le C\int_0^t \mathbb{E}[h_s^{(K+1)} - \tilde h_s^{(K+1)}]^2\,ds \le C\int_0^t \mathbb{E}[\sup_{u \le s}|h_u^{(K+1)} - \tilde h_u^{(K+1)}|^2]\,ds.
$$

where the first inequality uses Eq. (18) and Eq. (20). Combining above three estimates, we have

$$
\mathbb{E}[\sup_{u \le t}|h_u^{(K+1)} - \tilde h_u^{(K+1)}|^2] \le C\int_0^t \mathbb{E}[\sup_{u \le s}|h_u^{(K+1)} - \tilde h_u^{(K+1)}|^2]\,ds.
$$

It then follows from Gronwall's inequality that

$$
\mathbb{E}[\sup_{u \le T}|h_u^{(K+1)} - \tilde h_u^{(K+1)}|^2] = 0.
$$

This gives uniqueness of $h_t^{(K+1)}$. Since $\phi'$ is bounded, similar arguments as above give uniqueness of $g_t^{(K+1)}$. Therefore the statement holds for $K+1$ and this completes the proof by induction. $\qquad\square$

Before proving the convergence rate as $L \to \infty$, we will need the following two preparation results. Recall $t_L := \lfloor \frac{t}{T/L} \rfloor \frac{T}{L}$ and $\tilde{t}_L := \lceil \frac{t}{T/L} \rceil \frac{T}{L}$ are the times corresponding to the discrete step.

**Lemma 8.** *If $\{h_t^{(k)}, g_t^{(k)}, k = 0, 1, \ldots, \kappa\}$ is a solution, then*

$$\mathbb{E}\|h_t - h_{t_L}\|^2 \leq C(t - t_L) \leq C/L, \quad \mathbb{E}\|g_t - g_{\tilde{t}_L}\|^2 \leq C(\tilde{t}_L - t) \leq C/L.$$

*Proof of Lemma 8.* Using Lemma 7 and Lipscthiz property of $\phi$, we can deduce

$$\mathbb{E}\|b_t\|^2 \leq C, \qquad \|\sigma_t\|^2 = \mathrm{trace}(\sigma_t \sigma_t^\top) = \mathrm{trace}(\Sigma_t) \leq C,$$

and similarly $\mathbb{E}\|c_t\|^2 \leq C$ and $\|\theta_t\|^2 \leq C$. These give the desired result. $\qquad \square$

Recall the infinite width limit $h_\ell^L := (h_t^{(0),L}, \ldots, h_t^{(K),L})$ and $g_\ell^L := (g_t^{(0),L}, \ldots, g_t^{(K),L})$.

**Lemma 9.** $\sup_{L \geq 1} \sup_{\ell = 0, \ldots, L} \mathbb{E}\|h_\ell^L\|^2 < \infty$ *and* $\sup_{L \geq 1} \sup_{\ell = 0, \ldots, L} \mathbb{E}\|g_\ell^L\|^2 < \infty$.

*Proof of Lemma 9.* We will prove by induction. By Lemmas 3 and 4, the statement holds for $K = 0$.

Now suppose the statement holds for $K$, namely

$$\sup_{L \geq 1} \sup_{\ell = 0, \ldots, L} \sum_{k=0}^{K} \mathbb{E}[h_\ell^{(k),L}]^2 < \infty, \quad \sup_{L \geq 1} \sup_{\ell = 0, \ldots, L} \sum_{k=0}^{K} \mathbb{E}[g_\ell^{(k),L}]^2 < \infty.$$

We will show that

$$\sup_{L \geq 1} \sup_{\ell = 0, \ldots, L} \mathbb{E}[h_\ell^{(K+1),L}]^2 < \infty, \quad \sup_{L \geq 1} \sup_{\ell = 0, \ldots, L} \mathbb{E}[g_\ell^{(K+1),L}]^2 < \infty.$$

From the evolution of $h_\ell^{(K+1),L}$ and independence of $z_\cdot^{(K+1),L}$, we have

$$\mathbb{E}[h_\ell^{(K+1),L}]^2$$

$$\leq 4\mathbb{E}[h_0^{(K+1),L}]^2 + 4\mathbb{E}\left[\sum_{u=1}^{\ell} \sqrt{\frac{T}{L}} z_u^{(K+1),L}\right]^2$$

$$+ 4\mathbb{E}\left[\sum_{u=1}^{\ell} \eta_0 \frac{T}{L} \sum_{k=0}^{K} \mathcal{L}'(k, L) g_u^{(k),L} \mathbb{E}(\phi(h_{u-1}^{(k),L}) \phi(h_{u-1}^{(K+1),L}))\right]^2$$

$$+ 4\mathbb{E}\left[\sum_{u=1}^{\ell} \eta_0 \frac{T^2}{L^2} \sum_{k=0}^{K} \mathcal{L}'(k, L) g_u^{(k),L} \mathbb{E}(\phi(h_{u-2}^{(k),L}) \phi(h_{u-2}^{(K+1),L})) \mathbb{E}(\phi'(h_{u-1}^{(k),L}) \phi'(h_{u-1}^{(K+1),L}))\right]^2$$

$$\leq C + \frac{C}{L} \sum_{u=1}^{\ell} \mathbb{E}\phi^2(h_{u-1}^{(K+1),L}) + \frac{C\ell}{L^2} \sum_{u=1}^{\ell} \mathbb{E}\phi^2(h_{u-1}^{(K+1),L})$$

$$+ \frac{C\ell}{L^4} \sum_{u=1}^{\ell} \left[\mathbb{E}\phi^2(h_{u-2}^{(K+1),L}) + \mathbb{E}[\phi'(h_{u-1}^{(K+1),L})]^2\right]$$

$$\leq C + \frac{C}{L} \sum_{u=0}^{\ell-1} \mathbb{E}[h_u^{(K+1),L}]^2.$$

It then follows from discrete Gronwall's lemma again that

$$\mathbb{E}[h_\ell^{(K+1),L}]^2 \leq C e^{C\ell/L}.$$

Therefore $\sup_{L \geq 1} \sup_{\ell = 0, \ldots, L} \mathbb{E}[h_\ell^{(K+1),L}]^2 < \infty$. A similar argument applied to Eq. (15) gives $\sup_{L \geq 1} \sup_{\ell = 0, \ldots, L} \mathbb{E}[g_\ell^{(K+1),L}]^2 < \infty$ and hence the statement also holds for $K + 1$. This completes the proof by induction. $\qquad \square$

Now we couple $h_\ell^L$ and $g_\ell^L$ with $h_t$ and $g_t$ respectively, and state our result on the convergence rate of $1/L$ as $L \to \infty$. Denote by $L_t := \lfloor \frac{t}{T/L} \rfloor$, $L_s := \lfloor \frac{s}{T/L} \rfloor$, $\tilde{L}_t := \lceil \frac{t}{T/L} \rceil$, and $\tilde{L}_s := \lceil \frac{s}{T/L} \rceil$ for $s, t \in [0, T]$. We can write $h_\ell^L = h_{\ell T/L}^{(L)}$ and $g_\ell^L = g_{\ell T/L}^{(L)}$, where $h_t^{(L)}$ and $g_t^{(L)}$ are continuous interpolations using the same Brownian motions $W_t$ and $B_t$:

$$dh_t^{(L)} = b_{L_t}^{(L)} \, dt + \sigma_{L_t}^{(L)} \, dW_t,$$
$$dg_t^{(L)} = c_{\tilde{L}_t}^{(L)} \, dt + D_{\tilde{L}_t}^{(L)} \theta_{\tilde{L}_t}^{(L)} \, dB_t.$$

Here $b_\ell^{(L)} = (b_{\ell,k}^{(L)})_{k=0}^K$ and $c_\ell^{(L)} = (c_{\ell,k}^{(L)})_{k=0}^K$ are vectors given by

$$b_{\ell,k}^{(L)} = -\eta_0 \frac{T}{L} \sum_{i=0}^{k-1} \mathcal{L}'(i, L) g_\ell^{(i),L} \mathbb{E}(\phi(h_{\ell-1}^{(i),L}) \phi(h_{\ell-1}^{(k),L}))$$

$$- \eta_0 \frac{T^2}{L^2} \sum_{i=0}^{k-1} \mathcal{L}'(i, L) g_\ell^{(i),L} \mathbb{E}(\phi(h_{\ell-2}^{(i),L}) \phi(h_{\ell-2}^{(k),L})) \mathbb{E}(\phi'(h_{\ell-1}^{(i),L}) \phi'(h_{\ell-1}^{(k),L})),$$

$$c_{\ell,k}^{(L)} := -\eta_0 \frac{T}{L} \phi'(h_{\ell-1}^{(k),L}) \sum_{i=0}^{k-1} \mathcal{L}'(i, L) \phi(h_{\ell-1}^{(i),L}) \mathbb{E}[g_\ell^{(i),L} g_\ell^{(k),L}]$$

$$- \eta_0 \frac{T^2}{L^2} \phi'(h_{\ell-1}^{(k),L}) \sum_{i=0}^{k-1} \mathcal{L}'(i, L) \phi(h_{\ell-1}^{(i),L}) \mathbb{E}(g_{\ell+1}^{(i),L} g_{\ell+1}^{(k),L}) \mathbb{E}[\phi'(h_\ell^{(i),L}) \phi'(h_\ell^{(k),L})], \quad (22)$$

$D_\ell^{(L)}$ is a diagonal matrix given by

$$D_\ell^{(L)} = \text{diag}\{\phi'(h_{\ell-1}^{(0),L}), \ldots, \phi'(h_{\ell-1}^{(K),L})\},$$

and $\sigma_\ell^{(L)}$ and $\theta_\ell^{(L)}$ are (the Cholesky decomposition) such that

$$\sigma_\ell^{(L)} (\sigma_\ell^{(L)})^\top = \Sigma_\ell^{(L)} := \mathbb{E}[(\phi(h_{\ell-1}^{(0),L}), \ldots, \phi(h_{\ell-1}^{(K),L}))^\top (\phi(h_{\ell-1}^{(0),L}), \ldots, \phi(h_{\ell-1}^{(K),L}))],$$
$$\theta_\ell^{(L)} (\theta_\ell^{(L)})^\top = \Theta_\ell^{(L)} := \mathbb{E}[(g_\ell^{(0),L}, \ldots, g_\ell^{(K),L})^\top (g_\ell^{(0),L}, \ldots, g_\ell^{(K),L})].$$

The following proposition says that the $L^2$ error decays at a rate of $1/L$ for the coupled difference between $h_\ell^L$ (resp. $g_\ell^L$), the finite depth process at discrete step $\ell$, and $h_{\ell T/L}$ (resp. $g_{\ell T/L}$), the corresponding infinite-depth process at time $\ell T/L$.

**Proposition 17.** *Suppose Assumption 1 holds. Then for all $L \geq 1$,*

$$\sup_{\ell=0,1,\ldots,L} \mathbb{E}\|h_\ell^L - h_{\ell T/L}\|^2 \leq C/L, \qquad \sup_{\ell=0,1,\ldots,L} \mathbb{E}\|g_\ell^L - g_{\ell T/L}\|^2 \leq C/L.$$

*Proof of Proposition 17.* We first note that the Lipschitz estimates in Eq. (13), Eq. (18), and Eq. (20) still hold when comparing $\sigma_{s_L}$ and $\sigma_{L_s}^{(L)}$, thanks to Assumption 1. We will again prove by induction. By Propositions 11 and 14, the statement holds for $K = 0$.

Now suppose the statement holds for $K$, namely

$$\sup_{\ell=0,\ldots,L} \sum_{k=0}^K \mathbb{E}[h_\ell^{(k),L} - h_{\ell T/L}^{(k)}]^2 \leq C/L, \qquad \sup_{\ell=0,\ldots,L} \sum_{k=0}^K \mathbb{E}[g_\ell^{(k),L} - g_{\ell T/L}^{(k)}]^2 \leq C/L.$$

We will show that

$$\sup_{\ell=0,\ldots,L} \mathbb{E}[h_\ell^{(K+1),L} - h_{\ell T/L}^{(K+1)}]^2 \leq C/L, \qquad \sup_{\ell=0,\ldots,L} \mathbb{E}[g_\ell^{(K+1),L} - g_{\ell T/L}^{(K+1)}]^2 \leq C/L.$$

Note that

$$\mathbb{E}[h_\ell^{(K+1),L} - h_{\ell T/L}^{(K+1)}]^2 \leq 3\mathbb{E}[h_0^{(K+1),L} - h_0^{(K+1)}]^2 + 3\mathbb{E}\left[\int_0^{\ell T/L} (b_{L_s,K+1}^{(L)} - b_{s,K+1}) \, ds\right]^2$$

$$+ 3\mathbb{E}\left[\int_0^{\ell T/L} (\sigma_{21,L_s}^{(L)} - \sigma_{21,s}, \sigma_{22,L_s}^{(L)} - \sigma_{22,s}) \, dW_s\right]^2.$$

By induction assumption,

$$\mathbb{E}[h_0^{(K+1),L} - h_0^{(K+1)}]^2 \le C/L.$$

By Lemmas 7, 8, and 9 we have

$$\mathbb{E}\left[\int_0^{\ell T/L} (b_{L_s,K+1}^{(L)} - b_{s,K+1})\, ds\right]^2$$

$$\le C\int_0^{\ell T/L} \mathbb{E}|b_{L_s,K+1}^{(L)} - b_{s_L,K+1}|^2\, ds + C\int_0^{\ell T/L} \mathbb{E}|b_{s_L,K+1} - b_{s,K+1}|^2\, ds$$

$$\le \frac{C}{L}\sum_{u=0}^{\ell-1} \mathbb{E}[h_u^{(K+1),L} - h_{uT/L}^{(K+1)}]^2 + \frac{C}{L^2} + \frac{C}{L}.$$

By Lemmas 7, 8, and 9, and Lipschitz property of $\sigma$'s, we have

$$\mathbb{E}\left[\int_0^{\ell T/L} (\sigma_{21,L_s}^{(L)} - \sigma_{21,s}, \sigma_{22,L_s}^{(L)} - \sigma_{22,s})\, dW_s\right]^2$$

$$= \int_0^{\ell T/L} [\|\sigma_{21,L_s}^{(L)} - \sigma_{21,s}\|^2 + \|\sigma_{22,L_s}^{(L)} - \sigma_{22,s}\|^2]\, ds$$

$$\le 2\int_0^{\ell T/L} [\|\sigma_{21,L_s}^{(L)} - \sigma_{21,s_L}\|^2 + \|\sigma_{22,L_s}^{(L)} - \sigma_{22,s_L}\|^2]\, ds$$

$$+ 2\int_0^{\ell T/L} [\|\sigma_{21,s_L} - \sigma_{21,s}\|^2 + \|\sigma_{22,s_L} - \sigma_{22,s}\|^2]\, ds$$

$$\le \frac{C}{L}\sum_{u=0}^{\ell-1} \mathbb{E}[h_u^{(K+1),L} - h_{uT/L}^{(K+1)}]^2 + \frac{C}{L}.$$

Combining the above estimates gives

$$\mathbb{E}[h_\ell^{(K+1),L} - h_{\ell T/L}^{(K+1)}]^2 \le \frac{C}{L}\sum_{u=0}^{\ell-1} \mathbb{E}[h_u^{(K+1),L} - h_{uT/L}^{(K+1)}]^2 + \frac{C}{L}.$$

It then follows from discrete Gronwall's lemma that

$$\mathbb{E}[h_\ell^{(K+1),L} - h_{\ell T/L}^{(K+1)}]^2 \le \frac{C}{L}e^{C\ell/L}.$$

Therefore $\sup_{\ell=0,1,\ldots,L} \mathbb{E}[h_\ell^{(K+1),L} - h_{\ell T/L}^{(K+1)}]^2 \le C/L$. A similar argument applied to Eq. (15) and Eq. (22) gives $\sup_{\ell=0,1,\ldots,L} \mathbb{E}[g_\ell^{(K+1),L} - g_{\ell T/L}^{(K+1)}]^2 \le C/L$ and hence the statement holds for $K+1$.
This completes the proof by induction. $\square$

Theorem 1 then follows from Proposition 17.

# H DISCUSSION ON TWO-LAYER RESIDUAL BLOCKS

Many practical architectures—most notably Transformers—use residual blocks containing two or more internal layers. Depth-$\mu$P was originally motivated and analyzed for one-layer residual blocks and is known empirically to provide stable signal propagation in deep networks (Yang et al., 2024; Bordelon et al., 2024c). However, subsequent studies have observed that hyperparameter transfer fails when the residual block contains more than one internal layer (Dey et al., 2025). Recent work has tried alternative depth scalings such as $1/L$, but the underlying reason why depth-$\mu$P succeeds for one-layer blocks but fails for multi-layer blocks remains unclear.

To understand this phenomenon, we analyze a two-layer residual block

$$\boldsymbol{h}_\ell = \boldsymbol{h}_{\ell-1} + \frac{1}{\sqrt{Ln}}\boldsymbol{W}_{\ell,2}\phi\left(\boldsymbol{x}_\ell\right), \quad \boldsymbol{x}_\ell = \frac{1}{\sqrt{n}}\boldsymbol{W}_{\ell,1}\boldsymbol{h}_{\ell-1}, \qquad \forall \ell \in \{1, 2, \cdots, L\}, \qquad (23)$$

where $\boldsymbol{x}_\ell$ denotes the **internal representation** after the first internal layer. The $1/\sqrt{n}$ factor ensures that $\boldsymbol{x}_\ell$ remains $\Theta(1)$ in the large-width limit. Similar to Section E, backward pass gives:

$$\boldsymbol{g}_{\ell-1} = \boldsymbol{g}_\ell + \frac{1}{\sqrt{L}} \left( \frac{1}{\sqrt{n}} \boldsymbol{W}_{\ell,1} \right)^\top \left[ \left( \frac{1}{\sqrt{n}} \boldsymbol{W}_{\ell,2} \right)^\top \boldsymbol{g}_\ell \odot \boldsymbol{\phi}'(\boldsymbol{x}_\ell) \right]. \tag{24}$$

Then, with a simple recursive argument, we can show that both $\boldsymbol{h}_\ell$ and $\boldsymbol{g}_\ell$ remain stable in the large-depth and large-width limit under the depth-$\mu$P for two-layer residual blocks.

Notably, the forward pass in Eq. (23) and backward pass in Eq. (24) involve the factors

$$\frac{1}{\sqrt{n}} \boldsymbol{W}_{\ell,1} \boldsymbol{h}_{\ell-1}, \quad \frac{1}{\sqrt{n}} \boldsymbol{W}_{\ell,2}^\top \boldsymbol{g}_\ell, \tag{25}$$

both of which are $\Theta(1)$ under width-wise $\mu$P, provided $\boldsymbol{h}_{\ell-1}$ and $\boldsymbol{g}_\ell$ are already $\Theta(1)$ at that layer. These factors are passed through an additional layer, but width-wise $\mu$P again guarantees they stay $\Theta(1)$ as $n \to \infty$. Then, multiplying these $\Theta(1)$ quantities by the depth-$\mu$P factor $1/\sqrt{L}$ shows that the residual update added to both $\boldsymbol{h}_\ell$ and $\boldsymbol{g}_{\ell-1}$ is of order $1/\sqrt{L}$. As a result, the updated activations and gradients remain $\Theta(1)$ as both $n \to \infty$ and $L \to \infty$, meaning that depth-$\mu$P indeed preserves forward and backward stability even for two-layer residual blocks.

### H.1 VANISHING FEATURE LEARNING IN THE FIRST INTERNAL LAYER

The scaled gradient of $f$ w.r.t. the first-layer weight $\boldsymbol{W}_{\ell,1}$ is given by

$$\frac{\sqrt{n}}{\alpha} \frac{\partial f}{\partial \boldsymbol{W}_{\ell,1}} = \frac{1}{\sqrt{Ln}} \left[ \left( \frac{1}{\sqrt{n}} \boldsymbol{W}_{\ell,2} \right)^\top d\boldsymbol{h}_\ell \odot \boldsymbol{\phi}'(\boldsymbol{x}_\ell) \right] \boldsymbol{h}_{\ell-1}^\top. \tag{26}$$

After one-step gradient update with learning rate $\eta_1$, the inner-block representation $\boldsymbol{x}_\ell^+$ becomes:

$$\boldsymbol{x}_\ell^+ = \frac{1}{\sqrt{n}} \left( \boldsymbol{W}_{\ell,1} + \Delta \boldsymbol{W}_{\ell,1} \right) \boldsymbol{h}_{\ell-1}^+$$

$$= \frac{1}{\sqrt{n}} \boldsymbol{W}_{\ell,1} \boldsymbol{h}_{\ell-1}^+ - \frac{\eta_1}{n\sqrt{L}} \mathcal{L}'(f,y) \left[ \left( \frac{1}{\sqrt{n}} \boldsymbol{W}_{\ell,2} \right)^\top \boldsymbol{g}_\ell \odot \boldsymbol{\phi}'(\boldsymbol{x}_\ell) \right] \frac{\langle \boldsymbol{h}_{\ell-1}, \boldsymbol{h}_{\ell-1}^+ \rangle}{n}.$$

Under depth-$\mu$P with $\alpha = 1/\sqrt{n}$ and $\eta_1 = \eta_c n$, each coordinate of $\boldsymbol{x}_\ell^+$ satisfies:

$$\boldsymbol{x}_{\ell,i}^+ = \frac{1}{\sqrt{n}} \left[ \boldsymbol{W}_{\ell,1} \bar{\boldsymbol{h}}_{\ell-1} \right]_i + \mathcal{O} \left( \frac{1}{\sqrt{L}} \right), \quad \forall i \in [n].$$

Thus, the feature update contributed by $\Delta \boldsymbol{W}_{\ell,1}$ decays as $1/\sqrt{L}$ and vanishes in the infinite-depth limit. The first internal layer essentially cannot learn meaningful features: it only passes along information from $\boldsymbol{h}_{\ell-1}$ but does not update its transformation. This causes a structural learning collapse and explains why HP transfer breaks for multi-layer residual blocks.

To counter this, we propose a **depth-aware learning rate**:

$$\eta_1 = \eta_c n \sqrt{L}.$$

which eliminates the $1/\sqrt{L}$ suppression and restores effective learning in the first layer. We empirically quantify the collapse by comparing the true $\boldsymbol{x}_\ell^{(k)}$ with a "frozen-weights" approximation

$$\widetilde{\boldsymbol{x}}_\ell^{(k)} = \frac{1}{\sqrt{n}} \boldsymbol{W}_\ell^{(0)} \boldsymbol{h}_{\ell-1}^{(k)}. \tag{27}$$

When depth increases, $\boldsymbol{x}_\ell^{(k)} \approx \widetilde{\boldsymbol{x}}_\ell^{(k)}$, confirming that the first layer stops learning; the depth-aware learning rate restores nontrivial updates and HP transfer (Figure 5).

## H.2 Internal-Dynamics Learning in the Second Layer

A subtle yet important point is that the vanishing effect in the first internal layer can be easily overlooked if one only monitors weight updates. This is because the gradients of the second-layer weights $\boldsymbol{W}_{\ell,2}$ exhibit the **same** $1/\sqrt{L}$ vanishing factor as those for the first-layer weight $\boldsymbol{W}_{\ell,1}$. However, from the perspective of NFD, the learning roles of these two layers are fundamentally different:

- The first internal layer $\boldsymbol{W}_{\ell,1}$ controls the **internal representation** $\boldsymbol{x}_\ell$. Vanishing updates here directly imply that feature learning collapses.
- The second internal layer $\boldsymbol{W}_{\ell,2}$ controls the **internal residual-stream dynamics of $\boldsymbol{h}_\ell$**, which correspond to the drift and diffusion terms of the NFD limit.

From this view, the meaningful quantity for analyzing learning in deeper layers is not the size of the weight update itself, but the induced change in the residual-stream increment $\boldsymbol{h}_\ell - \boldsymbol{h}_{\ell-1}$. Even if the raw weight updates $\Delta \boldsymbol{W}_{\ell,2}$ shrink with depth, its effect on $\boldsymbol{h}_\ell$ can remain stable and non-vanishing, unlike the update to $\boldsymbol{x}_\ell$.

Therefore, correctly diagnosing the collapse of feature learning requires monitoring the updates of the **internal representation $\boldsymbol{x}_\ell$** and the **internal residual-stream dynamics $\boldsymbol{h}_\ell - \boldsymbol{h}_{\ell-1}$**, rather than relying solely on weight-space measurements. The rest of this subsection demonstrates this idea.

The scaled gradient for the second-layer weight $\boldsymbol{W}_{\ell,2}$ is given by

$$\frac{\sqrt{n}}{\alpha} \frac{\partial f}{\partial \boldsymbol{W}_{\ell,2}} = \frac{1}{\sqrt{Ln}} \boldsymbol{g}_\ell \phi(\boldsymbol{x}_\ell)^\top. \tag{28}$$

After one-step gradient update with learning rate $\eta_2$, the residual-stream dynamics of $\boldsymbol{h}_\ell^+$ becomes

$$\boldsymbol{h}_\ell^+ = \boldsymbol{h}_{\ell-1}^+ + \frac{1}{\sqrt{Ln}} (\boldsymbol{W}_\ell + \Delta \boldsymbol{W}_\ell) \phi(\boldsymbol{x}_\ell^+) \tag{29}$$

$$= \boldsymbol{h}_{\ell-1}^+ + \frac{1}{\sqrt{Ln}} \boldsymbol{W}_\ell \phi(\boldsymbol{x}_\ell^+) - \frac{\eta_2}{nL} \mathcal{L}'(f,y) \boldsymbol{g}_\ell \frac{\langle \phi(\boldsymbol{x}_\ell), \phi(\boldsymbol{x}_\ell^+) \rangle}{n}. \tag{30}$$

Under depth-$\mu$P with $\alpha = 1/\sqrt{n}$ and $\eta_2 = \eta_c n$, the internal dynamics of $\boldsymbol{h}_\ell$ satisfies

$$\boldsymbol{h}_\ell^+ - \boldsymbol{h}_{\ell-1}^+ = \sqrt{\frac{\tau}{n}} \boldsymbol{W}_\ell \phi(\boldsymbol{x}_\ell^+) - \tau \eta_c \mathcal{L}'(f,y) \boldsymbol{g}_\ell \frac{\langle \phi(\boldsymbol{x}_\ell), \phi(\boldsymbol{x}_\ell^+) \rangle}{n}, \tag{31}$$

where $\tau = 1/L$. Unlike the update for $\boldsymbol{x}_\ell$, the diffusion and drift terms in the update of $\boldsymbol{h}_\ell$ scale as $\sqrt{\tau}$ and $\tau$, exactly matching the drift/diffusion terms of the stochastic nature of the NFD. Therefore, no vanishing occurs for the second layer: its contribution to the internal dynamics of $\boldsymbol{h}_\ell$ remains meaningful even as $L \to \infty$. Hence, in our depth-aware learning rate for $\boldsymbol{W}_{\ell,2}$ remains:

$$\eta_2 = \eta_c n. \tag{32}$$

To verify this, we measure the internal residual-stream dynamics updates via:

$$\partial_t \boldsymbol{h}_{t_\ell}^{(k)} \simeq \frac{\boldsymbol{h}_\ell^{(k)} - \boldsymbol{h}_{\ell-1}^{(k)}}{\tau} = \sqrt{\frac{L}{n}} \boldsymbol{W}_{\ell,2}^{(k)} \phi\left(\boldsymbol{x}_\ell^{(k)}\right) \tag{33}$$

$$\partial_t \widetilde{\boldsymbol{h}}_{t_\ell}^{(k)} \simeq \sqrt{\frac{L}{n}} \boldsymbol{W}_{\ell,2}^{(0)} \phi\left(\boldsymbol{x}_\ell^{(k)}\right) \tag{34}$$

Empirically, unlike the vanishing that occurred in $\boldsymbol{x}_\ell$, $d\boldsymbol{h}^{(k)}/dt$ remains stable and non-vanishing across depths, consistent with theory (Figure 5)..

## I Additional Experiments

In this section, we provide supplementary experiments that complement the results reported in the main paper.

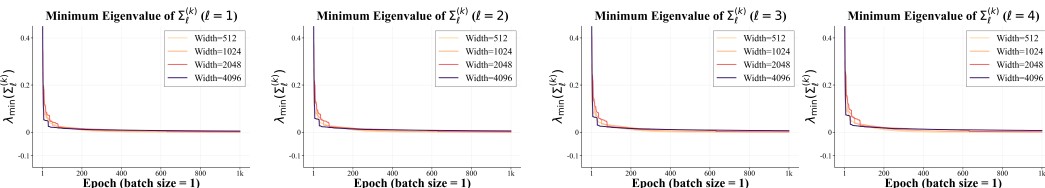

Figure 6: **Minimum eigenvalues of the covariance matrices during training.** We evaluate ResNets on CIFAR-10 using online SGD (learning rate 0.1, batch size 1) across 5 seeds, with 4 hidden layers and widths ranging from 512 to 4096. The minimum eigenvalues of $\mathbf{\Sigma}_t^{(k)}$ and $\mathbf{\Theta}_t^{(k)}$ remain strictly positive across layers, validating Assumption 1. Although the eigenvalues decrease over training, they grow with network width; insufficiently wide networks (e.g., width 512) yield eigenvalues very close to zero.

### I.1 MINIMUM-EIGENVALUE BEHAVIOR AND VALIDATION OF ASSUMPTION 1

The convergence proof to NFD in Theorem 1 relies on Assumption 1 that the smallest eigenvalues of the covariance matrices $\{\mathbf{\Sigma}_\ell^{(k)}\}_\ell$ and $\{\mathbf{\Theta}_\ell^{(k)}\}_\ell$ are uniformly bounded away from zero. To examine this assumption, we train a four-layer ResNet under depth-$\mu$P and track the spectra of these matrices. As shown in Figure 6, the smallest eigenvalues of $\{\mathbf{\Sigma}_\ell^{(k)}\}_\ell$ remain strictly positive across layers, although they gradually decrease during training as the covariance matrices expand in dimension. Moreover, the smallest eigenvalues increase with network width, suggesting that wider networks preserve more diverse feature representations and may generalize better to downstream tasks.

### I.2 GIA RESTORATION AND DYNAMICS ALIGNMENT

In this subsection, we empirically evaluate the restoration of GIA predicted by Theorem 1 and Corollary 1. For each architecture (vanilla DNN, $\mu$P-ResNet, and depth-$\mu$P ResNet), we compare *standard* training—where the backward pass reuses the forward weights—with a *decoupled* variant in which the backward pass uses an i.i.d. copy of the forward weights. Figure 4 shows the evolution of training and test loss for width 128 across increasing depths. As depth grows, DNNs exhibit vanishing gradients and the standard and decoupled trajectories remain misaligned; $\mu$P-ResNets also fail to align, overfit at larger depths, and even encounter exploding gradients and NaN values once the depth exceeds 18. In contrast, depth-$\mu$P achieves consistent improvements in both training and test performance, and the two trajectories converge toward each other, confirming the restoration of GIA in the infinite-depth limit. To assess robustness across widths, Figure 7 reports the corresponding width-256 results, which closely mirror the trends in Figure 4: larger width mitigates some instability and overfitting in DNNs and $\mu$P-ResNets but does not resolve their underlying pathologies, while further sharpening the trajectory alignment and performance gains observed under depth-$\mu$P.

## J RELATED WORK

**Empirical neural scaling laws.** Large empirical studies have shown that neural networks often exhibit simple *power-law trends*: as model size, dataset size, or compute increases, performance improves predictably (Kaplan et al., 2020; Henighan et al., 2020; Rosenfeld et al., 2020; Hoffmann et al., 2022). These observations have led to practical "compute-optimal" training rules and guidelines for allocating model and data size. Subsequent work extended these findings across modalities, including vision and diffusion, while also identifying cases where scaling laws break down due to unstable optimization, low-quality data, or insufficient training (Zhai et al., 2022; Cherti et al., 2023; Muennighoff et al., 2023). Although highly influential, these studies remain primarily empirical and do not provide a *mechanistic* explanation for when scaling laws succeed, saturate, or fail.

**Theoretical studies on scaling laws.** Most theoretical work focuses on predicting scaling exponents using random-feature and fixed-kernel models, where representations are held constant during training. Bahri et al. (2024) showed that power-law test-loss behavior in teacher–student and random-feature models arises from the power-law decay of kernel spectra, revealing regimes deter-

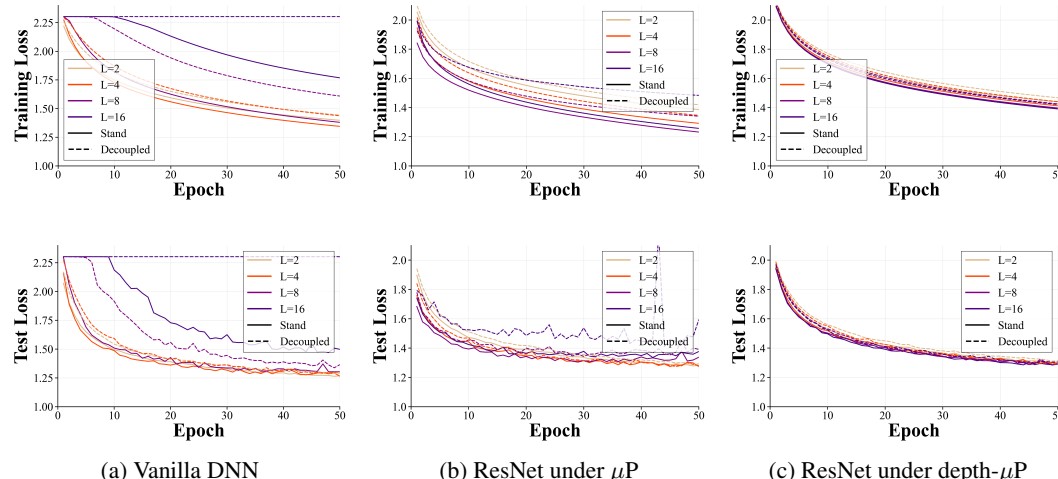

(a) Vanilla DNN         (b) ResNet under $\mu$P         (c) ResNet under depth-$\mu$P

Figure 7: **Empirical evaluation of GIA restoration at width 256.** We repeat the experiment of Figure 4 using width 256, again comparing *standard* training (shared forward/backward weights) with a *decoupled* setup that uses an i.i.d. copy of the forward weights for the backward pass. Relative to width 128, increasing the width reduces instability in both **(a)** vanilla DNNs and **(b)** $\mu$P-ResNets, but the two trajectories still fail to align and the finite-depth pathologies remain. **(c)** Under depth-$\mu$P, larger width further smooths the dynamics and strengthens the agreement between standard and decoupled trajectories, reinforcing the empirical restoration of GIA predicted by our theory.

mined by whether data or model size is the bottleneck. Maloney et al. (2022) refined this picture using a solvable RF model with finite spectral support, demonstrating that scaling laws plateau once all modes supplied by the kernel or data are resolved. Simon et al. (2024) analyzed kernel ridge regression and proved that increasing data or model size always improves generalization when regularization is tuned appropriately, while spectrally heavy-tailed tasks require obligatory overfitting. Complementing these static analyses, Bordelon et al. (2024a) introduced a dynamical RF model showing that gradient flow resolves kernel eigenmodes over time, yielding a distinct *training-time* scaling exponent. Paquette et al. (2024) further established the 4+3 compute-optimal phases by analyzing a power-law RF model via a Volterra equation, providing a theoretical basis for the Chinchilla scaling law. Together, these works illuminate scaling behavior in the fixed-kernel regime, where representations do not evolve. However, they cannot explain when scaling laws hold or fail once feature learning becomes significant.

**Infinite-width and kernel regimes.** The infinite-width Neural Tangent Kernel (NTK) (Jacot et al., 2018b) and related linearization results (Lee et al., 2019; Arora et al., 2019) recast neural network training as kernel regression with a fixed NTK. This provides strong convergence guarantees (Allen-Zhu et al., 2019; Du et al., 2019; Zou et al., 2020) and explains how wide models generalize despite interpolation (Arora et al., 2019). However, NTK models operate in the lazy-training regime (Woodworth et al., 2020; Chizat & Bach, 2019), where features remain frozen and cannot capture representation learning or dynamic scaling behavior.

**Mean-field analysis and its depth limitations.** Mean-field (MF) parameterizations (Mei et al., 2018; Chizat & Bach, 2018; Sirignano & Spiliopoulos, 2020) allow nonlinear feature evolution as width grows and yield global convergence guarantees for two- and three-layer networks (Chen et al., 2020; Nitanda et al., 2022; Pham & Nguyen, 2021). However, for deeper architectures, standard MF scalings suffer from degeneracies: signals and gradients collapse to zero (Fang et al., 2021; Nguyen & Pham, 2023), preventing MF theory from describing the rich feature learning observed in modern deep networks. While MF identifies which scalings preserve feature dynamics at moderate depth, it does not provide a training-time theory for deep architectures.

**Tensor Programs and $\mu$P.** Tensor Programs (TP) provide a unified formalism for analyzing wide networks and their training dynamics (Yang, 2019; 2020b;a; Yang & Hu, 2021; Yang et al., 2021).

A key result is the TP Master Theorem, which ensures law-of-large-numbers behavior for wide networks. Yang & Hu (2021) established the *Dynamical Dichotomy*: any stable width scaling lies either in the kernel regime (features fixed) or the feature-learning (FL) regime (features evolve), but not both. Within this classification, $\mu$P (Yang & Hu, 2021; Yang et al., 2021) maximizes feature evolution and enables hyperparameter transfer across widths. Recent TP extensions incorporate depth scaling (Yang et al., 2024). However, these works face critical challenges when residual blocks have more than one layer. This indicates a lack of rigorous characterization of feature learning dynamics in the large-depth limit.

**Infinite-depth analyses.** Prior work on infinite-depth networks focused mainly on stability at initialization. For MLPs, signal-propagation analyses identified "edge-of-chaos" conditions (Poole et al., 2016; Schoenholz et al., 2017). Later work emphasized the noncommutativity of width and depth limits (Li et al., 2022). For ResNets, scaling the residual branch by $1/\sqrt{L}$ restores commutativity and yields stable propagation (Hayou & Yang, 2023; Marion et al., 2025). Extensions into training further showed that this scaling enables reliable hyperparameter transfer (Bordelon et al., 2024c) and that feature learning persists even at infinite depth (Yang et al., 2024). Yet these analyses remain focused on identifying stable depth scalings, without modeling the complete training-time coupling of features and gradients.

**Summary.** Empirical studies have revealed striking power-law regularities, NTK and mean-field analyses have clarified width-wise limits, Tensor Programs and $\mu$P have established principled feature-learning parameterizations and dynamics in the infinite-width limit, and infinite-depth works have begun probing stability at initialization. However, none provides a unified account of *training-time* feature and gradient dynamics in the infinite-depth limit. Our work fills this gap by introducing Neural Feature Dynamics (NFD), which characterizes the coupled forward-backward stochastic dynamics of deep ResNets in the joint limit, offering a tractable and principled mechanism for understanding when scaling laws succeed, when they fail, and how HP transfer can be restored.

## K  THE USE OF LARGE LANGUAGE MODELS

We used large language models (LLMs) as a writing assistant to polish the presentation of our paper and to help identify related work during the literature review stage. The tool was not involved in designing experiments, deriving theoretical results, or interpreting findings; it was used strictly for language refinement and for improving the clarity and accessibility of our writing.

