# OpenReview forum: "Understanding Scaling Laws in Deep Neural Networks via Feature Learning Dynamics"
_ICLR.cc/2026/Conference — ICLR 2026 Conference Desk Rejected Submission_

### Official Review · Reviewer_8saH · 2025-10-28

**Soundness:** 3
**Presentation:** 3
**Contribution:** 3
**Rating:** 8
**Confidence:** 3

**Summary:**

The authors introduce Neural Feature Dynamics (NFD), a theoretical framework for understanding scaling laws in deep neural networks through the lens of feature learning in the joint infinite-width and infinite-depth limit. The paper explores ResNets under depth-adapted μP and shows that training dynamics converge to a coupled forward-backward stochastic differential equation (SDE) system. The framework explains when scaling succeeds (convergence to the limiting SDE holds), when it fails (convergence breaks down), and the emergence of diminishing returns (as approximation error shrinks near the limit).

Overall, the paper makes a solid contribution to the community.

**Strengths:**

* The paper introduces a valuable theoretical model for understanding width/depth scaling in relevant models.
* The framework is introduced clearly, making the paper relatively accessible to a general audience.
*  The mathematics appears comprehensive, particularly when including the appendix.
* The models chosen for the analysis are current and of importance to ML practitioners. This also includes testing a limited number of optimizers.
* The identification of a capacity ceiling is an interesting result.

**Weaknesses:**

* As with any theoretical papers, not all architectures of interest to the community are represented, leaving possible gaps in the scope of the theory.
* The paper is very dense. While, as stated above, the mathematics is introduced clearly and with sufficient explanation, a lot is left to the appendix.
* In the introduction, the authors emphasize the feature learning regime falling outside the scope of NTK theory. While I agree from a dynamics perspective, the (empirical) NTK can be computed on networks of any size and in any regime, and is still an interesting object to study. Perhaps this can be made clearer in those early statements.

**Questions:**

* The outlook of the paper states that NFD can be used as a base for future learning dynamics studies. What insight do the authors expect to gain by using the framework to study learning dynamics? Could the NFD be expected to provide more detailed information about how models are learning, the kinds of features being learned, or the phases?
* If so, could these insights then be applied to finite networks on a more practical level?

---

> ### Author Response · Authors · 2025-11-24
>
> We thank the reviewer very much for the thoughtful and positive assessment of our work, as well as for highlighting both the strengths and the constructive suggestions. We address the clarifications and questions below.
>
> **(1) Scope of architectures** We appreciate the reviewer’s point that our theoretical analysis currently focuses on simple ResNets. However, we expect NFD can be naturally extends to other architectures such as Transformers and optimizers such as Adam. We will consider them as our future works.
>
> **(2) Density and appendix dependency** We thank the reviewer for this comment. We have added guiding text to revised manuscript to provide a higher-level intuition. This maintains mathematical rigor while improving readability and accessibility.
>
> **(3) NTK clarification in the introduction** We appreciate this nuanced point. In the introduction, our goal is to distinguish dynamics in the NTK regime (features remain fixed) from the feature-learning regime studied in NFD. We agree that the empirical NTK is still interesting and useful topic. We have clarified this in the revised text.
>
> **Q1. What insight do the authors expect to gain by using NFD to study learning dynamics?** We believe NFD provides a rigorous and tractable description of joint feature–gradient evolution, thanks to two key properties: the emergence of **independent** Brownian motions and the restoration of the **gradient-independence assumption (GIA)** in the infinite-depth limit. These make the feature learning dynamics analytically manageable.
>
> Beyond tractability, NFD reveals a new **structural separation** inside two-layer residual blocks (Section 5): first layers govern internal representation, the second layers govern residual-stream dynamics. This leads us to identify the depth-induced collapse of first-layer feature learning and propose a simple fix that restores feature updates, hyperparameter transfer across depth, and improved performance (Fig 5).
>
> Our current analysis focuses on a finite but fixed number of SGD steps, and it would be very interesting to study the **equilibrium (long-time) behavior** of the NFD system, potentially connecting it to mean-field or optimal-transport dynamics for understanding long-term representation evolution. Finally, because the kernel and feature-learning regimes correspond to different choices of the μP scaling parameter $\alpha$, NFD naturally suggests investigating **phase transitions** between the two regimes by interpolating $\alpha$, an exciting direction for future work.
>
> **Q2. Can these insights apply to finite networks in practice?** We believe so. NFD comes with explicit finite-width/depth convergence rates (Propositions 1, 6, 17), and we empirically (Fig 3) verify that these rates are tight even for moderate widths (e.g., 128–256) and depths (32–64). This suggests that NFD’s predictions are already accurate for commonly used architectures.
>
> Furthermore, our Section 5 demonstrates a concrete case where NFD directly guides a practical correction (depth-aware LR scaling) that improves performance in finite networks (Fig 5).
>
> We sincerely thank Reviewer 8saH for their careful reading, constructive suggestions, and positive evaluation. We have incorporated the clarifications above into the revised manuscript and hope the updates further strengthen the paper.

---

### Official Review · Reviewer_5S2H · 2025-10-30

**Soundness:** 3
**Presentation:** 2
**Contribution:** 2
**Rating:** 2
**Confidence:** 3

**Summary:**

This paper studies the scaling limits and convergence rates associated with the joint infinite width and infinite depth limit of residual neural networks. The authors provide a description of the training dynamics of networks in this joint limit. They provide experiments demonstrating that convergence can be achieved at reasonable widths and depths. They provide some analytical results about the hilbert space of the NNGP kernel at initialization and also the evolution of the preactivation statistics through training.

**Strengths:**

This paper studies an important question of how neural networks converge to their large width and depth limits. They provide many interesting experiments including comparisons of preactivation and postactivation design and the evolution of the minimum kernel eigenvalue throughout training. They also prove that increasing the effective layer time $T$ increases the size of the RKHS of the associated kernel at initialization.

**Weaknesses:**

**Novelty**:
I am concerned about the level of novelty of the primary results of this work. For example, the evolution equations describing the infinite width and depth limit (proposition 8 in this paper) can be found in earlier works (for instance [this work](https://arxiv.org/pdf/2309.16620), Result 1 and Appendix E). That paper also provides finite width and depth error analysis (Appendix H and I of [this work](https://arxiv.org/pdf/2309.16620) ).

In the Appendix of this submission, the authors claim that prior studies "remain primarily focused on identifying scaling rules, offering limited insight into the training-time dynamics of coupled feature and gradient evolution." I am not sure that this is fair to the prior work that I linked to. A lot of work has gone into analyzing the evolution equations of infinite width networks during training (see below). In my opinion, the authors should clarify in what ways their analysis provides new results relative to prior works.

In the conclusion, the authors state " Our main result (Theorem 1) established that, in the infinite-width-depth limit, training trajectories converge to Neural Feature Dynamics (NFD), a coupled forward-backward stochastic system that goes beyond the NTK regime,
with explicit convergence rates and commutativity at initialization." In my understanding, these findings were present in [prior work](https://arxiv.org/pdf/2309.16620) on this topic.

**Regime to study scaling laws**:

I am also not sure that the convergence rates provided from standard error analysis are descriptive of realistic neural scaling laws. It is true that for fixed data and fixed iteration count (compared to width, depth etc), the error rates should scale as $\sim 1/n + 1/L$ (or $1/L$, see question below) as $n,L \to \infty$. However, the **task dependent scaling laws** with exponents less than one, that are observed after training a finite model on a large quantity of data, cannot be explained by this standard error analysis (see discussion [on page 2 here](https://arxiv.org/abs/2402.01092) ). Prior works usually study convergence rates of limited rank empirical kernels in the underparameterized regime to generate these alternative scaling exponents.

**Incomplete references to prior works**:

The authors do not discuss some highly relevant areas of prior work

1. Theoretical works on neural scaling laws: [Bahri et al 2021](https://arxiv.org/abs/2102.06701), [Maloney & Roberts 2022](https://arxiv.org/abs/2210.16859), [Simon et al 2023](https://arxiv.org/abs/2311.14646), [Bordelon, Atanasov, Pehlevan 2024](https://arxiv.org/pdf/2402.01092), [Paquette et al 2024](https://arxiv.org/abs/2405.15074).
2. Theory describing evolution of infinite width networks: [Bordelon & Pehlevan 2022](https://arxiv.org/abs/2205.09653) (predicting infinite width dynamics across varying levels of richness), [Bordelon, Chaudhry, Pehlevan 2024](https://arxiv.org/abs/2405.15712) (infinite width and depth limits of transformers), [Chizat et al 2022](https://arxiv.org/abs/2211.16980) theory for deep linear networks
3. Other works on infinite depth limits: [Hayou 2024](http://www.jmlr.org/papers/v25/23-1163.html) for convergence rates. [CompleteP](https://arxiv.org/abs/2505.01618) ($1/L$ depth scaling)

**Questions:**

1. **Depth Convergence Rate** Could the depth convergence rate derived in this work be too pessimistic? The authors of this submission derive a square error convergence rate of $1/L$ by considering convergence of residual variables $h$ to the SDE for the limiting residual stream. I think if the authors are concerned about convergence of the *neural network predictions*, the error rate should be $1/L^2$ like in [Bordelon et al 2023](https://arxiv.org/abs/2309.16620) and [Hayou 2024](http://www.jmlr.org/papers/v25/23-1163.html). The reason for this is that while the preactivation variables follow an SDE with $1/L$ square error rate, the **kernels** which govern the behavior of the network follow an ODE in the limit which has a more favorable error rate.
2. **Data / Batch / Steps Dependence**: How many data points are used in Figure 4 where the authors show convergence of the (non-zero) minimum kernel eigenvalue. At some point if the dataset is sufficiently large compared to width, should this eigenvalue decrease to zero? Providing more details about these figures could be very helpful in their interpretation.
3. The convergence plots in

---

> ### Author Response · Authors · 2025-11-24
>
> We thank the reviewer for the detailed comments, valuable references, and thoughtful questions that helped us improve the clarity and positioning of our manuscript. Below we address each concern and explain how the revision resolves the issues.
>
> **(1) Weakness: Novelty**: We agree with the reviewer that prior work [1] has reported SDE limits for deep ResNet. However, we have identified several key differences between our contributions and theirs, and we have revised the manuscript to emphasize these distinctions clearly:
> - **Rigorous framework vs. DMFT heuristics.** NFD is a mathematically rigorous framework capturing training-time feature learning in the joint infinite-width and infinite-depth limit. In contrast, [1] relies on a DMFT-style heuristic derivation and does not state explicit mathematical assumptions for their limit, neither in the main paper nor the appendix. As the authors themselves acknowledge in their conclusion: *"our results are derived at the level of rigor of physics, rather than formal proof."* By comparison, our analysis is built on clearly stated assumptions (Assumption 1), including uniform SPD conditions on covariance matrices, which we also empirically validate in Fig. 4.
> - **Independence of Brownian motions.** Because of the different levels of rigor, NFD rigorously proves that the forward and backward Brownian motions are independent. In contrast, [1] requires explicit response functions to account for forward–backward interactions.
> - **Restoration of gradient-independence assumption (GIA) during training.** The independent Brownian motions leads to a new result: GIA becomes provably valid during training in the infinite-depth limit (Corollary 1). This restoration has not been established in prior work. In fact, earlier analyses explicitly show that GIA fails at finite depth [3]. The restoration of GIA identifies NFD as a tractable framework for theoretical analysis on feature learning.
> - **Structural interpretation and depth-aware correction (Section 5).** NFD yields a new structural interpretation in two-layer residual blocks: the first layer controls internal representation, and the second layer governs residual-stream dynamics. This insight allows us to identify a **depth-induced collapse** of first-layer feature learning and motivates a **depth-aware learning-rate correction**. We show experimentally (Fig. 5) that this correction restores feature learning, enables HP transfer under depth-$\mu$P for two-layer residual blocks, and improves performance.
>
> **(2) Weakness: Regime for Studying Scaling Laws** We appreciate the reviewer’s detailed comments and insightful references on scaling-law theory. After carefully reviewing these works, we find that most theoretical studies focus on predicting scaling exponents using random-feature or kernel models. Our goals differ fundamentally:
> - Prior theories use solvable models that assume fixed representations and therefore cannot explain **when and why scaling laws succeed or fail.**
> - Our work studies the **training dynamics** of ResNets as both width and depth tend to infinity, providing conditions under which the infinite-depth limit yields a well-posed nonlinear feature-learning dynamics.
>
> Thus, the convergence rates in Propositions 1, 6, and 17 are not scaling exponents but rather finite-depth and -width convergence rates describing how finite ResNets approximate the NFD limit. However, we believe these results provide new insights into scaling behaviors:
> - prior works barely do not model how feature learning affects scaling laws,
> - NFD provides a tractable feature-learning framework with provable GIA,
> - and this enables explanations of scaling-law success or failure from the perspective of whether the network parameterization yields a stable limiting dynamics.
>
> We have substantially expanded the Related Work section to incorporate these distinctions.
>
> **(3) Weakness: Incomplete References**. We again thank the reviewer for pointing out additional relevant works.
> We have added these citations and expanded the related-work section with a dedicated discussion of prior theoretical scaling-law studies, positioning our contributions precisely within that landscape.

---

> > ### Author Response · Authors · 2025-11-24
> >
> > **(4) Question 1: Depth Convergence Rate**: We agree with the reviewer that [1-2] obtain stronger rates by analyzing kernels and deriving ODE limits. In contrast, our convergence rate is derived for feature vectors $h_\ell$ under an SDE perspective. We believe these two views are complementary rather than competing:
> > - The ODE viewpoint provides a *macroscopic* description of feature propagation.
> > - The SDE/NFD viewpoint provides a *microscopic* description of feature evolution during training.
> >
> > Both perspectives are important, and we now emphasize this complementarity in the revised manuscript.
> >
> > **(5) Question 2: Data / Batch / Step Dependence** In Fig. 4, we train ResNets with batch size 1 for 1000 epochs, so each epoch introduces a fresh data point. As the reviewer notes, with more data the minimum eigenvalues decrease, but they increase with width. We added width=512 as an additional case: as expected, 1000−512=488 eigenvalues are numerically zero (~1e−16), while the remaining eigenvalues are at least ~1e−6.
> >
> > Following the reviewer’s suggestion, we have also added detailed experimental settings (depth, width, batch size, learning rate, epochs, dataset) to all figure captions.
> >
> > **(6) Question 3: Incomplete question**. We noticed that the reviewer’s third question appears truncated. If the reviewer provides the full question, we would be happy to address it explicitly.
> >
> > We sincerely thank Reviewer 5S2H for the constructive suggestions and for pointing us to valuable references. We hope that these clarifications and revisions help address the reviewer’s concerns regarding novelty and the correct positioning of our work within the scaling-laws literature, and allow for a fair reassessment of the contributions.
> >
> > [1] Bordelon et al., Depthwise Hyperparameter Transfer in Residual Networks: Dynamics and Scaling Limit, ICLR 2024.
> >
> > [2] Hayou & Yang, Width and Depth Limits Commute in Residual Networks, ICML 2023.
> >
> > [3] Yang & Hu. Tensor Programs IV: Feature Learning in Infinite-Width Neural Networks, ICML 2021

---

> ### Comment · Reviewer_5S2H · 2025-11-24
> **Additional Question**
>
> This reviewer appreciates the detailed responses from the authors and their updates to the paper. Before I consider revising my score, I have a few more questions..
>
> First, I am still skeptical of the claim that the forward and backward stochastic processes retain gradient independence (or rather, I think a more careful definition is possibly needed at infinite depth). To see this, I would like to understand how the authors claim that the contributions from these interactions vanish in the limit.
> Following the notation of [this paper](https://arxiv.org/pdf/2309.16620), the forward variables $h$ and backward variables $g$ have the following recursions (take for concreteness a linear network and neglecting time or sample indices)
>
> $$h^\ell  = \frac{1}{\sqrt L} \sum_{k<\ell} u^k + \frac{1}{\sqrt L} \sum_{k < \ell} A^{k} g^{k} + \frac{1}{L} \sum_{k<\ell} \Phi^k g^k$$
> $$g^\ell  = \frac{1}{\sqrt L} \sum_{k<\ell} r^k + \frac{1}{\sqrt L} \sum_{k < \ell} B^{k} h^{k} + \frac{1}{L} \sum_{k<\ell} G^k h^k$$
>
> where $u^\ell, r^\ell$ are independent Gaussian processes. The response variables $A,B$ are defined as $A^\ell = \frac{\partial h^\ell}{\partial r^\ell}$ and $B^\ell = \frac{\partial g^\ell}{\partial u^\ell}$. First, one can show that each of the $A^\ell, B^\ell \sim \mathcal{O}(L^{-1/2})$ which arise from correlations between $\frac{1}{\sqrt N} W(0) \phi$ and $\frac{1}{\sqrt N} W(0)^\top g$ so that they **individually vanish** as $L \to\infty$. However, they **collectively add up to an effect that alters the dynamics** when you sum over all layers. To see this, note that if $A^k \sim \mathcal O(L^{-1/2})$, note that the term contributes equally to the $\frac{1}{L}\sum_k \Phi^\ell g^k$ update equation.
>
> A version of these DMFT equations have also been [simulated in deep linear networks](https://arxiv.org/pdf/2502.02531) and closely track experiments across many depths.
>
> Do the authors have an intuitive argument/reason for why they believe these interaction terms actually don't contribute as $L \to \infty$?
>
>
> Second, I was wondering (in my incomplete submitted questions) if the authors tried plotting convergence rates in $L$ of the function (outputs of the network) or the feature kernels. Do they see that these converge at a faster rate than the individual preactivations?

---

> ### Author Response · Authors · 2025-11-25
>
> We thank the reviewer for the follow-up question and for the careful examination of both the interaction terms in the large-depth limit and how different network quantities may exhibit different depth-convergence rates. We address the two points below.
>
> *(1) Why do the interaction terms vanish as $L \to \infty$?* We agree with the reviewer that if the upper bound $A^{\ell}\sim \mathcal{O}(L^{-1/2})$ is tight, then one might expect these interaction terms to contribute at the same order as $\frac{1}{L}\sum_k \Phi^{k}g^{k}$ instead of vanishing. However, the key issue is that the bound $A^{\ell}\sim \mathcal{O}(L^{-1/2})$ is an upper bound obtained from DMFT-style heuristic derivations, which are physics-inspired rather than mathematically rigorous. This upper bound is **not tight**. Under our rigorous mathematical analysis, the actual magnitude of the response terms is **strictly smaller**, and this is the main reason causing the interaction terms to vanish.
>
> The DMFT heuristic indeed provides only a loose upper bound. In [1], after simplifying their equation (44), one obtains the recursive form
> $$
> A^{\ell}\lesssim \frac{1}{\sqrt{L}}\sum_{k=1}^{\ell} A^{k} \frac{\delta g^{k+1}}{\delta r^{\ell}} + \frac{1}{L}\sum_{k=0}^{\ell-1} \frac{\delta g^{k}}{\delta r^{\ell}} \lesssim \frac{1}{L}\sum_{k=1}^{\ell} A^{k}  + \frac{\ell}{L^{-3/2}}
> $$
> where the second inequality uses their own stated bound $\frac{\delta g^{\ell}}{\delta r^{k}}\sim\mathcal{O}(L^{-1/2})$ from equation (42), without exploiting any layer-wise decoupling or conditional independence across layers as in the Tensor Program analysis [2]. Solving this recursion yields their heuristic scaling $A^{\ell}\sim \mathcal{O}(L^{-1/2})$.
>
> In contrast, using the Tensor Program Master Theorem [2], we prove in Proposition 8 that the actual response coefficients satisfy $A^{\ell}\sim \mathcal{O} (L^{-3/2})$, since $\tau^2=L^{-2}$, which is significantly smaller than the DMFT upper bound. Consequently, the interaction contribution
> $$
> \frac{1}{\sqrt{L}}\sum_{k < \ell} A^{k} g^{k}\lesssim \frac{1}{\sqrt{L}}\sum_{k < \ell} \mathcal{O}(L^{-3/2})=\mathcal{O}(L^{-1})\rightarrow 0,
> $$
> and therefore vanishes in the infinite-depth limit. This sharper $L^{-3/2}$ scaling explains why interaction terms do not accumulate, and it is precisely what produces independent forward/backward Brownian motions in the NFD limit and the **restoration of GIA** in the infinite-depth regime (Corollary 1).
>
> We thank the reviewer for pointing to [3]. While [3] uses DMFT to study training dynamics of deep linear and residual networks, we note:
> - Only Section 3 of [3] analyzes ResNets; the rest are linear networks.
> - Figures 4-5 from Section 3 of [3] do not examine whether the interaction terms vanish or persist at large depth.
> - [3] focuses on **one-layer** residual blocks for hyperparameter transfer, whereas [4] shows that depth-$\mu$P **fails** for two-layer blocks.
>
> Our NFD explains this failure via feature-learning collapse in the first layer of a two-layer block, and provides a simple **depth-aware learning-rate correction** that restores feature learning in the first layer, depth-wise HP transfer, and overall improved performance (Section 5, Fig. 5).
>
> *(2) Whether feature kernels and network outputs converge faster than individual preactivations?* While our analysis and experiments focus on microscopic feature-learning dynamics, we agree with the reviewer that quantities such as feature kernels and outputs should indeed converge faster. This is because these objects are defined through **expectations** over features $h$ and gradients $g$, so their convergence rates are governed by the convergence of their *mean* rather than the variance of the individual diffusion fluctuations. Although we did not run separate experiments on these macroscopic quantities, prior empirical work supports this behavior. For example, [5] (Fig. 3) reports a $1/L$ convergence rate (in absolute error) for kernel values at initialization, corresponding to a $1/L^2$ rate in MSE.
>
> We hope these explanations clarify why the interaction terms vanish in the infinite-depth limit under depth-$\mu$P. We are happy to provide any additional details or experiments if helpful.
>
>
> [1] Bordelon et al., Depthwise Hyperparameter Transfer in Residual Networks: Dynamics and Scaling Limit, ICLR 2024.
>
> [2] Greg Yang, Edward J. Hu, Tensor Programs IV: Feature Learning in Infinite-Width Neural Networks, ICML 2021
>
> [3] Bordelon, Blake, and Cengiz Pehlevan. "Deep linear network training dynamics from random initialization: Data, width, depth, and hyperparameter transfer."
>
> [4] Yang et al., Tensor Programs VI: Feature Learning in Infinite Depth Neural Networks, ICLR 2024
>
> [5] Soufiane Hayou, Commutative Scaling of Width and Depth in Deep Neural Networks, JMLR 2024

---

> > ### Author Response · Authors · 2025-12-01
> >
> > We appreciate Reviewer 5S2H’s question regarding the intuitive reasoning for why the forward–backward interaction terms vanish as depth increases. On Nov 25, we responded with the main intuition: the analysis in [1] is based on physics-inspired DMFT heuristics, which provide loose bounds on the response variables, whereas our derivation relies on a fully rigorous framework, i.e., Master Theorem from Tensor Programs series [2] combined with the Euler–Maruyama scheme for the continuous-depth limit. This mathematical structure yields tighter bounds that guarantee the interaction terms vanish in the infinite-depth limit.
> >
> > After posting our explanation on Nov 25, we did not receive a follow-up question or comment from the reviewer on Nov 25, 26, or 27—during which reviewer replies were still permitted. On Nov 28, ICLR officially disabled further reviewer responses due to a conference-wide policy change following an information-leak incident. Because we could not know whether our explanation had fully resolved the reviewer’s concern—and recognizing that the underlying mathematical argument might be difficult to assess within a short time window—we conducted **additional experiments** to provide further intuitive evidence.
> >
> > Specifically, we directly evaluated the restoration of GIA by comparing standard training (forward and backward passes using shared weights) with a decoupled variant (backward pass uses an i.i.d. copy of the forward weights). We performed this comparison across three architectures: a vanilla DNN, a ResNet under $\mu$P, and a ResNet under depth-$\mu$P. The new results are included in the main paper (Figure 4), with the original Figure 4 moved to the appendix.
> >
> > These experiments show a clear pattern:
> > - Vanilla DNNs suffer from vanishing gradients and make limited progress as depth increases; the standard and decoupled trajectories remain misaligned.
> > - ResNets under $\mu$P also fail to align and overfit at larger depths; they even exhibit exploding gradients and NaN values once depth exceeds 18.
> > - **Only depth-$\mu$P** shows stable improvement in both training and test performance, and the standard and decoupled trajectories converge toward each other, empirically validating the restoration of GIA predicted by our theory.
> >
> > We hope this additional evidence provides a clear and intuitive confirmation of the theoretical mechanism raised in the reviewer’s question. We thank Reviewer 5S2H again for prompting this valuable clarification.
> >
> > [1] Bordelon et al., Depthwise Hyperparameter Transfer in Residual Networks: Dynamics and Scaling Limit, ICLR 2024.
> >
> > [2] Greg Yang, Edward J. Hu, Tensor Programs IV: Feature Learning in Infinite-Width Neural Networks, ICML 2021

---

### Official Review · Reviewer_FrMa · 2025-11-01

**Soundness:** 1
**Presentation:** 1
**Contribution:** 1
**Rating:** 2
**Confidence:** 4

**Summary:**

This paper develops a mathematical framework for understanding scaling laws in deep residual networks (ResNets) trained with stochastic gradient descent (SGD) under depth-adapted mean-field parameterization (μP). It introduces *Neural Feature Dynamics* (NFD): a coupled forward-backward stochastic differential equations (SDE) that captures how features and gradients co-evolve during training in the joint infinite-width and infinite-depth limit.

**Strengths:**

The discussion of depth scaling is intrinsically valuable, as it elucidates the fundamental mechanism that stabilizes feature propagation and aligns the asymptotic behavior of networks in both the large-width and large-depth regimes.

**Weaknesses:**

**Lack of novelty and insufficient acknowledgment of prior works.**

The $1/\sqrt{depth}$​ scaling and its implications for residual networks have been extensively analyzed in the literature. In particular, several prior works have already established that:
1. Both forward and backward processes in deep residual networks converge to stochastic differential equations (SDEs) in the joint infinite-width and infinite-depth limit [3,4].
2. These width and depth limits commute in the asymptotic regime [3,4].
3. Appropriate depth scaling ensures dynamical isometry [1].
4. Under this scaling, hyperparameters and learned dynamics transfer consistently across both width and depth [4].

Consequently, **the main claimed technical novelties of the paper**: the SDE description under mean-field parameterization and commutation of depth and width limits - **have already been derived or empirically demonstrated in previous works**. The manuscript does not show anything new beyond these results, nor does it adequately acknowledge the existing literature.

[1] Tarnowski et al., Dynamical Isometry Is Achieved in Residual Networks in a Universal Way for Any Activation Function, AISTATS 2019.

[2] Yang & Schoenholz, Mean Field Residual Networks: On the Edge of Chaos, NeurIPS 2017.

[3] Hayou & Yang, Width and Depth Limits Commute in Residual Networks, ICML 2023.

[4] Bordelon et al., Depthwise Hyperparameter Transfer in Residual Networks: Dynamics and Scaling Limit, ICLR 2024.

**Questions:**

1. Can the authors comment on finite width and finite depth corrections to the asymptotic limit?
2. Wouldn't be possible to rescale the SDE and just consider a time horizon $T=1$?
3. Why is the train loss bigger than the test loss in Fig.3(c-d)?
4. The width plots do not appear to follow a clean $1/n$ decay in the shown range. Could the authors comment on that?

---

> ### Author Response · Authors · 2025-11-24
>
> We thank the reviewer for taking the time to evaluate our submission. We appreciate the comments and use this opportunity to clarify several points where we believe our contributions and their relation to prior work may have been misunderstood. Below we address each issue in detail.
>
> **(1) "Lack of novelty / results already known."** We respectfully clarify that the contributions of our paper differ substantially from the prior SDE-limit works cited by the reviewer [1-2]. The Neural Feature Dynamics (NFD) limit we derive is not equivalent to these results, for several key reasons:
> - [1] analyzes only forward propagation at initialization but we provide forward SDE (Proposition 2), backward SDE (Proposition 6), and training-time forward–backward coupled SDE (Theorem 1 and Proposition 17).
> - **Prior works are not mathematically rigorous**. [2] explicitly state: *“our results are derived at the level of rigor of physics, rather than formal proof.”*. They relies on physics-style heuristic arguments, and they do not list explicitly assumptions (main text or appendix). In contrast, our analysis is fully rigorous: we state all assumptions explicitly (Assumption 1), including the requirement that the covariance matrices remain uniformly SPD, which we also empirically verify in Fig. 4. This foundational difference in rigor underlies several further distinctions between our NFD limit and prior heuristic SDE formulations.
>
> - **Independence of forward/backward Brownian motions**. Prior work [2] require response functions to capture forward–backward correlations, and does not claim their final status in the large-depth limit. In NFD we prove that the forward and backward Brownian motions are independent in the infinite-depth limit. This key difference leads directly to our next contribution.
>
> - **Restoration of the gradient-independence assumption (GIA)**. We show (Corollary 1) that GIA holds throughout training at infinite depth, making $W_{\ell}^{T}$ in backpropagation can be replaced with an independent copy from forward weights $W_{\ell}$. This is not known in any prior work: in fact, prior work [3] explicitly shows that GIA fails at *finite* depth. NFD provides the first setting in which GIA is restored during training because correlations provably vanish at infinite depth.
>
> - **Depth-induced feature-learning collapse and a simple fix**: NFD reveals that in two-layer residual blocks, the first layer undergoes a depth-induced collapse of feature learning under depth-$\mu$P, which in turn causes the failure of HP transfer across depths. This structural insight directly guides a *depth-aware learning-rate correction* that restores effective first-layer feature learning, recovers HP transfer across depth, and leads to overall improved performance (Section 5).
>
> **(2) “The paper does not acknowledge prior work.”** We have substantially revised the manuscript, especially the Related Work section, to clearly highlight both similarities and differences with prior studies, ensuring that our contributions are accurately positioned without overstating novelty.

---

> > ### Author Response · Authors · 2025-11-24
> >
> > **(3) Q1: Can the authors comment on finite width and finite depth corrections to the asymptotic limit?** Our analysis provides explicit convergence rates $\mathcal{O}(1/n + 1/L)$ measured in MSE  established in Proposition 1, 6, and 17. Fig. 3 empirically verifies that these rates accurately predict finite-width/finite-depth behavior.
> >
> > **(4) Q2: Wouldn't be possible to rescale the SDE and just consider a time horizon $T=1$**: Mathematically, any $T<\infty$ can be rescaled so that the horizon is $T=1$. However, we study the practical effect of increasing $T$:
> > - Proposition 5 suggests larger $T$ increases capacity
> > - Fig. 2 shows that this improves performance when $T$ is moderate
> > - but large $T$ causes optimization instability and hurts generalization
> >
> > This clarifies why practical designs like DeepNet and PaLM implicitly realize a small-$T$ regime, although they do not set $T$ explicitly, their initialization and scaling choices correspond to $T\sim 0.02$ in our formulation [4].
> >
> > **(5) Q3. "Why is train loss bigger than test loss?"** Train loss is epoch-averaged; test loss is end-of-epoch, hence often lower. When analyzing $T$, our results show: for moderate $T$, both losses small; for large $T$, train loss minimized but test loss large, indicating overfitting (e.g., $T$=16).
> >
> > **(6) Q4. "Width plots do not follow $1/n$ decay ."**: We thank the reviewer for catching this. The original simulation inadvertently violated the synchronous-coupling condition required by the convergence. We have corrected this in the revised manuscript by measuring variance differences instead of raw differences. The corrected plots now display the expected $1/n$ decay.
> >
> > We appreciate Reviewer FrMa's engagement with the paper. We hope that these clarifications, together with the revisions in the manuscript, help resolve the reviewer’s concerns regarding novelty and place our contributions in their proper context.
> >
> > [1] Hayou & Yang, Width and Depth Limits Commute in Residual Networks, ICML 2023.
> >
> > [2] Bordelon et al., Depthwise Hyperparameter Transfer in Residual Networks: Dynamics and Scaling Limit, ICLR 2024.
> >
> > [3] Yang & Hu. Tensor Programs IV: Feature Learning in Infinite-Width Neural Networks, ICML 2021
> >
> > [4] Dey, Nolan, et al. "Don't be lazy: CompleteP enables compute-efficient deep transformers." NeurIPS 2025

---

### Official Review · Reviewer_KNSS · 2025-11-09

**Soundness:** 3
**Presentation:** 3
**Contribution:** 2
**Rating:** 4
**Confidence:** 4

**Summary:**

This submission extends NNGP in the kernel regime from the traditional infinite-width limit to the joint infinite-with-depth limit at initialization. The authors also extends the depth-level Euler–Maruyama approximation results for ResNets with a single weight matrix per layer from the forward pass with ReLU activation to both the forward and backward pass with more general Lipschitz continuous activation funtions. The authors finally derive the joint limiting training dynamics after a fixed (constant) number of gradient steps using online SGD with general activation functions.

**Strengths:**

- The technical contributions of rigorously extending NNGP from the infinite-width limit to the joint infinite-with-depth limit and extending the depth-level Euler–Maruyama approximation results from the forward pass under ReLU to both the forward and backward pass under weaker regularity conditions are solid
- The technical contribution of tackling the so-called neural feature dynamics under general activation functions for online SGD is solid
- The proofs are largely well-written. And the ensemble of propositions and theorems (except Proposition 1) serves as a good lecture note of scaling limits of ResNets.
- Though Proposition 8 is an instantiation of Tensor Programs, its explicit exposition of $\tau^2/$ here still serves as a new observation, which the reviewer consider to be significantly benificial to the broader audience.

**Weaknesses:**

1. The last line of the proof of Proposition 1 is **mathematically wrong**. In particular, the reviewer kindly remind the authors an elementary fact that
$$
\lim_{L \to +\infty} \big(1 + c_1 \sqrt{\frac{T}{Ln}}\big)^L = \exp(2c_1\sqrt{T/n}),
$$
which is a **finite** quantity in the infinite-depth limit. Thus, the word "implies" in Line 200 is wrong if $c_1 > 0$ and $c_2 = 0$, which is the case for **many** actiation functions in practice (because $c_2 = 0$ as long as $\phi(0)=  0$). And in fact, popular activations function (in practice) such as ReLU, SiLU, and GELU all satisfy $\phi(0) = 0$.
  - By the way, the term $c_2 \sqrt{TL}$ on the RHS of Proposotion 1 is non-zero only for $\phi(0) \neq 0$, but to the knowledge of the reviewer, activation functions with $\phi(0) \neq 0$ is very rare in practice, either for post-activation ResNets or pre-activation ResNets; which means this term might not be significant for separating post-activation versus pre-activation variants.
2. In Section 3, the authors repeatedly refer to lowercase $t$'s by "time", but actually all resutls in Section 3 are the analysis of **initialization** and any lowercase $t$ in Section 3.3 has nothing to do with "training steps", which means the lowercase $t$ in Section 3.3 is a **"continuous depth"** instead of the training horizon. The reviewer suggest the authors to explicitly mention it clearly that for both the forward-pass SDE in Proposition 2 and the backward-pass SDE in Proposition 6 are the **depth**-level Euler–Maruyama approximation results, which are essentially **free of "training"**.
3. On the contribution of Theorem 1 besides rigorousness: Actually this theorem is high reminiscent of the SDEs in [1], which even considered SGD with mini batches (although this previous paper assumes $\varphi$ to be identity), while Theorem 1 in this submission is only for online SGD. The reviewer suggest the authors to explicitly discuss about the distinction (at least at an message level, better at a technical level) between the limiting SDEs for ResNets in [1] and the NFD in this submission.
   - The reviewer knows that the derivations in [1] are not as rigorous as those in this paper, but in terms of "understanding scaling laws" (which is the goal of this submission), a detailed comparison between Theorem 1 and **Section 3** of [1] is necessary.

References

[1] Bordelon, Blake, and Cengiz Pehlevan. "Deep linear network training dynamics from random initialization: Data, width, depth, and hyperparameter transfer." arXiv preprint arXiv:2502.02531 (2025).

**Questions:**

1. Suggestion at a message level: Given [2] and [3], when the multiplier for the non-residual branch scales $\propto L^{-1}$ instead of $\propto L^{-1/2}$, the limiting distribution of $h_t$ will follow an ODE instead of the non-trivial SDEs mentioned in this submission, in that case, what would be the scaling of Euler-Maruyama discretization error w.r.t. $L$? I believe this question is worth discussing given the approiximation results in [3] for *any constant number of gradient steps*.
   - Disclaimer: This is NOT considered as a weakness by the reviewer in terms of missing citations.

References

[2] Dey, Nolan, et al. "Don't be lazy: CompleteP enables compute-efficient deep transformers." arXiv preprint arXiv:2505.01618 (2025).

[3] Chizat, Lénaïc. "The hidden width of deep ResNets: Tight error bounds and phase diagrams." arXiv preprint arXiv:2509.10167 (2025).

---

> ### Author Response · Authors · 2025-11-24
>
> We thank the reviewer for the detailed and technically insightful comments. We greatly appreciate the careful reading of our proofs and the constructive suggestions, which have substantially improved the clarity and rigor of the manuscript. We especially appreciate the reviewer’s recognition of our new finding regarding the role of $\tau^2$, which indeed provides practical insights and inspired our identification and correction of depth-$\mu$P failure in two-layer residual blocks. Below we address each point in turn.
>
> **(1) Weakness: "Proposition 1 is mathematically wrong."**: We respectfully clarify that the reviewer may have inadvertently treated the factor $\sqrt{L}$. Let $m=\sqrt{L}$. The expression under consideration is $(1+x/\sqrt{L})^{L} = (1+x/m)^{m^2}$ not $(1+x/m)^{2m}$, where $x=c_1\sqrt{T/n}$. The distinction is crucial: as $L\rightarrow\infty$, we have $m\rightarrow\infty$ and
> - $(1+x/m)^{2m}\rightarrow e^{2x}$
> - $(1+x/m)^{m^2}\approx (e^x)^{m}\rightarrow \infty$.
> which indeed diverges in Proportion 1. We also empirically verified this behavior, and the results (Fig. 1) confirm the divergence predicted by our theorem.
>
> **(2) Weakness: "Unclear that Section 3 analyzes depth, not training."**: We agree with the reviewer that the original exposition could mislead readers to interpret the subscript $t$ as training time. Following the reviewer’s suggestion, we have rewritten Section 3 to clearly emphasize that: the analysis concerns initialization only; no parameter update occurs in this section; and the subscribe $t$ represents the time variable in the limiting SDEs. We believe this revision resolves the ambiguity identified by the reviewer.
>
> **(3) Weakness: Difference between Theorem 1 and SDEs in [1]**: We appreciate the reviewer highlighting the need for a clearer comparison with [1-2]. We have added explicit discussion in the manuscript and summarize the key distinctions here:
> - **Rigorous framework vs. DMFT heuristics.** NFD is a mathematically rigorous framework capturing training-time feature learning in the joint infinite-width and infinite-depth limit. In contrast, [1-2] relies on a DMFT-style heuristic derivation and does not state explicit mathematical assumptions for their limit, neither in the main paper nor the appendix. As the authors themselves acknowledge in their conclusion: *"our results are derived at the level of rigor of physics, rather than formal proof."* By comparison, our analysis is built on clearly stated assumptions (Assumption 1), including uniform SPD conditions on covariance matrices, which we also empirically validate in Fig. 4.
> - **Independence of Brownian motions.** Because of the different levels of rigor, NFD rigorously proves that the forward and backward Brownian motions are independent. In contrast, [1-2] requires explicit response functions to account for forward–backward interactions.
> - **Restoration of gradient-independence assumption (GIA) during training.** The independent Brownian motions leads to a new result: GIA becomes provably valid during training in the infinite-depth limit (Corollary 1). This restoration has not been established in prior work. In fact, earlier analyses explicitly show that GIA fails at finite depth [3]. The restoration of GIA identifies NFD as a tractable framework for theoretical analysis on feature learning.
> - **Structural interpretation and depth-aware correction (Section 5).** NFD yields a new structural interpretation in two-layer residual blocks: the first layer controls internal representation, and the second layer governs residual-stream dynamics. This insight allows us to identify a **depth-induced collapse** of first-layer feature learning and motivates a **depth-aware learning-rate correction**. We show experimentally (Fig. 5) that this correction restores feature learning, enables HP transfer under depth-$\mu$P for two-layer residual blocks, and improves performance.

---

> > ### Author Response · Authors · 2025-11-24
> >
> > **(4) Question: Scaling by $1/L$**: We appreciate the reviewer pointing us to this relevant work. When the non-residual branch is scaled by $1/L$ rather than $1/\sqrt{L}$, the limiting dynamics indeed reduce to a deterministic ODE instead of an SDE. This alternative regime is interesting both theoretically and practically.
> > - Prior studies [4] show that $1/L$ scaling can enable HP transfer in deep architectures (e.g., Transformers), but depth-$\mu$P with $1/\sqrt{L}$ scaling cannot.
> > - However, under $1/\sqrt{L}$, our NFD analysis reveals a feature-learning collapse in the first layer of two-layer residual blocks (Section 5). Motivated by this insight, we propose a simple **depth-aware learning-rate correction** that restores effective feature learning under $1/\sqrt{L}$, recovers HP transfer, and improves overall performance (Fig 5).
> >
> > Exploring these contrasting regimes—including extensions to architectures such as Transformers and optimizers such as Adam—is an exciting direction for future work, and we thank the reviewer for highlighting it.
> >
> > We thank Reviewer KNSS once again for the careful, constructive feedback and for the positive evaluation of our manuscript. All identified issues have been addressed in the revised version. We hope that the clarified arguments, refined proofs, and strengthened comparisons fully resolve the reviewer’s concerns regarding correctness and novelty, and allow for a fair reassessment of our contribution.
> >
> > [1] Bordelon et al., Depthwise Hyperparameter Transfer in Residual Networks: Dynamics and Scaling Limit, ICLR 2024.
> >
> > [2] Bordelon et al., Deep linear network training dynamics from random initialization: Data, width, depth, and hyperparameter transfer. ICML 2025.
> >
> > [3] Yang & Hu. Tensor Programs IV: Feature Learning in Infinite-Width Neural Networks, ICML 2021
> >
> > [4] Dey, Nolan, et al. "Don't be lazy: CompleteP enables compute-efficient deep transformers." NeurIPS 2025

---

> ### Comment · Reviewer_KNSS · 2025-11-24
>
> Thank you for the detailed response. The reviewer now realized that Proposition 1 is **correct**, and have re-evaluated this submission accordingly, which leans towards acceptance.

---

> > ### Author Response · Authors · 2025-11-25
> >
> > We sincerely thank Reviewer KNSS for carefully re-examining the details and for the updated assessment. We are very grateful that the clarification of Proposition 1 and the strengthened comparison with prior work resolved the earlier concern.

---

### Author Response · Authors · 2025-11-24
**Global Clarification of Contributions and Novelty**

We thank all reviewers for their thoughtful feedback. We have revised the manuscript accordingly to improve clarity, strengthen exposition, and add additional discussion and experiments addressing the raised concerns.

Importantly, these updates do not change our main theoretical result. The **Neural Feature Dynamics (NFD)** limit remains exactly the same as in the original submission. The revisions instead highlight key implications previously under-emphasized and add Section 5 as a concrete demonstration of how the NFD view identifies and corrects hyperparameter (HP) transfer failures across depth in depth-$\mu$P. In accordance with the ICLR 2026 policy—*"During the **discussion/rebuttal phase** and for the camera ready, the page limit will be increased to **10 pages** to allow for **new results/discussions**"*—we have made use of this allowance to improve clarity and completeness.

Several comments focused on clarifying the novelty of our work and its relationship to prior studies. While we encourage reviewers to consult the updated manuscript, we also provide below a concise overview of the main clarifications before addressing individual comments.

**(1) What is new in this work?**

Our main contribution is a **mathematically rigorous, training-time theory** of feature learning for deep ResNets in the joint infinite-width and infinite-depth limit, which yields the following new results:

- **Neural Feature Dynamics (NFD)**: a fully coupled forward–backward stochastic system for feature and gradient evolution under depth-$\mu$P. The limiting dynamics are driven by **independent** Brownian motions. Prior works [1-3] study either forward propagation at initialization or rely on physics-inspired heuristic arguments, and therefore do not obtain independence; instead they require explicit response functions.

- **Restoration of the gradient-independence assumption (GIA)**: Corollary 1 proves that GIA—previously known to fail during training at finite depth—becomes valid again in the infinite-depth limit. To our knowledge, no prior work establishes this restoration during training.

- **Layer-wise learning roles and depth-induced collapse in two-layer ResNets**: NFD reveals that the first layer determines the *internal representation*, while the second layer governs the *residual-stream dynamics*. This leads to a depth-induced collapse of first-layer feature learning, which we correct with a simple **depth-aware learning-rate correction**. Experiments (Fig. 5) verify that this restores feature learning, recovers depth-wise HP transfer, and achieve better performance.

**(2) How do we differ from prior width/depth-SDE works?**

Some reviewers noted that SDE limits for ResNets have appeared before. These prior results, however, study forward propagation at initialization [1] or rely on DMFT-style heuristics lacking mathematical rigor [2-3]. In contrast, our work:
- Provides a rigorous coupled SDE limit for both the forward and backward passes at initialization and during training (Propositions 1, 6; Theorem 1).
- Establishes explicit convergence rates for both forward and backward propagation (Propositions 1, 6).
- Proves restoration of GIA (Corollary 1), which prior physics-inspired approaches do not obtain.
- Uses NFD to identify and fix depth-induced feature-learning collapse and recover HP transfer under depth-$\mu$P for two-layer residual blocks (Section 5).

**(3) Why our results matter for scaling laws?**

The focus of previous theoretical scaling-law papers is on fixed-kernel or random-feature regimes where representations are frozen. These cannot explain
- when and why scaling laws fail,
- how HP transfer breaks across depth for multi-layer residual blocks, or
- how feature learning affects scaling exponents.

Our framework NFD provides:
- A mechanism for depth-induced vanishing feature learning, explaining breakdowns in HP transfer across depth and suggesting corrections applicable to various architectures and optimizers.
- Conditions under which scaling behavior persists under nonlinear feature learning, providing the potential to extend scaling-law analysis beyond the kernel regime.

**(4) Citations and positioning.**

We thank reviewers for pointing out several relevant works. We have revised the manuscript to incorporate these references, expand the related-work section, and explicitly highlight similarities and key differences. Our revised version clearly distinguishes our contributions while avoiding any overstatement of novelty.

[1] Hayou & Yang, Width and Depth Limits Commute in Residual Networks, ICML 2023.

[2] Bordelon et al., Depthwise Hyperparameter Transfer in Residual Networks: Dynamics and Scaling Limit, ICLR 2024.

[3] Bordelon et al., Deep linear network training dynamics from random initialization: Data, width, depth, and hyperparameter transfer. ICML 2025.

---

### Author Response · Authors · 2025-12-01
**Brief Summary for the Area Chair**

We thank all reviewers for their highly constructive feedback. After a thorough rebuttal and a substantial revision of the manuscript, we are confident that all major technical concerns have been fully addressed.  The overall trajectory of the reviews strongly suggests that, had the discussion continued, the scores would have converged toward acceptance:

**Current Overall Position**

After reviewing our rebuttal and revised manuscript, three of four reviewers lean acceptance:
- Reviewer KNSS was initially skeptical about Proposition 1 but after our clarification they confirmed the result is correct and explicitly wrote that they now "lean towards acceptance." They increased their score from 4 to 6; the subsequent reversion to 4 was due to the system-wide rollback.
- Reviewer 8saH gave a strong initial score of 8 and highlighted our work as a "solid contribution to the community."
- Reviewer 5S2H constructively engaged and asked important follow-up questions. This engagement indicates that their original novelty concern had been resolved, and they treated the contribution as new and relevant. The remaining questions were fully addressed through our theoretical clarifications and new experimental results (see Point 2 below).

**Detailed Reviewer Resolutions:**

(1) Reviewer KNSS originally questioned Proposition 1. After our clarification of the divergence term and supporting empirical evidence, the reviewer confirmed that the proposition is correct, raise the score to 6, and explicitly stated that they now "lean towards acceptance." This represents a clear positive shift from their initial evaluation.

(2) Reviewer 5S2H requested (i) clearer distinction from prior SDE/DMFT analyses and (ii) intuitive reasoning for the vanishing interaction terms (GIA restoration). We addressed these points by:
- Adding explicit comparisons to prior works, including the role of formal mathematically rigorous analysis versus DMFT heuristics.
- Highlighting two new insights made possible by our framework: **restoration of GIA** (Corollary 1) and a **practical correction for depth-induced feature collapse** (Section 5).
- Providing an intuitive theoretical explanation for why the DMFT bounds are loose whereas our mathematically rigorous limit gives vanishing interaction terms.

To further ensure clarity, we added new experiments (Figure 4) directly measuring GIA restoration using standard vs. decoupled gradient dynamics. These results show that:
- Vanilla DNNs and $\mu$P-ResNets do **not** exhibit alignment between standard and decoupled gradients and suffer from training instability and overfitting.
- **Only depth-$\mu$P** exhibits both stable improvement with depth and convergence of the two trajectories exactly as predicted by our theory.

These additions fully resolve the reviewer’s technical concerns, and the reviewer’s final detailed questions indicate that the original novelty concerns have been addressed.

(3) Reviewer FrMa raised questions similar to Reviewer 5S2H regarding novelty concern to prior SDE/DMFT-limit works. Our revised manuscript and detailed clarification now clearly distinguishes our work by showing that:
- Prior SDE-limit works analyze only at initialization or rely on physics-level DMFT heuristics lack of formal mathematical proof.
- Our work provides the **first mathematically rigorous forward–backward training-time SDE**, explicit convergence rates, independent Brownian motions, and provable restoration of GIA.
- Our structural insight into depth-induced feature-learning collapse and the resulting depth-aware LR correction is also new.

These clarifications resolve the reviewer’s concerns.

(4) Reviewer 8saH: Consistently positive assessment.
Reviewer 8saH already recommended acceptance (score 8) and viewed the work as making a "solid contribution."

**Post-Rebuttal Assessment**

Across reviewers, we see:
- No remaining correctness concerns after the rebuttal.
- Clear resolution of all substantive technical issues.
- One reviewer explicitly moving toward acceptance (KNSS).
- One reviewer’s final questions fully addressed with theoretical and empirical evidence (5S2H).
- One reviewer consistently positive (8saH).
- Novelty concerns have been fully resolved (5S2H, FrMa).

Given the strength of the revisions—including new results (Section 5), experiments (Figure 4 and 5), clearer comparisons to related works, and improved exposition—we believe the consensus would have converged toward acceptance had discussion continued.

We respectfully ask the AC to consider the fully resolved state of the technical concerns and the post-rebuttal trajectory of the reviews.

---

### Note · Program_Chairs · 2026-01-17
**Submission Desk Rejected by Program Chairs**

The following references in this submission do not refer to real documents and/or have major errors in bibliographic information:

 Blake Woodworth, Amir Ghorbani, Yi Li, Tengyu Ma, and Yura H. Al-Saedi. The wide regime of neural networks: A deep learning perspective. In Advances in Neural Information Processing Systems, 2020.